# Estimation of stream water components and residence time in a permafrost catchment in the central Tibetan Plateau using long-term water stable isotopic data

**Shaoyong Wang**[1,2,★], **Xiaobo He**[1,★]**, Shichang Kang**[1,2]**, Hui Fu**[3]**, and Xiaofeng Hong**[4]

[1]State Key Laboratory of Cryospheric Science, Northwest Institute of Eco-Environment and Resources, Chinese Academy of Sciences, Lanzhou 730000, China
[2]College of Resources and Environment, University of Chinese Academy of Sciences, Beijing 100049, China
[3]State Key Laboratory of Simulation and Regulation of Water Cycle in River Basin, China Institute of Water Resources and Hydropower Research, Beijing 100038, China
[4]Water Resources Department, Changjiang River Scientific Research Institute, Wuhan 430010, China
[★]These authors contributed equally to this work.

**Correspondence:** Xiaobo He (hxb@lzb.ac.cn)

**Abstract.** Global warming has significantly impacted the hydrological processes and ecological environment in permafrost regions. Mean residence time (MRT) is a fundamental catchment descriptor that provides hydrological information regarding storage, flow pathways, and water source within a particular catchment. However, water stable isotopes and MRT have rarely been investigated due to limited data collection in the high-altitude permafrost regions. This study uses the long-term stable isotopic observations to identify runoff components and applied the sine-wave exponential model to estimate water MRT in a high-altitude permafrost catchment (5300 m a.s.l.) in the central Tibetan Plateau (TP). We found that the isotope composition in precipitation, stream, and supra-permafrost water exhibited obvious seasonal variability. The freeze–thaw process of the permafrost active layer and direct input of precipitation significantly modified the stable isotope compositions in supra-permafrost and stream water. The hydrograph separation revealed that precipitation and supra-permafrost water accounted for $35 \pm 2\%$ and $65 \pm 2\%$ of the total discharge of stream water, respectively. MRT for stream and supra-permafrost water was estimated at 100 and 255 d, respectively. Such shorter MRTs of supra-permafrost and stream water (compared to the non-permafrost catchments) might reflect the unique characteristics of the hydrological process in permafrost catchments. Moreover, the MRT of supra-permafrost water was more sensitive to environmental change than that of stream water. Climate and vegetation factors affected the MRT of stream and supra-permafrost water mainly by changing the thickness of the permafrost active layer. Our results suggest that climate warming might retard the rate of water cycle in permafrost regions. Overall, our study expands our understanding of hydrological processes in high-altitude permafrost catchments under global warming.

## 1 Introduction

Permafrost, considered among the regions that are most sensitive to global warming (Wang et al., 2009), acts as an aquiclude which governs the surface runoff and its hydraulic connection with groundwater and alters groundwater flow paths due to its thawing (Woo, 1990; Hinzman et al., 2005; Sjöberg et al., 2021). The spatiotemporal variability in permafrost freeze–thaw cycles alongside the development of an active layer influences the catchment hydrology in a time-dependent manner, thereby affecting parameters such as soil water movement direction, storage capacity, and hydraulic conductivity (Tetzlaff et al., 2018; Gao et al., 2021). Over the

past decades, permafrost has experienced significant, rapid, and extensive degradation which have profoundly and extensively affected regional and even continental hydrological regimes, as well as alpine ecology (Cheng et al., 2019; Jin et al., 2021). Permafrost degradation has released valuable meltwater for recharging groundwater (Ma et al., 2019a). This degradation, accompanied by a progressive increase in the active layer, could raise water content in the soil column, thereby increasing the groundwater storage capacity (Woo, 1990; Hinzman et al., 2005; Woo et al., 2008). Thus, compared to non-permafrost regions, the presence of an aquiclude and the freeze–thaw cycle of the permafrost active layer contribute to more complex hydrological processes in permafrost regions.

The mechanisms of runoff processes in permafrost regions (e.g. flow pathways, water source, and residence time) have attracted attention (Tetzlaff et al., 2018; Carey and Quinton, 2005; Li et al., 2020a; Yang et al., 2016). Isotope hydrograph separation is a technique that separates the contributions of new and old waters to stream hydrograph, which has been extensively used to track the runoff components and water flow in permafrost catchments (Taylor et al., 2002; Carey and Quinton, 2005; Li et al., 2020a; Yang et al., 2016). Mean residence time (MRT) refers to the average time required for input water to travel through water channels and to reach a catchment outlet in either a vertical or a horizontal flow path (McDonnell et al., 2010; Shah et al., 2017). MRT is a fundamental hydraulic descriptor that can be used to reveal information regarding water storage, flow paths, and water sources within a particular catchment (Shah et al., 2017; McGuire and McDonnell, 2006; Dunn et al., 2007). Furthermore, catchment water MRT is crucial for examining catchment response to environmental changes (McGuire et al., 2002). Therefore, by quantifying the MRT, the hydrological sensitivity to climatic changes can be identified (Zhou et al., 2021a). Moreover, this parameter provides new insights into catchment functions and runoff processes (Farrick and Branfireun, 2015). The influencing factors associated with MRT have been analysed by previous studies; however, most of them focused on the influencing factors of the catchment area, topographical factors, groundwater contribution, soil properties, and flow path length (Dunn et al., 2007; Ma et al., 2019b; Soulsby and Tetzlaff, 2008; Rodgers et al., 2005b). Therefore, the influence of permafrost changes, climatic factors, and vegetation variations on catchment MRT in a high-altitude permafrost catchment is rarely evaluated. For instance, Song et al. (2017) and Yang et al. (2021) have investigated water age in permafrost catchments in the hinterland of the Tibetan Plateau (TP) and explored the influence of vegetation and climatic factors on water age. The effects of permafrost freeze–thaw cycles on water MRT in Arctic permafrost catchments have also been reported (Tetzlaff et al., 2018). Meanwhile, given the complexities of the underlying surface alongside the remoteness and logistical difficulties associated with data collection in high-altitude

permafrost catchments, few studies have examined such permafrost catchments to estimate water MRT using isotopic tracers by long time series sampling.

The TP is the source region of many large Asian rivers and is often referred as the "water tower of Asia". However, the TP is the most vulnerable among the world's water towers under the changing climate in terms of the water-supplying role, the downstream dependence of ecosystems, and societal impacts (Immerzeel et al., 2020). The permafrost with the highest elevation and largest area in the mid-latitudes is mainly distributed in the TP, which has experienced significant and extensive degradation (Cheng et al., 2019). The climatic influences of MRT are non-stationary and manifested at an annual scale (Soulsby and Tetzlaff, 2008). Therefore, in this study, we estimated MRT using long-term water stable isotopic data (8-year data for stream water and 5-year data for supra-permafrost water) from a high-altitude (5300 m a.s.l.) permafrost catchment in the central TP. The main objectives of this study are to (1) characterize the isotope composition of the catchment water, (2) elucidate the potential drivers of isotope variations, (3) identify runoff components using two-component hydrograph separations, (4) estimate an approximation of water MRT of permafrost catchments using a sine-wave exponential model, and (5) quantify the climate and permafrost change effects on the permafrost hydrological process using the estimated catchment water MRT. The findings from our study will expand our understanding of the hydrological process in permafrost regions under global warming.

## 2 Materials and methods

### 2.1 Study area

This study was conducted in the Xiaoliuyu catchment located in the source region of the Yangtze River in the central TP, with an average altitude of 5300 m a.s.l. and a drainage area of 2.17 km$^2$, which falls under the transition zone between the monsoon and non-monsoon regions (Fig. 1). This study area is unique given the presence of extensive permafrost. In particular, the maximum active layer thickness of permafrost can be > 300 cm, according to the observational data. The area is covered by sedge and wormwood, including species such as *Kobresia capilifolia, Kobresia pygmaea*, and *Kobresia humilis* (L. Wang et al., 2020).

The climate in the study area features a cold and dry season from October to May and is primarily controlled by westerlies, while the warm and wet season from June to September is controlled by the Indian monsoon. The catchment receives > 90 % of the total annual precipitation during the warm and wet seasons (Li et al., 2016). The annual mean air temperature of this area is −5.3 °C, while the annual mean precipitation is 491.9 mm. The river in the catchment freezes during the cold season and only generates runoff dur-

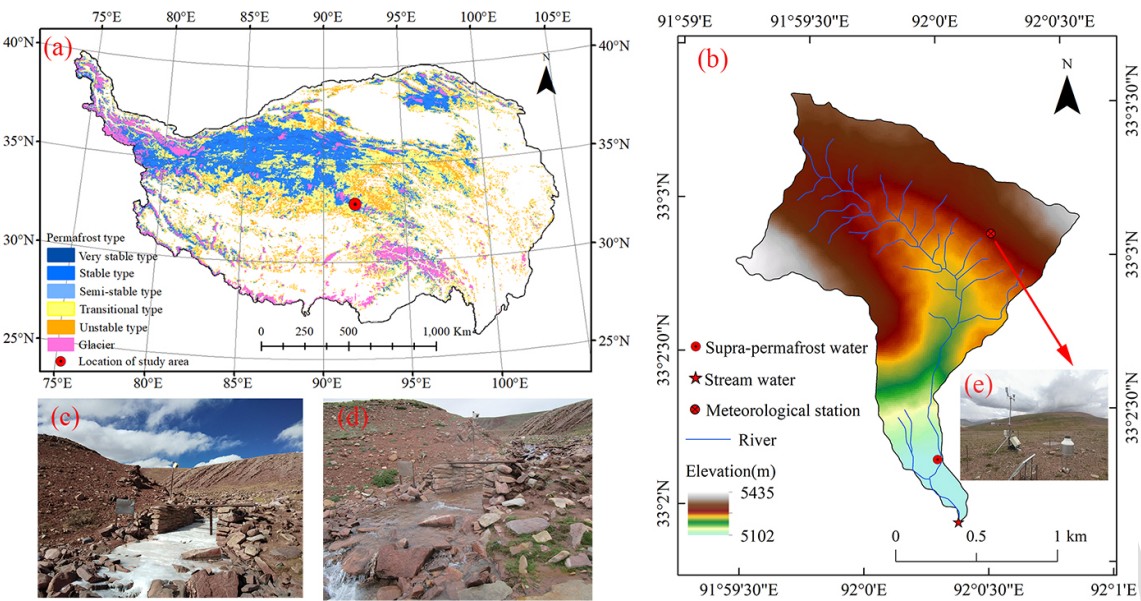

**Figure 1. (a)** Map of permafrost thermal stability on the Tibetan Plateau; **(b)** map of the study area and the sampling sites; **(c–d)** images of the frozen river in this catchment during the cold season and the runoff generated during the warm season. **(e)** Photograph of the underlying surface of the catchment and meteorological station. Thermal stability of permafrost distribution data are from the National Tibetan Plateau Data Center (TPDC) (http://data.tpdc.ac.cn/, last access: 13 September 2021) published by Ran et al. (2020). Digital elevation model (DEM) data available from Geospatial Data Cloud (http://www.gscloud.cn/, last access: 25 August 2021).

ing the warm season (see Fig. 1c and d). Figure 2 shows the daily variations in precipitation, air temperature, soil moisture, and temperature. The soil moisture and temperature data were measured at depths of 5, 20, 40, 60, 100, 160, 220, and 300 cm. The air temperature increases to $0\,°C$ in early May, indicating the onset of frozen soil thaw, while the peak air temperature is observed in July–August. Soil temperature is also an important factor that governs the freezing and thawing of soil water in permafrost regions. Notably, the variations in soil moisture agree with the variations in soil temperature in the region. In early May, soil temperature increases to $>0\,°C$, thereby rapidly increasing soil moisture and manifesting the onset of rapid thawing. These seasonal dynamics of the soil water content are altered by freeze–thaw processes in the active layer (Wang et al., 2009).

## 2.2 Permafrost and meteorological data

The soil temperature data (active layer bottom temperature) and active layer thickness of permafrost were obtained from the *Blue Book on Climate Change in China 2021*, which published long-term data related to the permafrost along the Kunlun Mountain to the southern slope of Tanggula Mountain in central TP (CMA Climate Change Centre, 2021). The normalized differential vegetation index (NDVI) can reflect the growth status and coverage of vegetation. We collected NDVI data from MOD13A2 products provided by NASA, USA, with temporal and spatial resolutions of 16 d and $1 \times 1\,km$, respectively (https://ladsweb.modaps.eosdis.

nasa.gov/, last access: 10 December 2022). In our study, the spatial resolution of NDVI data was unified to $25 \times 25\,m$ by resampling. The growing-season NDVI was determined using the maximum synthesis method. Meteorological data, including precipitation amount, as well as air temperature, are available from the weather stations within the Xiaoliuyu catchment (Fig. 1b). The soil temperature, active layer thickness of permafrost, NDVI, and the meteorological data were obtained for 2012–2020, as shown in Table 1.

## 2.3 Field sampling and isotope analysis

In this study, the continuous sampling and high sampling frequency of stream water, supra-permafrost water, and precipitation were conducted from June to October. The sampling sites are shown in Fig. 1, and detailed information of sampling sites is summarized in Table 2. In total, the 416 precipitation samples were collected during the observation period in bulk collectors at the Tanggula Cryosphere and Environment Observation Station (TaCOS), Chinese Academy of Sciences, at the altitude of 5050 m a.s.l. Liquid precipitation samples were collected immediately following every precipitation event using a bulk collector to minimize the effects of evaporation. Solid precipitation (snow) samples were collected in a plastic bag and taken to a warm place to be thawed, following which water samples were transferred into 50 mL polyethylene (PE) bottles. The groundwater in the permafrost region can be classified into three categories: supra-permafrost water, intra-permafrost water,

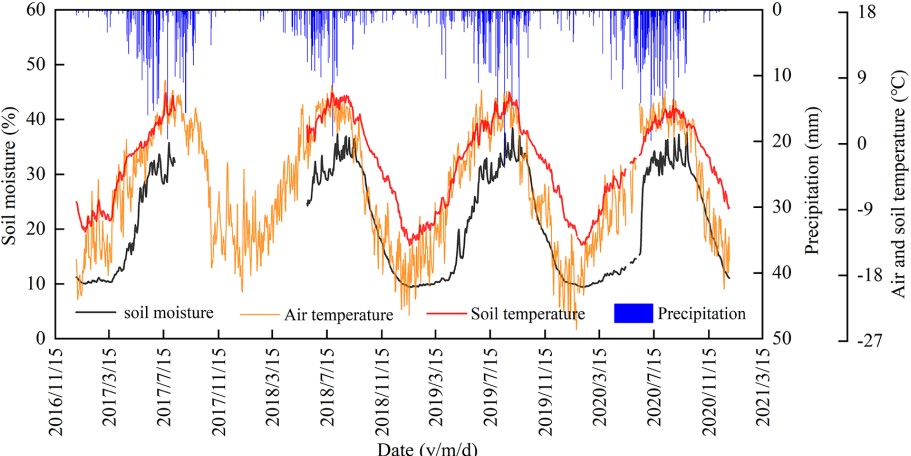

**Figure 2.** Daily variation in soil moisture, soil temperature, air temperature, and precipitation from 2017 to 2020. The soil moisture and temperature data were the daily average value of all the sensor data in the soil layer.

**Table 1.** Annual permafrost, normalized differential vegetation index (NDVI), and meteorological data from 2012–2020. The symbol "–" indicates no data were available for that year.

| Data type | 2012 | 2013 | 2014 | 2015 | 2016 | 2017 | 2018 | 2019 | 2020 |
|---|---|---|---|---|---|---|---|---|---|
| Active layer thickness (cm) | 223 | 229 | 228 | 237 | 240 | 240 | 245 | 243 | 237 |
| Soil temperature (°C) | −1.20 | −1.19 | −1.10 | −1.30 | −1.00 | −0.91 | −0.80 | −1.31 | −1.40 |
| Air temperature (°C) | −5.69 | – | −5.97 | −4.84 | −4.71 | −5.00 | −5.15 | −5.60 | −5.50 |
| Precipitation (mm) | 527 | 547 | 674 | 307 | 541 | 486 | 319 | 455 | 626 |
| NDVI | 0.2312 | 0.1336 | 0.1362 | 0.1842 | 0.1880 | 0.1771 | 0.1400 | 0.1807 | 0.1769 |

and sub-permafrost water (Gao et al., 2021; Cheng and Jin, 2013). Supra-permafrost water is the most widely distributed groundwater type in the permafrost regions of TP, which is mainly stored in the permafrost active layer (Li et al., 2020a). We collected 755 stream water samples and 296 supra-permafrost water samples at approximately 1 d intervals in the Xiaoliuyu catchment from June to October for each year (Fig. 1b). As the stream water and the permafrost layer are frozen due to low temperature in the cold season, the stream and supra-permafrost water is simply generated during the warm season. The supra-permafrost water samples were taken from a well that was drilled to a depth of 1.5 m. Given the logistical constraints in this study area, only a single stream and supra-permafrost water sampling site were collected.

All samples were collected in individual 50 mL PE bottles that were rinsed three times with the water from the source itself before sampling. The bottles were then sealed and stored in a refrigerator at a temperature of −10 °C to minimize the possibility of contamination from the external environment and isotopic fractionation induced by liquid water evaporation. Hydrogen ($\delta D$) and oxygen ($\delta^{18}O$) stable isotope compositions of all water samples were measured using the Liquid-Water Isotope Analyzer (DLT 100, Los Gatos, USA) at the State Key Laboratory of Cryospheric Sciences,

Chinese Academy of Sciences. The water samples were analysed six times. The first two results were discarded to eliminate "memory effects", and the average of the last four results was used as the isotope estimate of the water samples. The results were reported in per mil (‰) units relative to the Vienna Standard Mean Ocean Water (V-SMOW). The precision of measurement for $\delta^{18}O$ and $\delta D$ was ±0.2‰ and ±0.6‰, respectively.

### 2.4 Isotope hydrograph separation method

The isotopic hydrograph separation (IHS) method is based on a mass balance approach. It can be used to estimate the contributions of diverse potential water sources contributing to streamflow using isotopes ($\delta^{18}O$ or $\delta D$) as a tracer (Genereux, 1998; Uhlenbrook and Hoeg, 2003). The two-component method can be formalized using Eqs. (1) and (2), shown below:

$$Q_s = Q_p + Q_e, \tag{1}$$
$$Q_s C_s = Q_p C_p + Q_e C_e, \tag{2}$$

where $Q_s$, $Q_p$, and $Q_e$ represent stream water, pre-event water (supra-permafrost water), and event water (precipitation) volumes, respectively; $C_s$, $C_p$, and $C_e$ are the corresponding isotope values. In this study, the $\delta^{18}O$ data were used in the

**Table 2.** Information of samples and sampling sites.

| Sample type | Years | Altitude (m a.s.l.) | No. of samples |
|---|---|---|---|
| Precipitation | 2012, 2014–2020 | 5129 | 416 |
| Stream water | 2012, 2014–2020 | 5139 | 756 |
| Supra-permafrost water | 2012, 2017–2020 | 5146 | 296 |

two-component hydrograph separation. The contributions of pre-event water and event water to stream water can be calculated by combining Eqs. (1) and (2) to obtain Eqs. (3) and (4):

$$f_p = \frac{C_e - C_t}{C_e - C_p} \times 100\%, \tag{3}$$

$$f_e = \frac{C_t - C_p}{C_e - C_p} \times 100\%, \tag{4}$$

where $f_p$ and $f_e$ represent the relative contribution ratio of pre-event water and event water to stream water, respectively. In this study, the precipitation-weighted average of precipitation isotopes was used to assess stream water components on a monthly scale to determine the mechanism of the runoff process in permafrost catchments.

The uncertainty in hydrograph separations generally included two aspects: one is the analysis error of tracer concentrations, while the other is the spatial and temporal variations in the tracer of components (Uhlenbrook and Hoeg, 2003), calculated using the Gaussian error propagation technique (Genereux, 1998):

$$w_y = \sqrt{\left(\frac{\partial y}{\partial x_1} w_{x1}\right)^2 + \left(\frac{\partial y}{\partial x_2} w_{x2}\right)^2 + \ldots + \left(\frac{\partial y}{\partial x_n} w_{xn}\right)^2}, \tag{5}$$

where $w$ represents the uncertainty in the variable specified in the subscript, and $y$ is the contribution of a specific streamflow component $x$ to stream water.

### 2.5 Estimation of mean residence times

Earlier studies have demonstrated that the sine-wave exponential model can be used to estimate MRT using the seasonal variations in isotope composition in precipitation and stream water (McGuire et al., 2002; McGuire and McDonnell, 2006; Soulsby and Tetzlaff, 2008; Ma et al., 2019b; Zhou et al., 2021a). Mathematically, the movement of the conservative tracer through a catchment can be expressed by the convolution integral (McGuire and McDonnell, 2006; Simin et al., 2013), which states that the tracer concentration of output water $\delta_{out}(t)$ at any time and input tracer $\delta_{in}(t - \tau)$ that enter uniformly into the catchment in the past $t - \tau$, which becomes lagged by its transit time distribution $g(\tau)$. The common transit time distribution model $g(\tau)$ used in hydrologic systems include exponential, exponential piston flow, and dispersion models (Simin et al., 2013). The exponential model describes a catchment with flow times that

are exponentially distributed, including pathways with very short transit times (McGuire and McDonnell, 2006), which assumed that the system is in steady-state conditions and operates as a perfect mixer (Sánchez-Murillo et al., 2015; Chiogna et al., 2014). In our study, the exponential model was used to estimate water MRT:

$$\delta_{out}(t) = \int_0^\infty g(\tau)\delta_{in}(t - \tau)\,d\tau, \tag{6}$$

$$g(\tau) = \tau_m^{-1} \exp\left(\frac{\tau}{\tau_m}\right), \tag{7}$$

where $\tau$ is the transit time, $t$ is the time the tracer exits from the catchment, $t - \tau$ represents the time the tracer enters into the catchment, and $\tau_m$ is the mean residence time (MRT).

In our study, the seasonal variations in $\delta^{18}O$ in precipitation, supra-permafrost water, and stream water were modelled using a sine-wave function, defined by Eq. (8):

$$\delta(t) = X + A\left[\cos(ct - \theta)\right], \tag{8}$$

where $\delta(t)$ is the modelled $\delta^{18}O$ (‰), $X$ is the mean measured $\delta^{18}O$ (‰), $A$ is the amplitude of the measured $\delta^{18}O$, $c$ is the radial frequency constant given as $c = 2\pi/153\,d = 0.04105\,\text{rad}\,d^{-1}$ in our study (the total number of days from June to October considered in this study amounted to 153), and $\theta$ is the phase lag of modelled $\delta^{18}O$ in radians. The overall performance of the sine-wave model was evaluated using the goodness of fit ($R^2$) and root mean square error (RMSE).

The analytical solution of the MRT ($\tau_m$) for the exponential model can be derived by combining Eqs. (7) and (8) (McGuire and McDonnell, 2006):

$$MRT = c^{-1} \sqrt{\left[\left(\frac{A_z}{z}\right) \quad 1\right]}, \tag{9}$$

where $A_{z1}$ is the amplitude of precipitation $\delta^{18}O$, $A_{z2}$ is the amplitude of modelled $\delta^{18}O$ in supra-permafrost or stream water in our study, and $c$ is the radial frequency, defined in Eq. (8). The uncertainty of the MRT estimates were quantified by determining the 95 % confidence of the fitted sine wave's amplitude – specifically, applying the 95 % confidence of the fitted sine wave's amplitude to Eq. (9) producing MRT error (Morales and Oswald, 2020).

# 3 Results

## 3.1 Stable isotope composition in different waters

For a detailed examination of the differences in isotopic composition in the different waters and their mutual transformation, the relationships between $\delta^{18}O$ and $\delta D$ for precipitation, stream water, and supra-permafrost water were quantified (Fig. 3). During the sampling periods, the isotopic composition in precipitation ($-30.9\,‰$ to $2.9\,‰$ for $\delta^{18}O$; $-244.2\,‰$ to $34\,‰$ for $\delta D$) strongly varied compared with that in stream water ($-19.6\,‰$ to $-4.0\,‰$ for $\delta^{18}O$; $-146.3\,‰$ to $-39.8\,‰$ for $\delta D$) and in supra-permafrost water ($-17.5\,‰$ to $4.6\,‰$ for $\delta^{18}O$; $-127.6\,‰$ to $50.9\,‰$ for $\delta D$) (Table 3). This finding indicates that precipitation was an important recharge source for supra-permafrost and stream water in the analysed permafrost catchment. In addition, ice meltwater from deeper soil layers is also a source of water replenishment (Wang et al., 2022; Sugimoto et al., 2003; Throckmorton et al., 2016), although a previous study found that melting ground ice in permafrost made little contribution to the observed runoff variations (Landerer et al., 2010).

We identified the substantial difference among the water line in different water: the stream water (SWL: $\delta D = 6.66\delta^{18}O - 7.61\,‰$), supra-permafrost water (PWL: $\delta D = 6.10\delta^{18}O - 15.89\,‰$), and local meteoric water line (LMWL: $\delta D = 8.01\delta^{18}O + 13.48\,‰$). The lower slopes of SWL (6.66) and PWL (6.1) strikingly indicate the non-equilibrium fractionation caused by evaporation, reducing the slope of the $\delta D–\delta^{18}O$ correlation line (Throckmorton et al., 2016). Alternatively, we surmise that there could be precipitation mixed with water from previous events stored in the permafrost active layer. In contrast, supra-permafrost water exhibited the lower slope, which was most likely caused by thawing and freezing of the active soil layer (Wang et al., 2009). During freezing and thawing, with the thickening of the active layer, part of ice meltwater (old water) will be mixed with precipitation. This old water tends to have a older water age and is subjected to evaporation over a long period, thereby lowering the slope of water lines (Throckmorton et al., 2016; Song et al., 2017).

## 3.2 Temporal variations in stable isotopes in different waters and their potential drivers

The variations in $\delta^{18}O$ and $\delta D$ compositions in different water samples exhibited similar tendencies, as seen in Fig. 3. Therefore, $\delta^{18}O$ was selected as the representative isotope in the following analysis. The temporal variations in $\delta^{18}O$ in stream and supra-permafrost water are shown in Fig. 4. The analysis revealed that substantial seasonal variability in the stable isotope signature was identified in different waters. The precipitation isotopes deplete heavy isotopes during the period of the Indian summer monsoon (from late May to early September) and enrich heavy isotopes at the beginning and end of the summer monsoon period (late May and mid-October). This pattern is similar to those of the monsoon region of the southern TP due to the shifting moisture source between the Bay of Bengal and the southern Indian Ocean (Yao et al., 2013). The stream water samples also exhibited a depletion in the isotopic signature, which is virtually consistent with the precipitation results from the overall variation trend (Fig. 4). The isotope compositions in supra-permafrost water for the certain year (such as 2018 and 2019) also reflected the isotope signals of precipitation but exhibited the least variability for overall supra-permafrost water (lowest standard deviation shown in Table 3) compared to precipitation and stream water. This finding suggests that mixing processes within the active layer attenuate variations in the $\delta^{18}O$ signal from precipitation.

The seasonal dynamics of the soil temperature and moisture of the active layer are the most important drivers of hydrological processes in the permafrost regions (Sugimoto et al., 2003; Woo and Xia, 1996; Wang et al., 2015). Rising air temperature promotes permafrost thawing and thickening of the active layer, thereby increasing soil moisture, since more water sources with different isotopic compositions, such as atmospheric precipitation and meltwater from subsurface ice, input or release into the permafrost active layer (Sugimoto et al., 2003; Throckmorton et al., 2016; Song et al., 2017). Hence, we conducted the correlation analysis between the isotope compositions in supra-permafrost and stream water, as well as soil temperature and moisture (Fig. 5).

The isotopic compositions in supra-permafrost water were significantly associated with soil temperature (Fig. 5c: $R^2 = 0.18$, $P < 0.000$). Moreover, we identified a variable relationship between the supra-permafrost water isotopes and soil temperature (first increasing and then decreasing). When soil temperature was $<4\,°C$, the active layer was relatively thinner, and less water was stored in the permafrost active layer, which is more susceptible to evaporation, causing isotope enrichment. With the further increase in soil temperature ($>4\,°C$), the active layer becomes thicker, and the soil moisture also increases, yielding the supra-permafrost water recharged by greater amounts of precipitation that depleted heavy isotopes (precipitation depleted heavy isotopes in August). The positive correlation between supra-permafrost water with precipitation isotopes (Fig. 5f: $R^2 = 0.39$, $P < 0.000$) also confirms this finding. These findings reveal the differences in the water movement mechanisms at different stages of permafrost freezing and thawing processes.

The isotopic compositions in stream water are negatively correlated with soil temperature (Fig. 5a: $R^2 = 0.08$, $P < 0.000$) and soil moisture (Fig. 5b: $R^2 = 0.31$, $P < 0.000$). They are also strongly positively correlated with the isotopic compositions in precipitation (Fig. 5e: $R^2 = 0.44$, $P < 0.000$). These findings indicate that the stream water was controlled by precipitation and the freeze–thaw cycle of the permafrost active layer. Notably, our correlation results are in line with the previous studies in the Zuomaokong water-

**Table 3.** Mean, range, and standard deviation of $\delta^{18}O$ (‰) and $\delta D$ (‰) for precipitation, stream water, and supra-permafrost water.

| Sample type | Mean | | Minimum | | Maximum | | Standard deviation | |
|---|---|---|---|---|---|---|---|---|
| | $\delta^{18}O$ | $\delta D$ | $\delta^{18}O$ | $\delta D$ | $\delta^{18}O$ | $\delta D$ | $\delta^{18}O$ | $\delta D$ |
| Precipitation | −13.2 | −92.6 | −30.9 | −244.2 | 2.9 | 34.0 | 7.1 | 58.0 |
| Supra-permafrost water | −14.0 | −101.5 | −17.5 | −127.6 | −4.6 | 50.9 | 1.7 | 10.8 |
| Stream water | −13.5 | −97.4 | −19.6 | −146.3 | −4.0 | −39.8 | 2.4 | 16.3 |

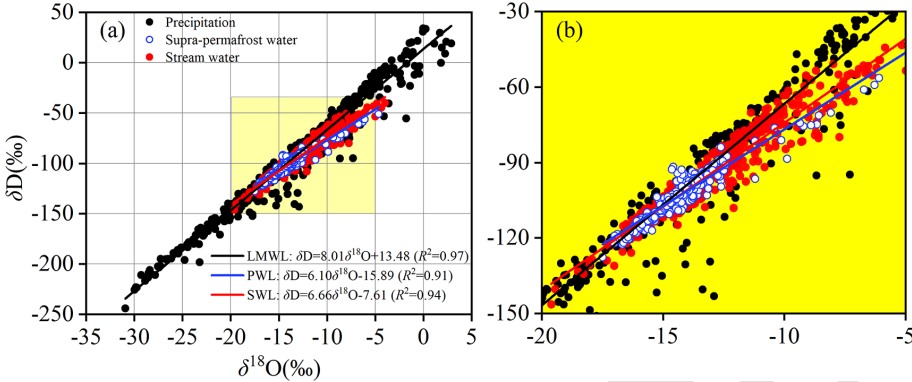

**Figure 3.** Relationships between $\delta^{18}O$ and $\delta D$ of precipitation, stream water, and supra-permafrost water. **(a)** All isotope data; **(b)** the data, constrained by a yellow square.

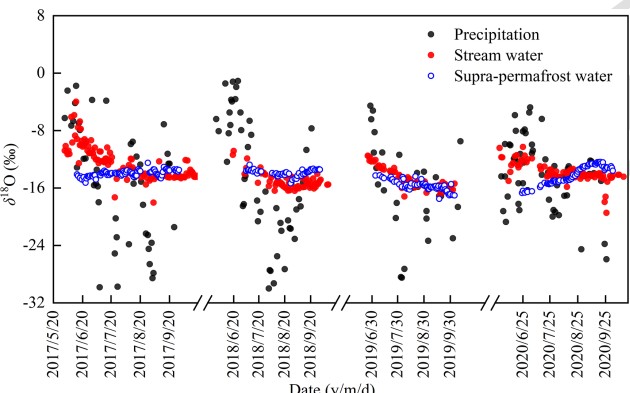

**Figure 4.** Comparison of variations in precipitation $\delta^{18}O$ (‰) and stream and supra-permafrost water $\delta^{18}O$ (‰) during the warm seasons from 2017 to 2020.

shed of central TP (Song et al., 2017). At the same time, a very strong correlation was observed between stream water and supra-permafrost water isotopes (Fig. 5d: $R^2 = 0.23$, $P < 0.000$). These significant positive correlations between precipitation and permafrost isotopes and stream water isotopes indicate that precipitation and supra-permafrost water are important recharge sources of stream water.

### 3.3 Hydrograph separations of stream water

The EMMA model has been used to identify the mixing processes and quantify the contribution of each endmember. The monthly mixing diagram using the mean $\delta^{18}O$ and $\delta D$ showed that the isotope values in stream water are very close to the other endmember (precipitation or supra-permafrost water), indicating that the component of stream water was dominated by different sources at different stages (Fig. 6). However, in some mixing diagrams, stream water was located outside the range composed of the two endmembers (atmospheric precipitation and supra-permafrost water) possibly due to the influence of vegetation transpiration and soil evaporation on precipitation and ground ice that were mixed and stored in the active layer (Li et al., 2020b). Nevertheless, the isotopic composition in stream water is very close to one of the endmembers (Fig. 6). Overall, supra-permafrost water and precipitation can be treated as the two endmembers in hydrograph separations of stream water. For quantitative evaluation of the results above, the source proportion of stream water was quantitatively determined using the isotopic data and the IHS model (Eqs. 1–4). The results indicate that precipitation and supra-permafrost water contributed $35 \pm 2\%$ and $65 \pm 2\%$ of the total discharge of stream water, respectively. Seasonal patterns showed that the precipitation contribution decreased from June to August and then increased in September; for the supra-permafrost water, the contribution to streamflow increased from June to August and then slightly decreased in September (Fig. 7).

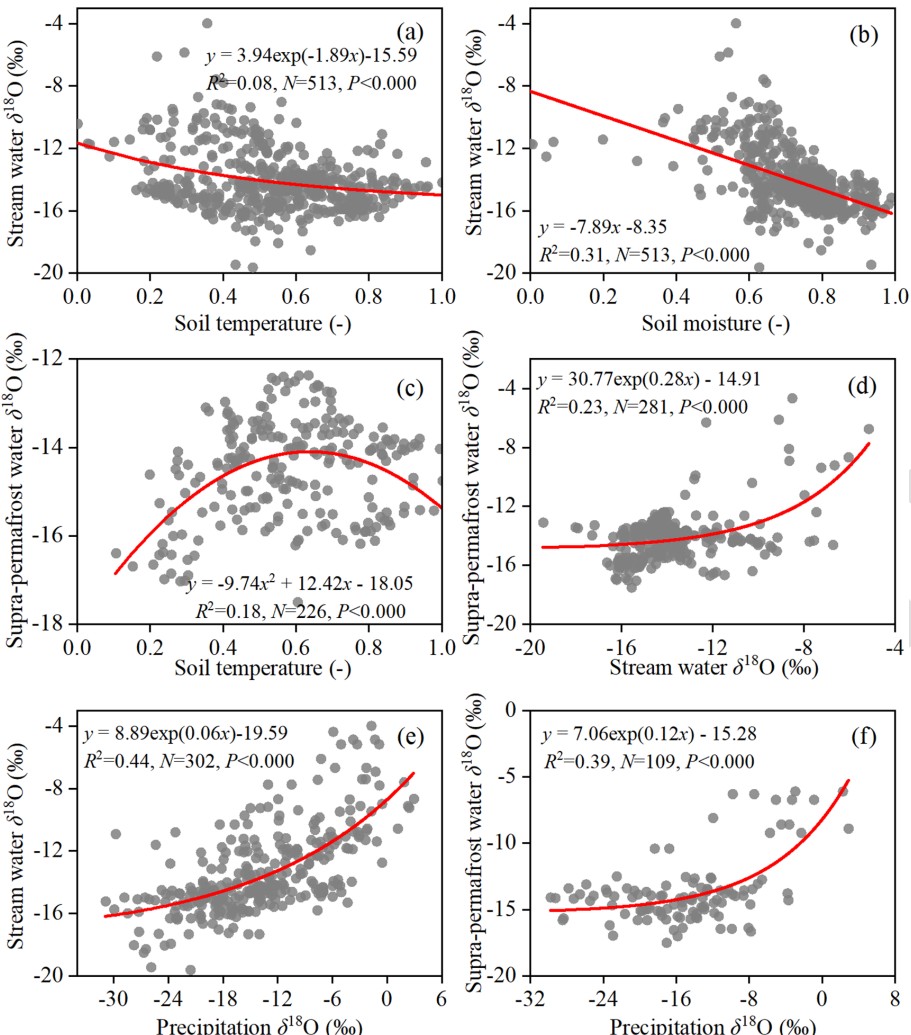

**Figure 5.** Correlation analysis between stable isotopes and influencing factors. Panels **(a)** and **(b)** are correlations between stream water $\delta^{18}O$ with soil temperature and moisture, respectively; **(c)** and **(d)** correlations between supra-permafrost water $\delta^{18}O$ with soil temperature and stream water $\delta^{18}O$, respectively; **(e)** and **(f)** correlations between precipitation $\delta^{18}O$ with stream and supra-permafrost water $\delta^{18}O$, respectively. The red line is the fitted line. Soil moisture and temperature data used in correlation analysis were normalized. "−" indicates dimensionless. The soil moisture and temperature data were the daily average value of all the sensor data in the soil layer.

During the initial thawing stage of permafrost (June), precipitation recharge was the primary source of stream water, approximately accounting for 78 % of the total discharge of streamflow. Notably, in June 2018 and 2019, we observed no supra-permafrost water in the sampling well; therefore, the stream water during these periods is almost solely attributed to precipitation. However, the stream water is primarily derived from the active layer water in the thawing and end of thawing stages of permafrost (July–September), approximately accounting for 79 % of the streamflow. Especially in August, the contribution of supra-permafrost water to stream water can reach 98 %. These findings suggested that supra-permafrost water was the dominant source of the stream water during the warm season in the study area.

### 3.4   Estimation of mean residence and its association with environmental variables

The modelled amplitudes can reflect the observed patterns of variability in water isotope compositions due to great mixing; meanwhile, seasonal variations in the output waters are far more significant with larger amplitude, thereby indicating greater responsiveness to recent precipitation inputs (Rodgers et al., 2005a). Moreover, we found that different water isotopes showed obvious seasonal variation, and precipitation was an important input of stream and supra-permafrost water. Thus, we fit seasonal sine-wave curves to the annual $\delta^{18}O$ variations in precipitation, stream, and supra-permafrost water. The sine-wave regression parameters for $\delta^{18}O$ in precipitation, supra-permafrost water, and

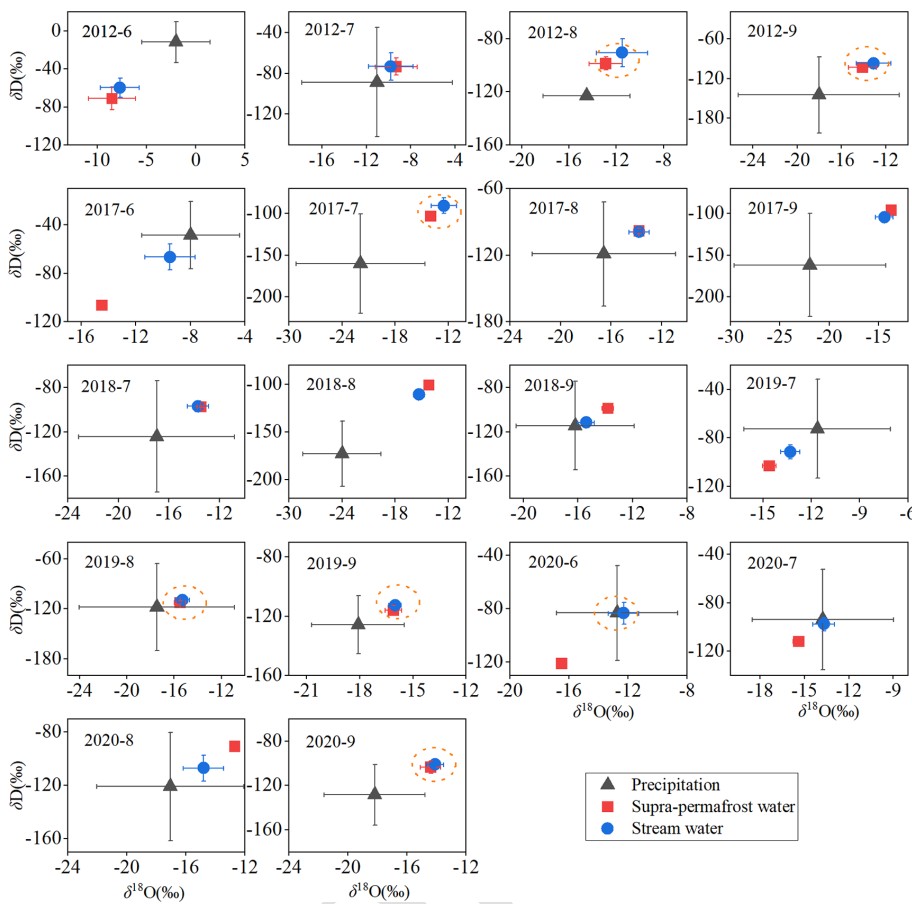

**Figure 6.** Monthly mixing diagram using the mean $\delta^{18}O$ and $\delta D$ values for stream water. The dotted orange ellipse indicates that the isotopic value in stream water is very close to the other endmember.

stream water are shown in Figs. 8–10. We found that the modelled $\delta^{18}O$ fits well to the observed isotope values, with RMSE of 4.35‰–6.58‰ for precipitation, 0.37‰–1.95‰ for supra-permafrost water, and 0.72‰–1.99‰ for stream water. Meanwhile, the results of the periodic regression analysis of isotope compositions in different waters were all statistically robust ($P < 0.01$). The $\delta^{18}O$ seasonal variations in supra-permafrost and stream water exhibited weak amplitudes compared with the precipitation, which is a consequence of the mixing processes and longer residence time. The mean amplitude of stream water (1.54‰) in our study reasonably agrees with the previous results in the Zuomaokong watershed of the hinterland of the TP (1.76‰), which applied the same method to calculate the mean amplitude of stream water $\delta^{18}O$ in five permafrost catchments (Song et al., 2017).

Then, the sine-wave regression parameters were translated into the estimates of water MRT using Eq. (9). The estimated results of MRT are shown in Fig. 11. The calculated MRT for the stream water ranged from 42 to 270 d, with a mean value of 100 d and a standard deviation of 68 d. Meanwhile, for supra-permafrost water, the MRT varied from 23

to 596 d, with a mean value of 255 d and a standard deviation of 229 d. The estimated MRT revealed a large annual variability, regardless of stream water or supra-permafrost water. Generally, multiple factors affect the water storage in the active layer, including precipitation and soil temperature in permafrost regions (Wright et al., 2008). The climate differences and variability may have substantially affected the MRT estimates (Tetzlaff et al., 2007). In this study, the correlations between estimated MRT and soil parameters (active layer thickness and soil temperature), climate factors (air temperature and precipitation), and vegetation index (NDVI) were quantified (Table 4). The uncertainty of the estimated MRT may influence these correlations (Hu et al., 2020); hence we considered the uncertainty of the MRT in the regression analysis. The results showed that after considering the uncertainty of MRT, the $R^2$ of the regression analysis for active layer thickness was improved, with less obvious differences for the other factors. In the following analysis, we use the regression analysis results after considering the uncertainty.

The correlation analysis showed that the MRT of supra-permafrost water exhibits a strong positive correlation with air temperature ($R^2 = 0.67$, $P < 0.05$) and soil temperature

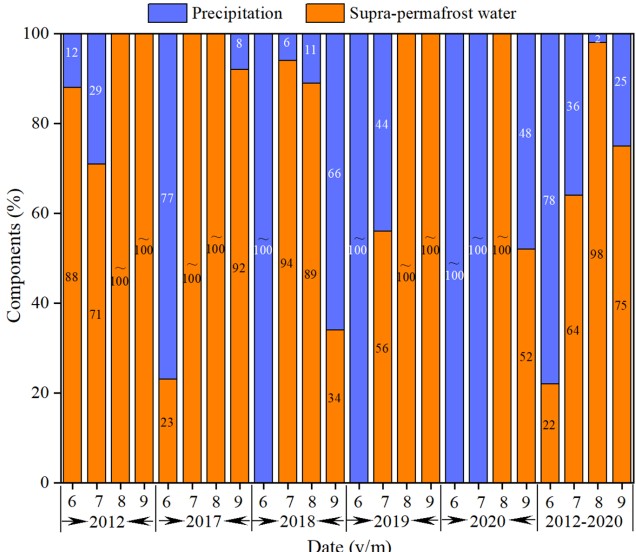

**Figure 7.** Monthly variation in components in stream water. "∼" indicates approximate estimation of component contributions to stream water.

($R^2 = 0.78$, $P < 0.05$) (Table 4). Meanwhile, the relatively stronger positive correlations were identified between stream water MRT and soil temperature ($R^2 = 0.79$, $P < 0.001$). These strong correlations may relate to the thickening of the active layer due to the increase in air and soil temperature because we also observed that MRTs of supra-permafrost and stream water were negatively correlated with the active layer thickness ($R^2 = 0.59$, $P < 0.01$ and $R^2 = 0.44$, $P < 0.05$, respectively). Meanwhile, the longest estimated MRTs (270 d for stream water and 596 d for supra-permafrost water) were observed in 2018 with the relatively lower precipitation amount (319 mm). Thus, we analysed the correlation between MRT and precipitation and found that supra-permafrost and stream water MRTs are both negatively correlated with precipitation ($R^2 = 0.64$, $P < 0.01$ and $R^2 = 0.32$, $P < 0.01$, respectively) (Table 4). Interestingly, we found that the stream and supra-permafrost water MRTs are both negatively correlated with NDVI ($R^2 = 0.26$, $P < 0.05$ and $R^2 = 0.65$, $P < 0.01$, respectively) (Table 4). In other words, an increase in vegetation coverage (high NDVI values) might lead to a shorter MRT in our catchment.

## 4 Discussion

### 4.1 Contribution of supra-permafrost water to stream water

Quantifying the components of stream water can provide insights into the hydrological effects of permafrost degradation (Li et al., 2020a). In this study, differences in the seasonal contributions of runoff components to stream water

were observed. The contribution of supra-permafrost water during the thawing and end of thawing stages of permafrost to stream water was higher than that of the initial thawing stage. In June, the active layers of permafrost were gradually thawing as temperature increased, yet at this time, the active layer remained relatively thin, while the precipitation increased, resulting in most precipitation directly converging in the river. Under higher temperature and precipitation conditions in July and August, the strong thawing of permafrost occurred. The thickening of the permafrost active layer functions as a water reservoir, thereby allowing for more precipitation recharge into the active layer. Previous studies found that summer rain was the predominant source for water within the active layer in permafrost catchments (Throckmorton et al., 2016; Li et al., 2020b; Zhu et al., 2019), which is attributed to the relatively high permeability of the active layer (Li et al., 2020a). Then the active layer water produces a direct recharge to stream water with a contribution rate of 81 % on average at this stage. As temperatures drop, the surface soil layer gradually begins to freeze, and the bottom of the active layer approaches freezing in late September, thereby gradually decreasing the volume of supra-permafrost water due to the freezing processes of the aquifer. This phenomenon slightly decreased the contribution of supra-permafrost water to stream water in September (∼ 75 %). More succinctly, seasonal variations in the freezing and thawing of permafrost directly trigger the runoff process (Li et al., 2020a).

In this study, approximately two-thirds of the stream water was attributed to supra-permafrost water. Previous studies have also reported that precipitation and thawing permafrost water contributes 55.2 % and 44.8 %, respectively, to the thermokarst lakes in the Beiluhe basin of the interior TP (4600 m a.s.l.) (Yang et al., 2016). A recent study has reported a greater contribution of supra-permafrost water (49 %) compared with precipitation (34 %) in the whole source region of the Yangtze River (SRYR) (Li et al., 2020a). The contribution rate of the supra-permafrost water to stream water in our catchment was relatively high compared with the whole SRYR. This finding is potentially related to the replenishment from other water sources, since for the whole SRYR, in addition to the precipitation and supra-permafrost water, there is a large amount of replenishment of glacier meltwater, which, to a certain extent, reduces the contribution of supra-permafrost water to stream water. In subarctic permafrost catchments, pre-event water was also a primary contributor of stream water (∼ 90 %) (Carey and Quinton, 2005). These studies demonstrated the significant contribution from cryosphere meltwater to water resources in high-altitude permafrost catchments. Considering that permafrost is widely distributed in the central TP and plays an important role in surface–groundwater exchange within the catchment, permafrost degradation will significantly influence the hydrological processes in alpine permafrost regions in the context of climate warming.

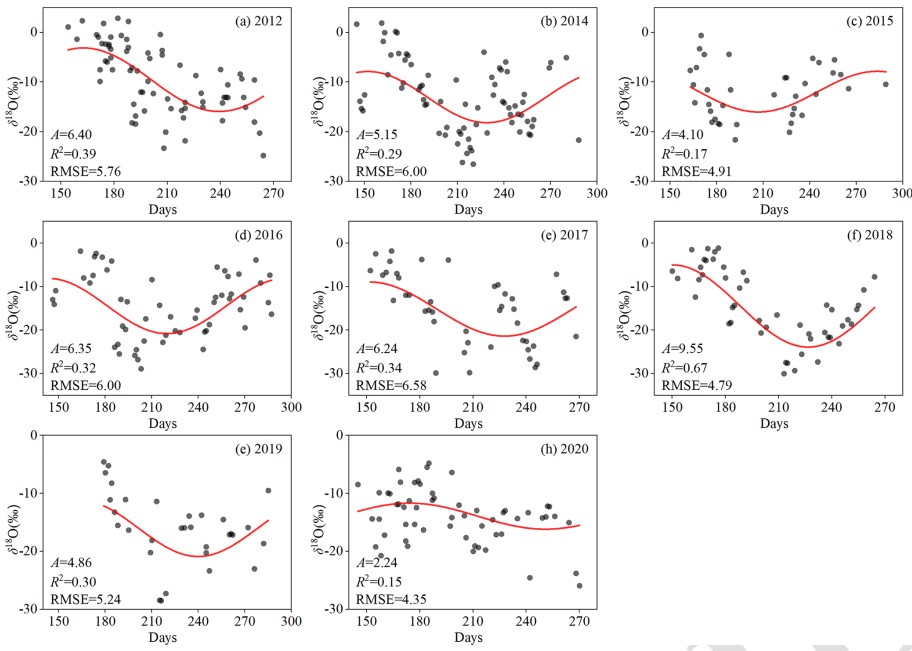

**Figure 8.** Fitted sine-wave regression models to $\delta^{18}$O in precipitation in Xiaoliuyu catchment during 2012 and 2014–2020. $A$ is the amplitude, and RMSE is root mean square error for the modelled isotopic signature.

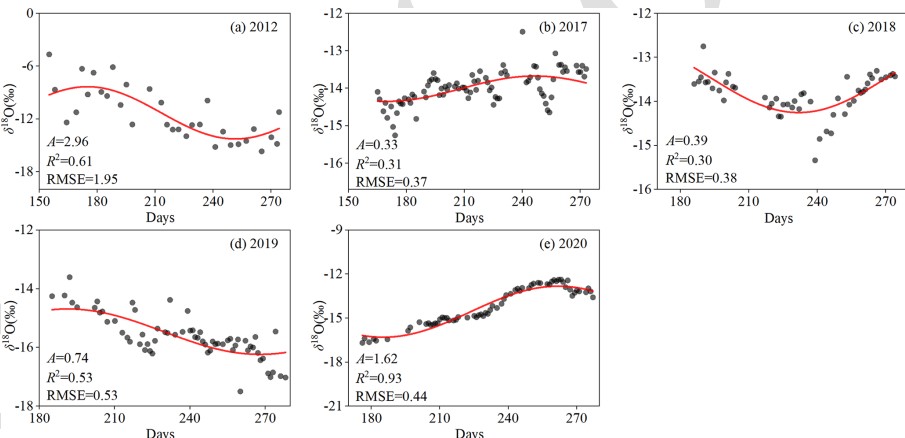

**Figure 9.** Fitted sine-wave regression models to $\delta^{18}$O in supra-permafrost water in Xiaoliuyu catchment during 2012 and 2017–2020. $A$ is the amplitude, and RMSE is root mean square error for the modelled isotopic signature.

## 4.2 Potential driving mechanism for MRT variability

In this study, the estimated MRT of supra-permafrost water was distinctly longer than that of stream water, which reflects the more complex water movement and recharge processes for supra-permafrost water. On the one hand, this is because supra-permafrost water stored in active layers is replenished by more old water compared with surface runoff. On the other hand, it is related to the longer flow path for supra-permafrost water since the active layer increases the length of the water flow path (Frampton and Destouni, 2015; Ma et al., 2019b). In turn, a short MRT of the stream water indicates

a relatively rapid response of surface water to precipitation. Overall, the estimated MRTs of stream and supra-permafrost water in our catchment were shorter compared to those estimated in most previous studies from non-permafrost catchments (Table 5). This may be related to whether there is groundwater recharge or not. Permafrost acts as an aquiclude while being usually characterized by rapid hydrograph responses (Tetzlaff et al., 2018). Moreover, it cuts off the interaction and mixing between deep groundwater and surface water and supra-permafrost water. Rodgers et al. (2005b) and Soulsby et al. (2006) have reported that catchment water MRT was correlated with the percentage that groundwater

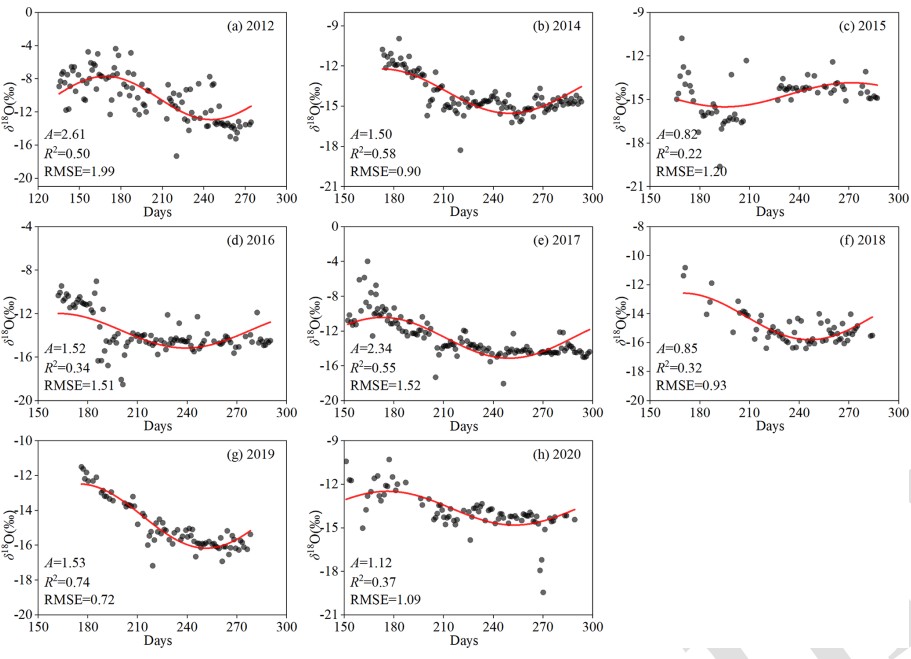

**Figure 10.** Fitted sine-wave regression models to $\delta^{18}$O in stream water in Xiaoliuyu catchment during 2012 and 2014–2020. $A$ is the amplitude, and RMSE is root mean square error for the modelled isotopic signature.

**Table 4.** Relationships between active layer thickness, soil temperature, air temperature, precipitation, NDVI, and MRT. $x$ indicates the factor as the independent variable.

| | | Regression based on mean MRT (days) | | | Regression based on MRT uncertainty (days) | | |
|---|---|---|---|---|---|---|---|
| | Factor | Regression equation | $R^2$ | Sig | Regression equation | $R^2$ | Sig |
| Supra-permafrost water | ALT | $y = 4.23\exp(0.24x) + 48.62$ | 0.11 | $P>0.05$ | $y = 1.43\exp(0.63x) + 150.68$ | 0.44 ↑ | **P<0.05** |
| | ST | $y = 929x + 1299$ | 0.87 ↑ | **P<0.05** | $y = 956x + 1337$ | 0.78 ↑ | **P<0.05** |
| | AT | $y = 752x + 4307$ | 0.69 ↑ | **P<0.05** | $y = 767x + 4283$ | 0.67 ↑ | **P<0.05** |
| | $P$ | $y = -1.89x + 1169$ | 0.58 | $P = 0.08$ | $y = 3479\exp(-0.005x) - 125$ | 0.64 ↓ | **P<0.01** |
| | NDVI | $y = 0.103x^{-4.405}$ | 0.51 | $P = 0.07$ | $y = 0.143x^{-4.286}$ | 0.65 ↓ | **P<0.01** |
| Stream water | ALT | $y = 4.67\exp(1.11x) + 71.56$ | 0.81 ↑ | **P<0.01** | $y = 3.02\exp(0.37x) + 59.23$ | 0.59 ↑ | **P<0.01** |
| | ST | $y = (1.7x + 1.35)^{-0.26}$ | 0.81 ↑ | **P<0.001** | $y = 3.79E + 14\exp(35.47x) + 76$ | 0.79 ↑ | **P<0.001** |
| | AT | $y = 51x + 372$ | 0.05 | $P>0.05$ | $y = 42x + 325$ | 0.01 | $P>0.05$ |
| | $P$ | $y = 4.94E + 5(x^{-1.77})$ | 0.47 ↓ | **P<0.01** | $y = 1.67E + 10(1.7x)^{-3.25} + 52.6$ | 0.32 ↓ | **P<0.01** |
| | NDVI | $y = 1.422x^{-2.394}$ | 0.20 ↓ | **P<0.05** | $y = 1.50x^{-2.384}$ | 0.26 ↓ | **P<0.05** |

Note: ALT = active layer thickness, ST = soil temperature (°C), AT = air temperature (°C), $P$ = precipitation (mm), NDVI = normalized differential vegetation index; Sig indicates statistical significance; ↑ and ↓ indicate significant trend of increase and decrease, respectively; bold font indicates that it passed significance test of 0.05.

contributes to stream water in non-permafrost catchments. Furthermore, previous studies have revealed longer MRTs for deep groundwater (Table 5). For instance, deep groundwater MRTs in the mountainous Brugga basin reached more than 5 years (Soulsby et al., 2000). Thus, the mixing or interaction between stream water and groundwater increases the stream water MRT in the non-permafrost catchments. However, this interaction is suspended by the permafrost layer in our catchment, causing the shorter MRT. This implies that the hydrological processes in high-altitude permafrost regions are unique compared with non-permafrost regions.

As the buffer layer between the permafrost and atmosphere, the active layer is vulnerable to climate change (Xu

and Wu, 2021). The increase in air temperature can alter the temperature of shallow permafrost due to strong land–atmosphere interactions. This, in turn, increases the thickness of the permafrost active layer, thereby allowing soil water to move into the deeper soil layer. In this study, the significant positive correlation between MRT and active layer thickness supports previous findings showing that the MRT of permafrost catchments is highly dependent on the depth of the active layer due to the warming effects (Frampton and Destouni, 2015). From the mechanism perspective, the deepening of the active layer can increase the length of the water flow pathway and reduce transport velocities due to a shift in flow direction from horizontal saturated ground-

**Table 5.** Statistics of MRT-related research results.

| Site | Altitude (m) | Water type | Model type | Tracer | Data length (years) | MRT | References |
|---|---|---|---|---|---|---|---|
| Our study site[a] | 5100–5435 | Stream water | Exponential | $\delta^{18}O$ | 8 | 100 d | This study |
| Huanjiang[b] | 272–627 | Stream water | Exponential | $\delta^{18}O$ | 2 | 300 d | F. Wang et al. (2020) |
| Mandava[b] | 383 | Stream water | Exponential | $\delta^{18}O$ | 2 | 444 d | Sanda et al. (2017) |
| Upper Váh[b] | 1500 | Stream water | Exponential | $\delta^{18}O$ | 4 | 390–570 d | Dosa et al. (2011) |
| Dee[b] | 1000 | Stream water | Exponential | $\delta^{18}O$ | 3 | 601 d | Soulsby et al. (2010) |
| Minjiang upper[b] | 300–7100 | Stream water | Exponential | $\delta^{18}O$ | 1 | 698 d | Xia et al. (2021) |
| Our study site[a] | 5100–5435 | Groundwater | Exponential | $\delta^{18}O$ | 5 | 255 d | This study |
| Himalaya[a] | 1600–5200 | Groundwater | Exponential | $\delta D$ | 1 | 4.5 months | Shah et al. (2017) |
| Vermigliana[b] | 1221 | Groundwater | Exponential | $\delta^{18}O$ | 1 | 1.3 years | Chiogna et al. (2014) |
| Allt a' Mharcaidh[b] | 300–1111 | Groundwater | Exponential | $\delta^{18}O$ | 4 | >5 years | Soulsby et al. (2000) |
| Huanjiang[b] | 272–627 | Groundwater | Exponential | $\delta^{18}O$ | 2 | 161–1407 d | F. Wang et al. (2020) |

Note: [a] indicates the catchment covered by permafrost; [b] indicates the catchment not covered by permafrost. The symbol "–" indicates no data were available in the references.

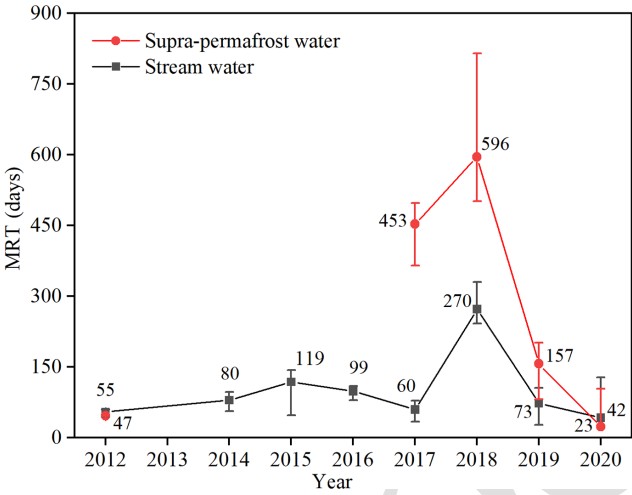

**Figure 11.** Inter-annual variation in stream water MRT during 2012 and 2014–2020 and supra-permafrost water MRT during 2012 and 2017–2020.

water flow to vertical flow infiltrating into the deeper subsurface, thereby increasing water MRT in permafrost catchments (Frampton and Destouni, 2015). Moreover, a previous study has reported that the potential thaw of permafrost layers due to climate change could increase MRT at the catchment scale by 20 %–45 % (Lyon et al., 2010). A similar study in the non-frozen regions reported that MRT of forestland and shrubland water both increased with soil depth (Ma et al., 2019b).

Precipitation is an important part of the water cycle and is the main input for catchment water sources. However, precipitation in the permafrost region of TP significantly increased in recent decades (Zhao et al., 2019). In this study, significant positive correlation between precipitation and MRT was observed, indicating that the increased precipitation or wetter climatic conditions may accelerate the water cycle process in permafrost regions. In central TP, the increase in precipitation thinned the permafrost active layer by decreasing soil

heat flux, thereby cooling the soil and alleviating permafrost degradation (Zhou et al., 2021b; Luo et al., 2020). This phenomenon subsequently triggered more water to rapidly flow into the river channel in the form of surface runoff, thereby reducing the MRT of catchment water in the end. However, a previous study suggested that a much wetter climate than average probably causes higher MRT in the low-altitude temperate regions (Soulsby et al., 2006). Our study does not resonate with these results, possibly because the aquiclude effect of permafrost reduces surface infiltration and enhances surface runoff generation (Woo, 1990; Hinzman et al., 2005; Gao et al., 2021).

The soil layer is the main source of nutrients and water required for vegetation growth (Xu et al., 2019). Water shortage in soil layer may occur when the water MRT is too short, but a longer MRT can hamper the water infiltration consequently to root anoxia, thus affecting the plant growth (Ma et al., 2019b). In this study, stream and supra-permafrost water MRTs both showed significant and negative correlations with NDVI, indicating that vegetation coverage may influence water residence time in permafrost catchments. Previous studies have noted the vegetation cover was one of the most important factors that governs the hydrological processes and thermal cycles in permafrost catchments (Wang et al., 2012a). The decline of vegetation coverage elevated soil temperature and moisture, which in turn accelerated the permafrost thawing and thickened the active layer (Wang et al., 2012b). Moreover, water MRT variability can be influenced by the development of the active layer; therefore, we believe that the effect of vegetation on MRT is also driven by the variations in the thickness of the permafrost active layer.

In this study, comparisons of the controlling factors of MRT between stream and supra-permafrost water indicated that supra-permafrost water is more sensitive to environmental change compared with stream water. Previous findings have also suggested that the supra-permafrost water in the seasonally thawed layer is sensitive to climate (Cheng and Jin, 2013) and is, therefore, significantly impacted by precip-

itation, temperature, and vegetation. From a mechanistic perspective, climate and vegetation factors affected the MRT of stream and supra-permafrost water by modifying the active layer thickness of permafrost. Considering that MRT is a fundamental descriptor of hydrological functions within catchments (Shah et al., 2017; McGuire and McDonnell, 2006), therefore, changes in hydrological processes in permafrost watersheds can be investigated by assessing water MRT in the context of climate and environmental change.

### 4.3 Uncertainty and limitations

In this study, long-term stable isotopic data of stream and supra-permafrost water were used to estimate water MRT and determine the mechanism of MRT variability in a high-altitude permafrost catchment of the TP. Nonetheless, some uncertainty remains in the results of MRT estimation, including model assumptions, spatial variability in isotope input and output, and isotopic fractionation.

Different transit time distribution (TTD) models are applicable to different watershed conditions (Małoszewski and Zuber, 1998), which may affect the assessment of residence time. The exponential model, a commonly used model for MRT estimation, describes the catchment with flow times that are exponentially distributed (McGuire and McDonnell, 2006), which assumed that the system is in steady-state conditions and operates as a perfect mixer (Sánchez-Murillo et al., 2015; Chiogna et al., 2014). This perfect mixer indicates that the mixing between input and baseflow is rapid and complete, whereas an ideal mixing cannot exist in an aquifer, which is an important uncertainty source of the applied model (Małoszewski et al., 1983; Fenicia et al., 2010). Nevertheless, an exponential model is suitable for MRT estimation in unconfined aquifers with shallow sampling points (Małoszewski and Zuber, 1998; Małoszewski et al., 1983; Stewart and McDonnell, 1991). In effect, the exponential TTD model could also approximate TTD in some non-steady cases (Haitjema, 1995; Rodhe et al., 1996). In this study area, the underlying surface was relatively uniform with less landscape heterogeneity and characterized by rapid hydrological processes. Moreover, the active layer of permafrost belonged to an unconfined aquifer and functioned as a water reservoir, thereby allowing for more precipitation recharge into the active layer to mix with old water. The amplitudes of output isotopes (stream and supra-permafrost water) were much lower than those of input (precipitation) and the dominant contribution of supra-permafrost water to stream water, both of which indicated that the precipitation was well mixed with other water within the catchment. Thus, the exponential model is suitable for application in permafrost catchments to some extent.

In general, measurement inputs represent spatial and temporal inputs for the entire catchment (McGuire and McDonnell, 2006). At the catchment scale, elevation, air temperature, and rainfall intensity may cause considerable variation in the isotopic composition of precipitation, particularly in mountainous areas (Ingraham, 1998). Thus, inputs of a tracer to the catchment system are highly variable in space and time, which is an important source of uncertainty in the interpretation of catchment response (McGuire and McDonnell, 2006; Hrachowitz et al., 2009). A previous study suggested that precipitation at high altitudes is characterized by high isotopic amplitudes (Jasechko et al., 2016), which may result in underestimation of MRT in our study area due to one sampling site for precipitation. In practice, the isotopic composition of precipitation is often sampled at one site (McGuire and McDonnell, 2006). Considering the catchment area of our study was relatively small ($2.7\,\text{km}^2$) with an altitude drop of 300 m, the size of the selected catchment in this study was much smaller than that of most catchments previously reported (McGuire and McDonnell, 2006). Therefore, the effects of elevation on meteorological data and precipitation isotopic variability are minor, and one precipitation sampling location could represent the whole catchment to some extent. Additionally, this study only collected supra-permafrost water from one sampling point due to economic and logistical constraints in the alpine regions, which is a limitation in estimating MRT. Given that the supra-permafrost water is primarily derived from precipitation, the spatial variability in isotopes in supra-permafrost water may also be minor in such a small catchment. Even so, the spatial variability in isotopes in supra-permafrost water may result in underestimation of MRT in the study area.

The fractionation effects attributed to evaporation may potentially increase the uncertainty of water age estimation due to its impact on isotopic compositions and signals (Richardson and Kimberley, 2010; McDonnell et al., 2010; Song et al., 2017). Hence, the fractionation effects during the transformation from actual precipitation to effective input must be considered (McDonnell et al., 2010; Rusjan et al., 2019). In the study area, the atmospheric precipitation is primarily solid; the solid precipitation will be melted rapidly over a short period following deposition to form liquid water that enters soil and river channels, and therefore it is difficult for snowpack to exist within this catchment. Thus, we did not collect snowpack or snow meltwater as an input signal for MRT estimation. Nonetheless, solid precipitation may be subjected to evaporative fractionation to some degree when melted to be surface and subsurface runoff, thereby increasing the uncertainty of MRT estimation. Considering the rapid transformation of snow into infiltrated water and low air temperature, the potential effect of evaporation on the isotopic composition in precipitation, and consequently on MRT estimates, is relatively limited, which was not considered in the MRT estimation in this catchment.

To further analyse the uncertainty of MRT derived from the seasonal variability in isotope composition in the hydrological component, we used the amplitude coefficient of input and output to estimate the uncertainty of MRT and found it larger for water with long residence time. Regression anal-

ysis showed that after considering the uncertainty of MRT, the $R^2$ of the regression analysis for active layer thickness was improved, with less obvious differences for the other factors. This suggests that the uncertainty of estimated MRT may affect the sensitivity of MRT to specific factors (Hu et al., 2020), indicating that the uncertainty of estimated MRT should be considered when discussing MRT influencing factors. Therefore, future research should consider the uncertainty of MRT and improve the accurate assessment of MRT in alpine catchments. In addition, the permafrost data used in the regression analysis are regional average data, which may increase the uncertainty of regression analysis (Table 4). Overall, although there remain uncertainty and limitations for MRT estimation in our study, isotope-based MRT estimation is valuable for identifying changes in hydrological processes of the permafrost regions, where there is a lack of observational data. Thus, it is necessary to utilize more measurements in different sub-catchments to augment the data representativeness in future research.

## 5 Conclusions

In this study, long-term observational stable isotopic data were used to estimate runoff components and water MRT in a high-altitude permafrost catchment of the TP. We found that the isotope composition in precipitation, stream, and supra-permafrost water exhibited obvious seasonal variability. The freeze–thaw cycles of permafrost active layer and direct input of precipitation significantly modified the stable isotope compositions in supra-permafrost and stream water. The two-component IHS model indicated that the supra-permafrost water was the dominant contribution to the total discharge of stream water. We estimated that the MRT ranged from 42 to 270 d (mean 100 d) and 23 to 596 d (mean 255 d) for stream water and supra-permafrost water, respectively. Such shorter MRTs of supra-permafrost and stream water (compared to the non-permafrost catchments) might reflect the unique characteristics of the hydrological process in permafrost catchments. Furthermore, the analysis of influencing factors revealed that the MRT of supra-permafrost water was more sensitive to environmental change than stream water. From the perspective of influencing mechanisms, climate and vegetation factors affected the water MRT in permafrost catchments which were mainly driven by changing the thickness of the permafrost active layer. Under the influence of global warming, the permafrost degeneration and active layer deepening may slow down the rate of the water cycle in permafrost regions. The findings of this study expand our understanding of the hydrological processes in high-altitude permafrost catchments under climate warming.

*Data availability.* We have uploaded the long-term isotope data and estimated MRT results in figshare with the URL of https://doi.org/10.6084/m9.figshare.21500229.v1 (Wang, 2022).

*Author contributions.* XHe and SK provided initial ideas of this work. SW and XHe performed the water isotope data collection and processing and result analysis. XHe provided the meteorological data of the study area. HF and XHo helped in analysing the results and revised the manuscript. All authors contributed to the writing.

*Competing interests.* The contact author has declared that none of the authors has any competing interests.

*Acknowledgements.* The authors thank Wang Ming, the observer of the Tanggula Cryosphere and Environment Observation Station (TaCOS), Chinese Academy of Sciences, for assisting with field data collection.

*Financial support.* This research has been supported by the joint research project of Three-River Headwaters National Park, Chinese Academy of Sciences, and the People's Government of Qinghai Province (grant no. LHZX-2020-11), the Second Tibetan Plateau Scientific Exploration (grant no. 2019QZKK0605), IWHR Research & Development Support Program (grant no. HY110145B0012021), National Public Research Institutes for Basic R&D Operating Expenses Special Project (grant no. CKSF2021485), and the project of State Key Laboratory of Cryospheric Science (grant no. SKLCS-ZZ-2022).

*Review statement.* This paper was edited by Ylva Sjöberg and reviewed by Genxu Wang and two anonymous referees.

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

**Remarks from the typesetter**

TS1  Please give an explanation of why this needs to be changed. We have to ask the handling editor for approval. Thanks.