# Peer review of "Estimation of streamwater components and residence time in a permafrost catchment in the Central Tibetan Plateau using long-term water stable isotopic data"

_The Cryosphere, 2022_

## Author Comment (AC1)

**Response to Anonymous Referee #1**

We are very grateful for your valuable and instructive comments and suggestions. The review comments are listed below and marked in blue, followed by the detailed responses marked in black. The sentences added in the revised manuscript were marked in red and italic.

Kind regards,

Xiaobo He

(on behalf of the co-authors)

This study uses stable water isotopes to look at the mean residence time (MRT) for a catchment in the Tibetan Plateau. The novelty here is the long-term nature of the data series being leveraged for the MRT estimate as these types of sampling campaigns are challenging to coordinate in cold and alpine regions. The study is well written and well structured making it easy to read. Still, I do struggle some with the uniqueness of the study presented as while these data types of are challenge to collect and not often presented in the literature, there is a question of what we learn here for this catchment that advances beyond previous regional efforts like in Song et al. (2017)? I think bringing forward the improved process understanding in face of the possible uncertainty is needed here to move this manuscript beyond a presentation of the uniqueness of place that leverages data alone.

Response: Thanks for your comment. Song et al (2017) previously focused on studying the young water fraction of river water in the Zuomaokong watershed of the hinterland of the Tibetan plateau (TP) and its controlling factors (topography and vegetation), but not considered the effects of permafrost changes, especially active layer changes, on permafrost hydrological processes. Comparatively, we investigated not only the streamwater MRT, but also the groundwater (supra-permafrost water) in a long time series (5-8 years). Moreover, we analysed the impact of permafrost changes on MRT to explore permafrost hydrological processes. We believe these are aspects go beyond previous research by Song et al (2017).

The MRT estimation in our study does have some uncertainty, thus, we recalculated the uncertainty of MRT and analyzed the possible reasons for this uncertainty, including model assumptions, spatial variability of isotope input and output, and isotopic fractionation. We added relevant discussion on MRT uncertainty in the subsequent reply and revised manuscript (line 440-498).

One aspect that needs attention is the intercomparison of MRTs between various catchments and studies presented in the manuscript. I appreciate the effort and thinking to place this one catchment in a broader context; however, the different methods and models used when estimating MRT can have significant impacts on the resolution MRT and the entire travel time distribution. Caution is needed when comparing absolute MRT with other catchments. I think if the authors want to keep these comparisons, more information needs to be added (like a column or two in Table 4) indicating the model type and technique used to estimate MRT. Further, a richer discussion of the impacts of the modeling assumptions should be provided as they pertain to this region. There has been significant research and

literature on these topics over the last decades and it seems some of the more modern interpretations are missing from this study. All in all, I would anticipate a more thoughtful consideration of the assumptions behind the convolution approach you are implementing here.

Response: Thank you for the suggestion. Indeed, MRT estimation based on different models may impact the intercomparison between various catchments and studies. Thus, we removed MRT studies that were not calculated by exponential model and added more information about the model type and technique in Table 5 (line 396-399).

***Table 5.** Statistics of MRT-related research results.*

| Site | Altitude (m) | Water type | Model type | Tracer | Data length (years) | MRT | References |
|---|---|---|---|---|---|---|---|
| **Our study site**[a] | 5100–5435 | Stream water | Exponential | $\delta^{18}O$ | 8 | 100 days | This study |
| Huanjiang t[b] | 272–627 | Stream water | Exponential | $\delta^{18}O$ | 2 | 300 days | (Wang et al., 2020) |
| Mandava[b] | 383 | Stream water | Exponential | $\delta^{18}O$ | 2 | 444 days | (Sanda et al., 2017) |
| Upper Váh[b] | 1500 | Stream water | Exponential | $\delta^{18}O$ | 4 | 390–570 days | (M. et al., 2011) |
| Dee[b] | 1000 | Stream water | Exponential | $\delta^{18}O$ | 3 | 601 days | (Soulsby et al., 2010) |
| Minjiang upper[b] | 300–7100 | Stream water | Exponential | $\delta^{18}O$ | 1 | 698 days | (Xia et al., 2021) |
| **Our study site**[a] | 5100–5435 | Groundwater | Exponential | $\delta^{18}O$ | 5 | 255 days | This study |
| Himalaya[a] | 1600–5200 | Groundwater | Exponential | $\delta D$ | 1 | 4.5 months | (Shah et al., 2017) |
| Vermigliana[b] | 1221 | Groundwater | Exponential | $\delta^{18}O$ | 1 | 1.3 years | (Chiogna et al., 2014) |
| Allt[b]Mharcaidh[b] | 300–1111 | Groundwater | Exponential | $\delta^{18}O$ | 4 | >5 years | (Soulsby et al., 2000) |
| Huanjiang[b] | 272–627 | Groundwater | Exponential | $\delta^{18}O$ | 2 | 161–1407 days | (Wang et al., 2020) |

*Note: "a" indicates the catchment covered by permafrost; "b" indicates the catchment not covered by permafrost. The symbol "—" indicates no data were available in the references.*

Meanwhile, the assumptions of exponential model have been added in "Materials and methods" (line 170-172) and "Uncertainty and limitations" sections (line 441-460):

*In this study, long-term stable isotopic data of stream and supra-permafrost water were used to estimate water MRT and determine the mechanism underlying MRT variability in a high-altitude permafrost catchment of the TP. Nonetheless, some uncertainty remains in the results of MRT estimation, including model assumptions, spatial variability of isotope input and output and isotopic fractionation.*

*Different transit time distribution (TTD) models are applicable to different watershed conditions (Małoszewski and Zuber, 1998), which may affect the assessment of residence time. The exponential model, a commonly used model for MRT estimation, describes the catchment with flow times that are exponentially distributed (Mcguire and Mcdonnell, 2006), which assumed that the system is in steady-state conditions and operates as a perfect mixer (Sánchez-Murillo et al., 2015; Smith, 1984; Chiogna et al., 2014). This perfect mixer indicates that the mixing between input and baseflow is rapid and complete, whereas an ideal mixing cannot exist in an aquifer, which is an important uncertainty source of the applied model (Małoszewski et al., 1983; Fenicia et al., 2010). Nevertheless, exponential model is suitable for MRT estimation in unconfined aquifers with shallow sampling points (Małoszewski and Zuber, 1998; Małoszewski et al., 1983; Stewart and Mcdonnell, 1991). In effect, the exponential TTD*

*model could also approximate TTD in some non-steady cases (Haitjema, 1995; Rodhe et al., 1996). In this study area, the underlying surface was relatively uniform with less landscape heterogeneity and characterized by rapid hydrological processes. Moreover, the active layer of permafrost belonged to an unconfined aquifer and functioned as a water reservoir, thereby allowing for more precipitation recharge into the active layer to mix with old water. The amplitudes of output isotopes (stream and supra-permafrost water) were much lower than those of input (precipitation) and the dominant contribution of supra-permafrost water to stream water, both of which indicated that the precipitation was well mixed with other water within the catchment. Thus, the exponential model is suitable for application in permafrost catchment to some extent.*

In addition, if there is a connection between the MRT and the unique processes in permafrost environment, it would be more insightful to describe them explicitly. Modeling literature (e.g. Frampton and Destouni, 2015) exists on the subject and would help reduce the ambiguity connecting water movement and process as they are considered in this study. Further, and connected with this comment, there is need to separate the result and discussion section in to two separate sections. Given the amount of data being presented and the analysis put forward, plenty of material for results. Also, mixing the two sections together as is currently done creates confusion about what your data show and how you are interpreting it relative to the science. And it would be good in a separate section of the discussion to consider more the potential limitations of the current study as they pertain to assumptions, data representativeness and the models being considered.

Response: Thank you for the suggestion. To be clear, we have described accordingly and explicitly the connection between the MRT and hydrological processes of permafrost in the revised manuscript as follows:

*In this study, the estimated MRT of supra-permafrost water was distinctly longer than that of stream water, which reflects the more complex water movement and recharge processes for supra-permafrost water. On the one hand, this is because supra-permafrost water stored in active layers is replenished by more old water compared with surface runoff. On the other hand, it is related to the longer flow path for supra-permafrost water since the active layer increases the length of water flow path (Frampton and Destouni, 2015; Ma et al., 2019).* (line 380-384)

*In this study, the significant positive correlation between MRT and active layer thickness support previous findings showing that the MRT of permafrost catchments is highly dependent on the depth of the active layer due to the warming effects (Frampton and Destouni, 2015). From the mechanism perspective, the deepening of the active layer can increase the length of water flow pathway and reduce transport velocities due to a shift in flow direction from horizontal saturated groundwater flow to vertical flow infiltrate into deeper subsurface, thereby increasing water MRT in permafrost catchment (Frampton and Destouni, 2015).* (line 404-408)

We separated the "Results and discussion" into two sections, and added a section "Uncertainty and limitations" to the "Discussion" section, where we discussed the uncertainty of estimated MRT, including model assumptions, spatial variability of isotope input and output, and isotopic fractionation. The following statements have been added in the revised manuscript (line 440-498):

*4.3 Uncertainty and limitations*

*In this study, long-term stable isotopic data of stream and supra-permafrost water were used to estimate water MRT and determine the mechanism underlying MRT variability in a high-altitude permafrost catchment of the TP. Nonetheless, some uncertainty remains in the results of MRT estimation, including model assumptions, spatial variability of isotope input and output and isotopic fractionation.*

*Different transit time distribution (TTD) models are applicable to different watershed conditions (Małoszewski and Zuber, 1998), which may affect the assessment of residence time. The exponential model, a commonly used model for MRT estimation, describes the catchment with flow times that are exponentially distributed (Mcguire and Mcdonnell, 2006), which assumed that the system is in steady-state conditions and operates as a perfect mixer (Sánchez-Murillo et al., 2015; Smith, 1984; Chiogna et al., 2014). This perfect mixer indicates that the mixing between input and baseflow is rapid and complete, whereas an ideal mixing cannot exist in an aquifer, which is an important uncertainty source of the applied model (Małoszewski et al., 1983; Fenicia et al., 2010). Nevertheless, exponential model is suitable for MRT estimation in unconfined aquifers with shallow sampling points (Małoszewski and Zuber, 1998; Małoszewski et al., 1983; Stewart and Mcdonnell, 1991). In effect, the exponential TTD model could also approximate TTD in some non-steady cases (Haitjema, 1995; Rodhe et al., 1996). In this study area, the underlying surface was relatively uniform with less landscape heterogeneity and characterized by rapid hydrological processes. Moreover, the active layer of permafrost belonged to an unconfined aquifer and functioned as a water reservoir, thereby allowing for more precipitation recharge into the active layer to mix with old water. The amplitudes of output isotopes (stream and supra-permafrost water) were much lower than those of input (precipitation) and the dominant contribution of supra-permafrost water to stream water, both of which indicated that the precipitation was well mixed with other water within the catchment. Thus, the exponential model is suitable for application in permafrost catchment to some extent.*

*In general, measurement inputs represent spatial and temporal inputs for the entire catchment (Mcguire and Mcdonnell, 2006). At the catchment scale, elevation, air temperature, and rainfall intensity may cause considerable variation in isotopic composition of precipitation, particularly in mountainous areas (Ingraham, 1998). Thus, inputs of tracer to the catchment system are highly variable in space and time and important sources of uncertainty in interpretation of catchment response (Mcguire and Mcdonnell, 2006; Hrachowitz et al., 2009). A previous study suggested that precipitation at high altitudes is characterized by high isotopic amplitudes (Jasechko et al., 2016), which may result in underestimation of MRT in the study area due to one sampling site for precipitation. In practice, the isotopic composition of precipitation is often sampled at one site (Mcguire and Mcdonnell, 2006).*

*Considering the catchment area of our study was relatively small (2.7 km$^2$) with an altitude drop of 300 m. The size of the selected catchment in this study was much smaller than that of most catchments previously reported (Mcguire and Mcdonnell, 2006). Therefore, the effects of elevation on meteorological data and precipitation isotopic variability are minor and one precipitation sampling location could represent the whole catchment to some extent. Additionally, this study only collected supra-permafrost water from one sampling point due to economic and logistical constraints in the alpine regions, which is a limitation in estimating MRT. Given that the supra-permafrost water is primarily derived from precipitation, the spatial variability of isotopes in supra-permafrost water may also be minor in such small catchment. Even so, the spatial variability of isotopes in supra-permafrost water may result in underestimation of MRT in the study area.*

*The fractionation effects attributed to evaporation may potentially increase the uncertainty of water age estimation due to its impact on isotopic compositions and signals (Richardson and Kimberley, 2010; Mcdonnell et al., 2010; Song et al., 2017). Hence, the fractionation effects during the transformation from actual precipitation to effective input must be considered (Mcdonnell et al., 2010; Rusjan et al., 2019). In the study area, the atmospheric precipitation is primarily solid; the solid precipitation will be melted rapidly over a short period following deposition to form liquid water that enters soil and river channels, therefore it is difficult for snowpack to exist within this catchment. Thus, we did not collect snowpack or snow melt water as an input signal for MRT estimation. Nonetheless, solid precipitation may be subjected to evaporative fractionation to some degree when melted to be surface and subsurface runoff, thereby increasing the uncertainty of MRT estimation. Considering the rapid transformation of snow into infiltrated water and low air temperature, the potential effect of evaporation on the isotopic composition in precipitation, and consequently on MTT estimates, is relatively limited, which was not considered in the MRT estimation in this catchment.*

*To further analyse the uncertainty of MRT derived from the seasonal variability of isotope composition in hydrological component, we used the amplitude coefficient of input and output to estimate the uncertainty of MRT and found it larger for water with long residence time. Regression analysis showed that after considering the uncertainty of MRT, an improved R$^2$ for the thickness of the active layer with fewer differences for other factors. This suggests that uncertainty of estimated MRT may affects the sensitivity of MRT to specific factors (Hu et al., 2020), indicating that the uncertainty of estimated MRT should be considered when discussing MRT influencing factors. Therefore, future research should consider the uncertainty of MRT and improve the accurate assessment of MRT in alpine catchments.*

*Overall, although there remain uncertainty and limitations for MRT estimation in our study, isotope-based MRT estimation is valuable for identifying changes in hydrological processes of the permafrost regions, where there is a lack of observational data. Thus, it is necessary to utilize more measurements in different sub-catchments to augment the data representativeness in future research.*

Given the complexity of sampling precipitation in cold regions, more information is needed to help the reader understand how you were sampling here. For example, were how was snow treated throughout the sampling? Were snowpacks or snow melt water collected and considered as inputs in any sense? Also, looking at the variation in elevation in the region, how representative of the catchment is the one meteorological station and precipitation sampling location? Rainfall isotopic composition is rather variable with elevation and snowpack and snow melt rates are really variable. How is the isotopic input variability considered within this study? It seems ignored based on the methodology presented.

Response: Thank you for the suggestions. We added more information about precipitation sampling in the revised manuscript as follows (line 118-120):

*Liquid precipitation samples were collected immediately following every precipitation event using bulk collector to minimize the effects of evaporation. Solid precipitation (snow) samples were collected into a plastic bag and taken to a warm place to be thawed, following which water samples were transferred into 50-mL PE bottles.*

The elevation, air temperature, and rainfall intensity may cause considerable variation of isotopic composition in precipitation, particularly in mountainous areas (Ingraham, 1998), which may result in underestimation of MRT in the study area due to one sampling site for precipitation. Considering the size of the selected catchment in our study is small, we believed that the effects of elevation on meteorological data and precipitation isotopic variability are minor and one precipitation sampling location could represent the whole catchment to some extent. The following statements have been added to the manuscript (line 462-472):

*At the catchment scale, elevation, air temperature, and rainfall intensity may cause considerable variation in isotopic composition of precipitation, particularly in mountainous areas (Ingraham, 1998). Thus, inputs of tracer to the catchment system are highly variable in space and time and important sources of uncertainty in interpretation of catchment response (Mcguire and Mcdonnell, 2006; Hrachowitz et al., 2009). A previous study suggested that precipitation at high altitudes is characterized by high isotopic amplitudes (Jasechko et al., 2016), which may result in underestimation of MRT in the study area due to one sampling site for precipitation. In practice, the isotopic composition of precipitation is often sampled at one site (Mcguire and Mcdonnell, 2006). Considering the catchment area of our study was relatively small (2.7 km$^2$) with an altitude drop of 300 m. The size of the selected catchment in this study was much smaller than that of most catchments previously reported (Mcguire and Mcdonnell, 2006). Therefore, the effects of elevation on meteorological data and precipitation isotopic variability are minor and one precipitation sampling location could represent the whole catchment to some extent.*

In the study area, the atmospheric precipitation is primarily solid; the solid precipitation will be melted rapidly over a short period following deposition to form liquid water that enters soil and river channels, therefore it is difficult for snowpack to exist within this catchment. Thus, we did not collect snowpack or snow melt water as an input signal for MRT estimation (see Fig 1e). Thus, we did not

collect snowpack or snow melt water as an input signal for MRT estimation. The following statements have been added to the manuscript (line 480-487):

*In the study area, the atmospheric precipitation is primarily solid; the solid precipitation will be melted rapidly over a short period following deposition to form liquid water that enters soil and river channels, therefore it is difficult for snowpack to exist within this catchment. Thus, we did not collect snowpack or snow melt water as an input signal for MRT estimation. Nonetheless, solid precipitation may be subjected to evaporative fractionation to some degree when melted to be surface and subsurface runoff, thereby increasing the uncertainty of MRT estimation. Considering the rapid transformation of snow into infiltrated water and low air temperature, the potential effect of evaporation on the isotopic composition in precipitation, and consequently on MTT estimates, is relatively limited, which was not considered in the MRT estimation in this catchment.*

[Figure]

Figure 1e Photograph of the underlying surface in the catchment and meteorological station taken in June 2018. We added this picture to the study area map in the revised manuscript.

The input variability and source water variability of only having one location for monitoring supra-permafrost water sampling seems as if it could confound the results and interpretation to some extent. Specifically, if there are large frozen regions upstream of the stream sampling location, these would have significant impacts on the ability of precipitation to transfer to the stream over the entire catchment. Variability of isotopic compositions in springs and sub-watersheds is well documented (e.g. Lyon et al. 2018). The spatial variability at play in the catchment must be either accounted for or the potential impacts at least taking into consideration via discussion within this study.

Response: Thank you for the suggestion. This study only collected supra-permafrost water from one sampling point due to economic and logistical constraints in alpine regions, which is a limitation for estimating MRT in our study. Considering the small size of the selected catchment in this study and the supra-permafrost water mainly derived from the contribution of precipitation, we believed that the

spatial variability of isotopes in supra-permafrost water also may be minor in such a small catchment. Even so, the spatial variability of isotopes in supra-permafrost water may also result in underestimation of MRT in the study area. The following statement have been added to the manuscript (line 469-476):

*The size of the selected catchment in this study was much smaller than that of most catchments previously reported (Mcguire and Mcdonnell, 2006). Therefore, the effects of elevation on meteorological data and precipitation isotopic variability are minor and one precipitation sampling location could represent the whole catchment to some extent. Additionally, this study only collected supra-permafrost water from one sampling point due to economic and logistical constraints in the alpine regions, which is a limitation in estimating MRT. Given that the supra-permafrost water is primarily derived from precipitation, the spatial variability of isotopes in supra-permafrost water may also be minor in such small catchment. Even so, the spatial variability of isotopes in supra-permafrost water may result in underestimation of MRT in the study area.*

Finally, some consideration of uncertainty should be presented. There are several fitted relationships that are being compared across the research. In and of themselves, these are wrought with uncertainty and confidence intervals that can impact the significance of the findings. I would want to see some assessment of the robustness of the results relative to the uncertainty or lack of representativeness of the data being presented. At the least, the two-component hydrograph can directly incorporate the uncertainty via the approach put forward by Genereux (1998). Without characterization of the uncertainty, I am left wondering how much of the results is driven by under-represented variability in the sampling at a catchment scale, the simplifying assumptions within the model, and the fitted equations that smooth out all the between event variability and extremes. That last point is rather important given potential flashy nature of these systems during certain times of the year and more dampened responses as the systems thaw seasonally.

Response: Thank you for the suggestions. The uncertainty of MRT results has been recalculated by the method described in Morales and Oswald (2020). We considered the uncertainty of the estimated MRT in the regression analysis. The results showed that after considering the uncertainty of MRT, the $R^2$ of the regression analysis for the thickness of permafrost active layer has been improved, while less difference for other factors (see Table 4, line 346-350). The following statement have been added to the manuscript (line 488-494):

*To further analyse the uncertainty of MRT derived from the seasonal variability of isotope composition in hydrological component, we used the amplitude coefficient of input and output to estimate the uncertainty of MRT and found it larger for water with long residence time. Regression analysis showed that after considering the uncertainty of MRT, an improved $R^2$ for the thickness of the active layer with fewer differences for other factors. This suggests that uncertainty of estimated MRT may affects the sensitivity of MRT to specific factors (Hu et al., 2020), indicating that the uncertainty of estimated MRT should be considered when discussing MRT influencing factors. Therefore, future*

*research should consider the uncertainty of MRT and improve the accurate assessment of MRT in alpine catchments.*

*Table 4. Relationships between active layer thickness, soil temperature, air temperature, precipitation, NDVI, and MRT. x indicates the factor as the independent variable.* (line 342)

| | Factor | Regression based on mean MTT (days) | | | Regression based on MTT uncertainty (days) | | |
|---|---|---|---|---|---|---|---|
| | | Regression equation | $R^2$ | Sig | Regression equation | $R^2$ | Sig |
| Supra-permafrost water | ALT | $y=4.23exp(0.24x)+48.62$ | 0.11 | P>0.05 | $y=1.43exp(0.63x)+150.68$ | 0.44 ↑ | **P<0.05** |
| | ST | $y=929x+1299$ | 0.87 ↑ | **P<0.05** | $y=956x+1337$ | 0.78 ↑ | **P<0.05** |
| | AT | $y=752x+4307$ | 0.69 ↑ | **P<0.05** | $y=767x+4283$ | 0.67 ↑ | **P<0.05** |
| | P | $y=-1.89x+1169$ | 0.58↓ | **P<0.01** | $y=3479exp(-0.005x)-125$ | 0.64↓ | **P<0.01** |
| | NDVI | $y=0.103x^{-4.405}$ | 0.51 | P=0.07 | $y=0.143x^{-4.286}$ | 0.65↓ | **P<0.05** |
| Stream water | ALT | $y=4.67exp(1.11x)+71.56$ | 0.81 ↑ | **P<0.01** | $y=3.02exp(0.37x)+59.23$ | 0.59 ↑ | **P<0.01** |
| | ST | $y=(1.7x+1.35)^{-0.26}$ | 0.81 ↑ | **P<0.001** | $y=3.79E+14exp(35.47x)+76$ | 0.79 ↑ | **P<0.001** |
| | AT | $y=51x+372$ | 0.05 | P>0.05 | $y=42x+325$ | 0.01 | P>0.05 |
| | P | $y=4.94E+5(x^{-1.77})$ | 0.47↓ | **P<0.01** | $y=1.67E+10(1.7x)^{-3.25}+52.6$ | 0.32↓ | **P<0.01** |
| | NDVI | $y=1.422x^{-2.394}$ | 0.20↓ | **P<0.05** | $y=1.50x^{-2.384}$ | 0.26↓ | **P<0.05** |

*Note: ALT = active layer thickness, ST = Soil temperature (℃), AT = air temperature (℃), P = Precipitation (mm), NDVI= normalized differential vegetation index; Sig indicates statistical significance; ↑ and ↓ indicates significant trend of increase and decrease, respectively; Bold font indicates that it passed significance test of 0.05.*

**Minor Comments**

L100: This sentence is random and does not make sense here. Further, not sure what you mean with efficiently?

Response: Thank you for the suggestion. This sentence has been deleted in the revised manuscript.

L171: This first sentence is odd. Separate the results and discussions to increase presentation clarity.

Response: Thank you for the suggestion. We have removed this sentence, spilt the "results and discussion" into two parts, and added a section "Uncertainty and limitations" to the "Discussion" section, which discussed the uncertainty of estimated MRT, including model assumptions, spatial variability of isotope input and output, and isotopic fractionation (line 440-498).

**References**

[revised manuscript text omitted]

---

## Author Comment (AC2)

**Response to Anonymous Referee #2**

We are very grateful for your valuable and instructive comments and suggestions. The review comments are listed below and marked in blue, followed by the detailed responses marked in black. The sentences added in the revised manuscript were marked in red and italic.

Kind regards,

Xiaobo He

(on behalf of the co-authors)

The paper by Wang and co-authors entitled "*Estimation of water residence time in a permafrost catchment in the Central Tibetan Plateau using long-term water stable isotopic data*" leverages a unique data set in a remote environment to shed light on the 'mean residence time' (MRT) of groundwater and stream water using transit time approaches. They seek to highlight how the active layer and permafrost influence MRT and draw inferences on permafrost hydrology.

The paper is well written and clear. The figures are straight-forward and interpretable. I have a number of editorial comments at the end, yet I have considerable concerns about the analysis that I believe must be addressed before this manuscript is suitable for publication. The data is novel and is of considerable value to the hydrological and permafrost community, yet there are large uncertainties and at times I believe mis-application of methods that the authors need to consider before this manuscript is suitable for publication.

Response: Thanks a lot for the comments and suggestions. We analyzed the uncertainty of runoff separation and MRT estimation, and modified the inaccurate application of correlation analysis methods in the revised manuscript. The detailed statements have been added to the subsequent reply and revised manuscript.

**General Comments:**

The authors use the exponential model as opposed to the more widely used gamma distribution. I am curious as to why this is. This will have considerable impact on the MRT calculation and should be discussed.

Response: Thank you for the comments. The reasons that we use the exponential model to assess MRT are as follows:

(1) The exponential model describes a catchment with flow times that are exponentially distributed, including pathways with very short transit times (Mcguire and Mcdonnell, 2006), which assumed that the system is in steady-state conditions and operates as a perfect mixer (Chiogna et al., 2014; Sánchez-Murillo et al., 2015a; Smith, 1984). This perfect mixer means that the mixing between input and baseflow is rapid and complete, whereas an ideal mixing cannot exist in an aquifer. Therefore, exponential model is suitable for MRT estimation in unconfined aquifers with shallow sampling points (Małoszewski et al., 1983; Stewart and Mcdonnell, 1991; Xia et al., 2021). In effect, the exponential

TTD model could also approximate TTDs in some non-steady cases (Haitjema, 1995; Rodhe et al., 1996). In our study, underlying surface is relative uniform with less landscape heterogeneity and characterized by rapid hydrological processes. Moreover, the active layer of permafrost belonged to an unconfined aquifer and functioned as a water reservoir, allowing more precipitation recharges into the active layer to mix with old water. Thus, the applicability of exponential model can be satisfied basically to some extent;

(2) The exponential model is essentially a special case of the gamma model with the shape factor parameter $\alpha$ equal to 1 (Mcguire and Mcdonnell, 2006). If catchments are modelled as well-mixed reservoirs, their travel times follow an exponential distribution (Kirchner et al., 2000). In our study, the amplitudes of output isotopes (stream and supra-permafrost water) were much lower than that of the input (precipitation), in effect, indicating that the precipitation was well mixed with other water within the catchment;

(3) We calculated the MRT of stream water using partial isotope data (2014-2020) based on exponential and gamma models, respectively, and found that these two results were very consistent ($R^2$=0.97, $P$<0.001). Given that exponential model is the most widely used model for MRT assessment over the past decades (Mcguire and Mcdonnell, 2006; Seeger and Weiler, 2014; Rusjan et al., 2019) and the calculation process of MRT using exponential model is often more convenient compared with the gamma model.

Aboveall, exponential model was selected to assess water residence time in our study.

Meanwhile, the following statements have been added to the revised manuscript as follows (line 446-460):

*The exponential model, a commonly used model for MRT estimation, describes the catchment with flow times that are exponentially distributed (Mcguire and Mcdonnell, 2006), which assumed that the system is in steady-state conditions and operates as a perfect mixer (Sánchez-Murillo et al., 2015b; Smith, 1984; Chiogna et al., 2014). This perfect mixer indicates that the mixing between input and baseflow is rapid and complete, whereas an ideal mixing cannot exist in an aquifer, which is an important uncertainty source of the applied model (Małoszewski et al., 1983; Fenicia et al., 2010). Nevertheless, exponential model is suitable for MRT estimation in unconfined aquifers with shallow sampling points (Małoszewski and Zuber, 1998; Małoszewski et al., 1983; Stewart and Mcdonnell, 1991). In effect, the exponential TTD model could also approximate TTD in some non-steady cases (Haitjema, 1995; Rodhe et al., 1996). In this study area, the underlying surface was relatively uniform with less landscape heterogeneity and characterized by rapid hydrological processes. Moreover, the active layer of permafrost belonged to an unconfined aquifer and functioned as a water reservoir, thereby allowing for more precipitation recharge into the active layer to mix with old water. The amplitudes of output isotopes (stream and supra-permafrost water) were much lower than those of input (precipitation) and the dominant contribution of supra-permafrost water to stream water, both of which*

*indicated that the precipitation was well mixed with other water within the catchment. Thus, the exponential model is suitable for application in permafrost catchment to some extent.*

The correlation analysis is highly flawed and must be revisited. It appears the authors use any type of correlation against variables with different units, etc., to 'look around for relationships'. This is not statistically robust in any way. Values must be transposed/normalized to compare among factors, and the type of correlation must be explained. Figure 6 shows two 'best fit' lines with either low or no relationships. This is a regression analysis. Furthermore, this data is ALL serially correlated which needs to be accounted for. As it stands, the authors have a lot of work to do to justify this analytical approach. Binning data together from across seasons, etc., truly make this confounding as the active layer changes.

Response: Thank you for the suggestions. Part of the statistical results of the correlation matrix in the first manuscript did not make sense. Thus, the correlation matrix was adjusted to the correlation analysis chart by bring together all the data from across seasons (Fig 5, line 255). Additionally, we added the type of correlation to these correlation analysis chart. The soil moisture and temperature data were normalized in the correlation analysis.

[Figure]

***Figure 5***. *Correlation analysis between stable isotopes and influencing factors. (a) and (b) correlations between stream water $\delta^{18}O$ with: soil temperature and moisture, respectively; (c) and (d) correlations between supra-permafrost water $\delta^{18}O$ with: soil temperature and stream water $\delta^{18}O$, respectively; (e) and (f) correlations between precipitation $\delta^{18}O$ with: stream and supra-permafrost water $\delta^{18}O$, respectively; Red line is the fitted line. Soil moisture and temperature data used in correlation analysis were normalized; "-" presents dimensionless.*

More information on the IHS method is needed.

Response: Thank you for the suggestions. In the first manuscript, we applied IHS to the entire period to get component proportions of stream water, but lacks the analysis of the mechanism of runoff process. Thus, we have assessed stream water components on a monthly scale and added relevant information on the IHS method in the revised manuscript. We added more information on the IHS method in the "Materials and methods" section in the manuscript as follows (line 148-158):

*In this study, the precipitation-weighted average of precipitation isotopes was used to assess stream water components on a monthly scale to determine the mechanism of runoff process in permafrost*

*catchments.*

*The uncertainty in hydrograph separations generally included two aspects, one is the analysis error of tracer concentrations, while the other is the spatial and temporal variations of the tracer of components (Uhlenbrook and Hoeg, 2003), calculated using the Gaussian error propagation technique (Genereux, 1998):*

$$w_y = \sqrt{\left(\frac{\partial y}{\partial x_1}w_{x1}\right)^2 + \left(\frac{\partial y}{\partial x_2}w_{x2}\right)^2 + \ldots + \left(\frac{\partial y}{\partial x_n}w_{xn}\right)^2} \tag{5}$$

*where w represents the uncertainty in the variable specified in the subscript and y is the contribution of a specific streamflow component x to streamwater.*

More appropriate literature is needed, along with less bold statements about the importance/influence of this work.

Response: Thank you for the suggestions. We added the following appropriate literatures in the revised manuscript accordingly:

1. Woo, M.K., 1990. Consequences of climatic change for hydrology in permafrost zones. Journal of Cold Regions Engineering, 4(1):15-20. (line 29, 421)

2. Hinzman, L.D., Kane, D.L., Woo, M. K., 2005. Permafrost Hydrology, Encyclopedia of Hydrological Sciences. (line 29, 421)

3. Woo, M.K., Kane, D.L., Carey, S.K., Yang, D.,: Progress in permafrost hydrology in the new millennium. Permafrost and Periglacial Processes, 19(2): 237-254, 2008. (line 37)

4. Woo, M. K. and Xia, Z.: Effects of Hydrology on the Thermal Conditions of the Active Layer, Hydrology Research, 27, 129-142, https://doi.org/10.2166/nh.1996.0024, 1996. (line 231)

5. Sugimoto, A., Naito, D., Yanagisawa, N., Ichiyanagi, K., Kurita, N., Kubota, J., Kotake, T., Ohata, T., Maximov, T. C., and Fedorov, A. N.: Characteristics of soil moisture in permafrost observed in East Siberian taiga with stable isotopes of water, Hydrological Processes, 17, 1073-1092, https://doi.org/10.1002/hyp.1180, 2003. (line 197, 234)

6. Throckmorton, H. M., Newman, B. D., Heikoop, J. M., Perkins, G. B., Feng, X., Graham, D. E., O'Malley, D., Vesselinov, V. V., Young, J., Wullschleger, S. D., and Wilson, C. J.: Active layer hydrology in an arctic tundra ecosystem: quantifying water sources and cycling using water stable isotopes, Hydrological Processes, 30, 4972-4986, https://doi.org/10.1002/hyp.10883, 2016. (line 203)

7.Woo, M. K. and Xia, Z.: Effects of Hydrology on the Thermal Conditions of the Active Layer: Paper presented at the 10th Northern Res. Basin Symposium. Hydrology Research, 27(1-2): 129-142, 1996. (line 231)

8. Sugimoto, A, Naito, et al.: Characteristics of soil moisture in permafrost observed in East Siberian taiga with stable isotopes of water. Hydrological Prochydrological Processes, 2003. (line 234)

Additionally, we adjusted and revised some sentences about importance and influence of this work accordingly:

*The findings from our study will expand our understanding of the hydrological process in permafrost regions under global warming.* (line 72, line 513-514).

*Therefore, changes in hydrological processes in permafrost watersheds can be investigated by assessing water MRT in the context of climate and environmental change* (line 438-439).

Many of the conclusions are not supported by the data.

Response: Thank you for the suggestions. We have revised and explained the conclusions that are not supported by the data based on the line comments. Specific revisions are detailed in line comment response and the revised manuscript accordingly.

We explained and rephrased this sentence "Line 329 These strong correlations indicate that soil and air temperature are potentially efficient predictors of supra-permafrost water MRT":

*The increase of air temperature can alter the temperature of shallow permafrost due to strong land-atmosphere interactions. This, in turn, increases the thickness of permafrost active layer, thereby allowing soil water to move into the deeper soil layer. In this study, the significant positive correlation between MRT and active layer thickness support previous findings showing that the MRT of permafrost catchments is highly dependent on the depth of the active layer due to the warming effects (Frampton and Destouni, 2015). From the mechanism perspective, the deepening of the active layer can increase the length of water flow pathway and reduce transport velocities due to a shift in flow direction from horizontal saturated groundwater flow to vertical flow infiltrate into deeper subsurface, thereby increasing water MRT in permafrost catchment (Frampton and Destouni, 2015)* (line 401-409).

We deleted these sentences "Line 364: However, it remains unclear whether a positive feedback mechanism exists between vegetation and permafrost active layer changes or not" and "Line 365: Moreover, the optimum residence time for vegetation growth should be elucidated in future studies as well."

The reason for removing these sentences is that we do not have sufficient evidence to demonstrate whether longer retention times or thicker active layers affect vegetation growth. Although the length of residence time may hinder or promote plants growth, because water shortage in soil layer may occur when water MRT is too short, but longer MRT can hamper the water infiltration consequently to root anoxia, thus, affecting the plant growth (Ma et al., 2019). Thus, these sentences have been deleted in the revised manuscript.

**Line Comments:**

Line 36. "The progressive increase of the permafrost active layer thickness has exacerbated the increased water storage capacity of permafrost and exerted a higher contribution of groundwater to river

water." This is statement is incorrect, do the authors mean the active zone? I do not think the active capacity of permafrost has increased.

Response: This sentence in our manuscript is inaccurate. Based on the findings of Hin zman et al (2005) and Woo et al (2008), we rephrased this sentence as (line 36-37):

*This degradation, accompanied by progressive increase of the active layer, could higher water content in the soil column, thereby increasing the groundwater storage capacity.*

Line 40. "However, it remains unclear how permafrost changes would alter water storage and movement in permafrost catchment." Is this true? I think there is considerable literature suggesting otherwise.

Response: Thank you for the comment. This sentence in our manuscript is incorrect. Because, by searching the literature, changes in hydrological process caused by permafrost changes have been extensively observed (Throckmorton et al., 2016; Li et al., 2020a). This sentence has been deleted in the manuscript.

Line 52: "Therefore, the influence of permafrost changes, climatic factors, and vegetation variations on catchment MRT in a high-altitude permafrost catchment is seldom evaluated". This is true, but MRT and water ages have been reported and should be cited.

Response: Thank you for the comment. We added relevant studies on water MRT in permafrost catchments in the revised manuscript (line 55-57):

*For instance, Song et al. (2017) and Yang et al. (2021) have investigated water age in permafrost catchment in the hinterland of TP and explored the influence of vegetation and climatic factors on water age. Effects of permafrost freeze–thaw cycles on water MRT in Arctic permafrost catchment have also been reported (Tetzlaff et al., 2018).*

Line 68: "The findings from our study will deepen our understanding of the hydrological process in permafrost regions and will be important for water resources supply and safety in the TP." I am unsure if this manuscript does this. There is little talk of water supply and safety and the last sentence of the introduction should be strengthened.

Response: Thank you for the suggestion. Although MRT estimation has broad implications for evaluating water quality and contaminant transport (Jasechko et al., 2016; Tetzlaff et al., 2014; Yang et al., 2021), we did not conduct the relationship between MRT and pollutant transport or water quality in this manuscript due to a lack of water quality data. Thus, we rephrased this sentence as (line 73-74):

*The findings from our study will expand our understanding of the hydrological process in permafrost regions under global warming.*

Line 171: The first sentence makes no sense and the first paragraph beginning line 170 is very confusing.

Response: Thank you for the comment. We deleted the first two sentences of this paragraph in the revised manuscript.

Line 178: What other source waters would there be other than precipitation?

Response: Precipitation and ice meltwater from a deeper soil layer was the important source waters in the permafrost catchment that not covered by snow (Sugimoto et al., 2003a; Throckmorton et al., 2016). But a previous study found that melting ground ice in permafrost had little contribution to observed runoff variations (Landerer et al., 2010a). We added the statement in the revised manuscript as follows (line 199-203):

*In addition, ice meltwater from deeper soil layers is also a source of water replenishment (Wang et al., 2022; Sugimoto et al., 2003b; Throckmorton et al., 2016), although a previous study found that melting ground ice in permafrost had little contribution to the observed runoff variations (Landerer et al., 2010b).*

Line 183. I do not believe Xia et al., 2021 is the appropriate reference.

Response: Thank you for the suggestion. We removed this reference Xia et al (2021) and added the appropriate historical reference "Throckmorton et al (2016)" in the revised manuscript (line 204).

Line 185. I am unsure how the thawing and freezing of soils affects this. More details are needed. Also the next sentence regarding the slower slope associated with longer residence time. This is confusing and I'm not sure correct. The final sentence (line 188/9) also needs appropriate support.

Response: Thank you for the comments. With the thickening of the active layer, part of ice melt water (old water) will be mixed with precipitation. This old water tends to have a large water age and is subjected to evaporation over a long period, thereby lowing the slope of water lines (Throckmorton et al., 2016; Song et al., 2017). Thus, we speculate that the lower slope of supra-permafrost water may be related to the long residence time. In fact, the result of the longer residence time of water on the permafrost layer calculated later also confirmed this deduction. The following statement have been added in the revised manuscript as follows (line 205-208):

*During freezing and thawing, with the thickening of the active layer, part of ice melt water (old water) will be mixed with precipitation. This old water tends to have a large water age and is subjected to evaporation over a long period, thereby lowing the slope of water lines (Throckmorton et al., 2016; Song et al., 2017).*

Line 212. Appropriate historical reference are needed.

Response: Thank you for the suggestion. The following references have been added accordingly in the revised manuscript (line 231):

1.Woo, M. K. and Xia, Z.: Effects of Hydrology on the Thermal Conditions of the Active Layer: Paper presented at the 10th Northern Res. Basin Symposium. Hydrology Research, 27(1-2): 129-142, 1996.

2. Sugimoto, A, Naito, et al. Characteristics of soil moisture in permafrost observed in East Siberian taiga with stable isotopes of water. Hydrological Prochydrological Processes, 2003.

Line 214. More information on how freeze-thaw cycles affect isotopic composition are needed if the authors are invoking it. The correlations yield some results that do not make a lot of sense and literature cited is incorrect. For example, line 221-223 not supported, and the Landerer 2010 reference is form a very different scale endnote appropriate.

Response: Thank you for the suggestion. We added relevant information on how freeze-thaw cycles affect isotopic composition in the revised manuscript as follows (line 231-234):

*Rising air temperature promotes permafrost thawing and thickening of the active layer, thereby increasing soil moisture since more water sources with different isotopic compositions, such as atmospheric precipitation and meltwater from subsurface ice, input or release into the permafrost active layer (Sugimoto et al., 2003b; Throckmorton et al., 2016; Song et al., 2017).*

Part of the statistical results of the correlation matrix in the first manuscript is not make a lot of sense. Thus, we adjusted the correlation matrix to the correlation analysis chart (Fig 5, line 260) and removed the inaccurate literature citations (Landerer 2010 reference) in the revised manuscript.

[Figure]

***Figure 5****. Correlation analysis between stable isotopes and influencing factors. (a) and (b) correlations between stream water $\delta^{18}O$ with: soil temperature and moisture, respectively; (c) and (d) correlations between supra-permafrost water $\delta^{18}O$ with: soil temperature and stream water $\delta^{18}O$, respectively; (e) and (f) correlations between precipitation $\delta^{18}O$ with: stream and supra-permafrost water $\delta^{18}O$, respectively; Red line is the fitted line. Soil moisture and temperature data used in correlation analysis were normalized. "-" presents dimensionless.*

Line 233/4 needs to be rewritten.

Response: Thank you for the suggestion. We have rewritten this sentence as "*These findings reveal the differences in the water movement mechanisms at different stages of permafrost freezing and thawing processes*" (line 244-245).

The paragraph starting Line 235 does not make sense to me. Particularly at the end. Precipitation obviously has an influence on active layer water - I'm not sure what the authors are getting at. Is it that

there is no relation between the isotopic composition of active layer waters and precipitation isotopes? If so, this should clearly be stated.

Response: We agree with the reviewer's comment. We re-examined the relationship between stream, supra-permafrost water isotopes, and precipitation isotopes, and found that there are strong relations between the isotopic composition of stream water and supra-permafrost water and precipitation isotopes. We deleted this paragraph and revised these results in the revised manuscript as follows (line 250-257):

*The isotopic compositions in stream water are negatively correlated with soil temperature (Fig 5a: $R^2 = 0.08$, $P < 0.000$) and soil moisture (Fig 5b: $R^2 = 0.31$, $P < 0.000$). They are also strongly positively correlated with the isotopic compositions in precipitation (Fig 5e: $R^2 = 0.44$, $P < 0.000$). These findings indicate that the stream water was controlled by precipitation and freeze-thaw cycle of the permafrost active layer. Notably, our correlation results are in line with the previous studies in the Zuomaokong watershed of central TP (Song et al., 2017). At the same time, a very strong correlation was observed between stream water and supra-permafrost water isotopes (Fig 5d: $R^2 = 0.23$, $P < 0.000$). The significant positive correlation between precipitation and permafrost isotopes and stream water isotopes indicates that precipitation and supra-permafrost water are important recharge sources of stream water.*

It is not clear how the two-component IHS is applied, and how the values are determined. Are the average precipitation values volume weighted? How was snow accounted for? Was the IHS applied for the entire period to get these numbers (62 and 38%?). To compare these results to others in the literature, there is a lot more information that is needed. Did the authors consider IHS among years to assess mechanisms of variability to link to process?

Response: Thank you for the suggestion. In the first manuscript, we applied IHS to the entire period to get component proportions of stream water, but lacked the analysis of the runoff process mechanism. Thus, we use the precipitation-weighted average of precipitation isotopes to assess the components of stream water on a monthly scale. Because there is no snow cover in this study area due to the rapid melting of solid precipitation (snow) (Fig 1e), thus, snow is not considered as one of the component endmembers.

[Figure]

Figure 1e Photograph of the underlying surface in the catchment and meteorological station taken in June 2018. We added this picture to the study area map in the revised manuscript.

Meanwhile, we added more information on the IHS method in the "Materials and methods" section in the manuscript as follows:

**2.4 Isotope hydrograph separation method (line 148-158)**

[revised manuscript text omitted]

Line 271 to 273 are very confusing and need to be rewritten.

Response: Our aim is to highlight isotopic variation and the important input of precipitation to steam water and groundwater. Thus, this sentence has been rewritten as "*Moreover, we found that different water isotopes showed obvious seasonal variation, and precipitation was an important input of stream and supra-permafrost water.* " *in the revised manuscript* (line 293-294).

Line 299/300. "The longer MRT reflects more complex soil water retention and recharge processes (Ma et al., 2019b)." This is not clear at all. Why? A link to process must be made.

Response: Thank you for the suggestion. Because the longer MRT is related to the longer flow path for supra-permafrost water due to the existence of active layer increases the length of water flow path (Frampton and Destouni, 2015; Ma et al., 2019). Additionally, compared with surface runoff, supra-permafrost water stored in active layers is replenished by more old water, resulting in the longer MRT of supra-permafrost water. The following statements have been added in the revised manuscript as follows (line 380-384):

*In this study, the estimated MRT of supra-permafrost water was distinctly longer than that of stream water, which reflects the more complex water movement and recharge processes for supra-permafrost water. On the one hand, this is because supra-permafrost water stored in active layers is replenished*

*by more old water compared with surface runoff. On the other hand, it is related to the longer flow path for supra-permafrost water since the active layer increases the length of water flow path (Frampton and Destouni, 2015; Ma et al., 2019).*

Line 327 - correct the terminology.

Response: Very sorry for these incorrect terminologies. "Land surface-atmosphere interactions" was changed to "Land-atmosphere interactions" (line 406); "the active layer thickness of permafrost" was changed to "thickness of permafrost active layer" in the revised manuscript (line 401-402).

Line 329: "These strong correlations indicate that soil and air temperature are potentially efficient predictors of supra-permafrost water MRT". I do not support this statement at all. What is the mechanism? Is it the correlation analysis? It is a bold statement with little support.

Response: Thank you for the comment. MRT of permafrost catchments is highly dependent on depth of the active layer due to the warming effects (Frampton and Destouni, 2015). Air and soil temperature may affect the MRT variability by changing the thickness of permafrost active layer. Thus, we rephrased these statements as (line 404-403):

*The increase of air temperature can alter the temperature of shallow permafrost due to strong land-atmosphere interactions. This, in turn, increases the thickness of permafrost active layer, thereby allowing soil water to move into the deeper soil layer.*

*In this study, the significant positive correlation between MRT and active layer thickness support previous findings showing that the MRT of permafrost catchments is highly dependent on the depth of the active layer due to the warming effects (Frampton and Destouni, 2015). From the mechanism perspective, the deepening of the active layer can increase the length of water flow pathway and reduce transport velocities due to a shift in flow direction from horizontal saturated groundwater flow to vertical flow infiltrate into deeper subsurface, thereby increasing water MRT in permafrost catchment (Frampton and Destouni, 2015)..*

Line 336: "These results also support the findings from previous studies in terms of a relationship between permafrost changes and residence time. In particular, (Wright et al., 2009) have stated that MRT of permafrost catchments is highly dependent on the annual development of the active layer." I could not find any information on MRT in the Wright paper.

Response: Very sorry for this incorrect cite. A new literature has been added in the text. We rephrased this statement in the text as follows (line 408-410):

*In this study, the significant positive correlation between MRT and active layer thickness support previous findings showing that the MRT of permafrost catchments is highly dependent on the depth of the active layer due to the warming effects (Frampton and Destouni, 2015).*

Line 350: "The larger precipitation corresponds to lower temperature, yielding a thinner active layer, which, in turn making the active layer water to be saturated sooner". Is this with respect to this study? This is no a general or predicted finding.

Response: Thank you for the comment. Our aim is to emphasize that the effects of precipitation on the development of permafrost active layers, thus, affecting MTR variability. Previous studies have found that increased precipitation thinned permafrost active layer on the Tibetan Plateau (Zhou et al., 2021; Luo et al., 2020). Thus, we rephrased this statement in the text as follows (line 415-418):

*In central TP, the increase in precipitation thinned the permafrost active layer by decreasing soil heat flux, thereby cooling the soil and alleviating permafrost degradation (Zhou et al., 2021; Luo et al., 2020). This phenomenon subsequently triggered more water to rapidly flow into the river channel in the form of surface runoff, thereby, reducing the MRT of catchment water in the end.*

Line 357: "Interestingly, we found that the stream and supra-permafrost water MRT are both negatively correlated with NDVI ($R^2$= 0.29 and 0.53, respectively)" The entire issue of linking MRT to 'factors' in a regression analysis is flawed. Processes and explanations must be provided. Why would this be? There is some speculation but this could easily be spurious.

Response: Thank you for the comment. We re-examined the regression analysis results and added the trend direction and significance level to the Table 5. We have found that changes of the active layer affect the water MRT variability. Moreover, the decline of vegetation coverage induced increases of soil temperature and moisture, in which case they accelerated the active soil thawing and thickened the permafrost active layer (Wang et al., 2012). Therefore, we believed that changes in vegetation coverage may influence the water MRT by changing the thickness of permafrost active layer. We rephrased this statement in the revised manuscript as follows (line 427-431):

*The decline of vegetation coverage elevated soil temperature and moisture, which in turn accelerated the permafrost thawing and thickened the active layer (Wang et al., 2012). Moreover, water MRT variability can be influenced by the development of the active layer; therefore, we believed that the effect of vegetation on MRT is also driven by the variations of thickness of permafrost active layer.*

Response: We agree with the reviewer's comment. In the previous analysis, we found that the effect of vegetation on MRT variability is driven by changing the thickness of permafrost active layer. The length of residence time may hinder or promote plants growth, because water shortage in soil layer may occur when water MRT is too short, but longer MRT can hamper the water infiltration consequently to root anoxia, thus, affecting the plant growth (Ma et al., 2019). However, we do not have sufficient evidence

to demonstrate whether longer retention times or thicker active layers affect vegetation growth. Thus, this sentence has been deleted in the revised manuscript.

Line 365: "Moreover, the optimum residence time for vegetation growth should be elucidated in future studies as well." I am unsure as to what the authors mean.

Response: Thank you for the comment. Likewise, we do not have sufficient evidence to show whether longer retention times or thicker active layers affect vegetation growth. Thus, this sentence has been deleted in the revised manuscript.

Line 371: "It can be deduced that the estimated MRT of supra-permafrost water is valuable for evaluating the extent of permafrost degradation. Most importantly, it can be used to infer the effects of long-term climate, permafrost changes, and vegetation on the hydrologic regime in permafrost regions." This sentence is clearly wishful thinking and I am not sure the authors have shown this at all. If they have, they need to suggest how and why and what the implications are.

Response: We agreed with the reviewer's comment. MRT is a fundamental indicator of hydrological function within catchment. Through the previous analysis of the manuscript, in the permafrost watershed of the TP, the variation of permafrost active layer is an important factor in controlling water MRT. At the same time, both climatic and vegetation factors affect the MRT by influencing the development of active layer. Therefore, we can explore the changes in hydrological processes in permafrost watersheds by assessing water MRT in the context of climate and environmental change. Thus, we rephrased this statement in the revised manuscript as follows (line 437-439):

*Considering that MRT is a fundamental descriptor of hydrological function within catchment (Shah et al., 2017; Mcguire and Mcdonnell, 2006). Therefore, changes in hydrological processes in permafrost watersheds can be investigated by assessing water MRT in the context of climate and environmental change.*

Table 5. The data in this table is highly specific and incorrectly applies regression methods within and among data sets.

Response: Thank you for the suggestion. We re-examined the regression analysis results and added the trend direction and significance level to the Table 4 (line 342).

*Table 4. Correlations between active layer thickness, soil temperature, air temperature, precipitation, NDVI, and MRT. x indicates the factor as the independent variable.*

|  | | Regression based on mean MTT (days) | | | Regression based on MTT uncertainty (days) | | |
|---|---|---|---|---|---|---|---|
|  | Factor | Regression equation | $R^2$ | Sig | Regression equation | $R^2$ | Sig |
|  | ALT | $y=4.23exp(0.24x)+48.62$ | 0.11 | P>0.05 | $y=1.43exp(0.63x)+150.68$ | 0.44 ↑ | **P<0.05** |
| Supra-permafrost | ST | $y=929x+1299$ | 0.87 ↑ | **P<0.05** | $y=956x+1337$ | 0.78 ↑ | **P<0.05** |
| water | AT | $y=752x+4307$ | 0.69 ↑ | **P<0.05** | $y=767x+4283$ | 0.67 ↑ | **P<0.05** |
|  | P | $y=-1.89x+1169$ | 0.58 ↓ | **P<0.01** | $y=3479exp(-0.005x)-125$ | 0.64 ↓ | **P<0.01** |

| | | | | | | | |
|---|---|---|---|---|---|---|---|
| | *NDVI* | $y=0.103x^{-4.405}$ | *0.51* | *P=0.07* | $y=0.143x^{-4.286}$ | *0.65* ↓ | ***P<0.01*** |
| | *ALT* | $y=4.67exp(1.11x)+71.56$ | *0.81* ↑ | ***P<0.01*** | $y=3.02exp(0.37x)+59.23$ | *0.59* ↑ | ***P<0.01*** |
| | *ST* | $y=(1.7x+1.35)^{-0.26}$ | *0.81* ↑ | ***P<0.001*** | $y=3.79E+14exp(35.47x)+76$ | *0.79* ↑ | ***P<0.001*** |
| *Stream water* | *AT* | $y=51x+372$ | *0.05* | *P>0.05* | $y=42x+325$ | *0.01* | *P>0.05* |
| | *P* | $y=4.94E+5(x^{-1.77})$ | *0.47* ↓ | ***P<0.01*** | $y=1.67E+10(1.7x)^{-3.25}+52.6$ | *0.32* ↓ | ***P<0.01*** |
| | *NDVI* | $y=1.422x^{-2.394}$ | *0.20* ↓ | ***P<0.05*** | $y=1.50x^{-2.384}$ | *0.26* ↓ | ***P<0.05*** |

*Note: ALT = active layer thickness, ST = Soil temperature (℃), AT = air temperature (℃), P = Precipitation (mm), NDVI= normalized differential vegetation index; Sig indicates statistical significance; ↑ and ↓ indicates significant trend of increase and decrease, respectively; Bold font indicates that it passed significance test of 0.05.*

References: The authors reference largely literature from China when discussing general permafrost hydrological knowledge. While I am not dismissing any of this work, suggesting that permafrost acts as an aquiclude, then citing Gao et al. 2021. This is not 'new information' and has been identified for many decades in the North American and Russian literature. Perhaps it has also been long-identified in the Chinese literature, and I would suggest the authors here and elsewhere cite appropriate historical works as opposed to ones focused on the TP unless the work is directly related to processes in the TP and not ones that are more general.

Response: Thank you for the suggestion. The appropriate historical literatures involving permafrost hydrology have been added in the revised manuscript accordingly as follows:

1. Woo, M.K., 1990. Consequences of climatic change for hydrology in permafrost zones. Journal of Cold Regions Engineering, 4(1):15-20. (line 29, 421)

2. Hinzman, L.D., Kane, D.L., Woo, M. K., 2005. Permafrost Hydrology, Encyclopedia of Hydrological Sciences. (line 29, 421)

3. Woo, M.K., Kane, D.L., Carey, S.K., Yang, D.,: Progress in permafrost hydrology in the new millennium. Permafrost and Periglacial Processes, 19(2): 237-254, 2008. (line 37)

[revised manuscript text omitted]

Tetzlaff, D., Birkel, C., Dick, J., Geris, J., and Soulsby, C.: Storage dynamics in hydropedological units control hillslope connectivity, runoff generation, and the evolution of catchment transit time

distributions, Water Resources Research, 50, 969-985, https://doi.org/10.1002/2013WR014147, 2014.

Tetzlaff, D., Piovano, T., Ala-Aho, P., Smith, A., Carey, S. K., Marsh, P., Wookey, P. A., Street, L. E., and Soulsby, C.: Using stable isotopes to estimate travel times in a data-sparse Arctic catchment: Challenges and possible solutions, Hydrological Processes, 32, 1936-1952, https://doi.org/10.1002/hyp.13146, 2018.

Throckmorton, H. M., Newman, B. D., Heikoop, J. M., Perkins, G. B., Feng, X., Graham, D. E., O'Malley, D., Vesselinov, V. V., Young, J., Wullschleger, S. D., and Wilson, C. J.: Active layer hydrology in an arctic tundra ecosystem: quantifying water sources and cycling using water stable isotopes, Hydrological Processes, 30, 4972-4986, https://doi.org/10.1002/hyp.10883, 2016.

Uhlenbrook, S. and Hoeg, S.: Quantifying Uncertainties in Tracer-Based Hydrograph Separations: A Case Study for Two-, Three- and Five-Component Hydrograph Separations in a Mountainous Catchment, Hydrological Processes, 17, 431-453, https://doi.org/10.1002/hyp.1134, 2003.

Wang, G., Guangsheng, L., Chunjie, L., and Yan, Y.: The variability of soil thermal and hydrological dynamics with vegetation cover in a permafrost region, Agricultural and Forest Meteorology, 162-163, 44-57, https://doi.org/10.1016/j.agrformet.2012.04.006, 2012.

Wang, L., Zhao, L., Zhou, H., Liu, S., Du, E., Zou, D., Liu, G., Xiao, Y., Hu, G., Wang, C., Sun, Z., Li, Z., Qiao, Y., Wu, T., Li, C., and Li, X.: Contribution of ground ice melting to the expansion of Selin Co (lake) on the Tibetan Plateau, The Cryosphere, 16, 2745-2767, 10.5194/tc-16-2745-2022, 2022.

Xia, C., Liu, G., Zhou, J., Meng, Y., Chen, K., Gu, P., Yang, M., Huang, X., and Mei, J.: Revealing the impact of water conservancy projects and urbanization on hydrological cycle based on the distribution of hydrogen and oxygen isotopes in water, Environmental Science and Pollution Research, 28, 40160-40177, 10.1007/s11356-020-11647-6, 2021.

Yang, Y., Wu, Q., Yun, H., Jin, H., and Zhang, Z.: Evaluation of the hydrological contributions of permafrost to the thermokarst lakes on the Qinghai–Tibet Plateau using stable isotopes, Global and Planetary Change, 140, 1-8, https://doi.org/10.1016/j.gloplacha.2016.03.006, 2016.

Yang, Y., Weng, B., Yan, D., Gong, X., Dai, Y., and Niu, Y.: A preliminary estimate of how stream water age is influenced by changing runoff sources in the Nagqu river water shed, Qinghai-Tibet Plateau, Hydrological Processes, 35, e14380, https://doi.org/10.1002/hyp.14380, 2021.

Zhou, Z., Zhou, F., Zhang, M., Lei, B., and Ma, Z.: Effect of increasing rainfall on the thermal—moisture dynamics of permafrost active layer in the central Qinghai—Tibet Plateau, Journal of Mountain Science, 18, 2929-2945, https://doi.org/10.1007/s11629-021-6707-5, 2021.

Zhu, X., Wu, T., Zhao, L., Yang, C., Zhang, H., Xie, C., Li, R., Wang, W., Hu, G., Ni, J., Du, Y., Yang, S., Zhang, Y., Hao, J., Yang, C., Qiao, Y., and Shi, J.: Exploring the contribution of precipitation to water within the active layer during the thawing period in the permafrost regions of central Qinghai-Tibet Plateau by stable isotopic tracing, Science of The Total Environment, 661, 630-644, https://doi.org/10.1016/j.scitotenv.2019.01.064, 2019.

---

## Author Response (AR5)

**Response to Anonymous Referee #1**

We are very grateful for your valuable and instructive comments and suggestions. The review comments are listed below and marked in blue, followed by the detailed responses marked in black. The sentences added in the revised manuscript were marked in red and italic.

Kind regards,

Xiaobo He

(on behalf of the co-authors)

This study uses stable water isotopes to look at the mean residence time (MRT) for a catchment in the Tibetan Plateau. The novelty here is the long-term nature of the data series being leveraged for the MRT estimate as these types of sampling campaigns are challenging to coordinate in cold and alpine regions. The study is well written and well structured making it easy to read. Still, I do struggle some with the uniqueness of the study presented as while these data types of are challenge to collect and not often presented in the literature, there is a question of what we learn here for this catchment that advances beyond previous regional efforts like in Song et al. (2017)? I think bringing forward the improved process understanding in face of the possible uncertainty is needed here to move this manuscript beyond a presentation of the uniqueness of place that leverages data alone.

Response: Thanks for your comment. Song et al (2017) previously focused on studying the young water fraction of river water in the Zuomaokong watershed of the hinterland of the Tibetan plateau (TP) and its controlling factors (topography and vegetation), but not considered the effects of permafrost changes, especially active layer changes, on permafrost hydrological processes. Comparatively, we investigated not only the streamwater MRT, but also the groundwater (supra-permafrost water) in a long time series (5-8 years). Moreover, we analysed the impact of permafrost changes on MRT to explore permafrost hydrological processes. We believe these are aspects go beyond previous research by Song et al (2017).

The MRT estimation in our study does have some uncertainty, thus, we recalculated the uncertainty of MRT and analyzed the possible reasons for this uncertainty, including model assumptions, spatial variability of isotope input and output, and isotopic fractionation. We added relevant discussion on MRT uncertainty in the subsequent reply and revised manuscript (line 440-498).

One aspect that needs attention is the intercomparison of MRTs between various catchments and studies presented in the manuscript. I appreciate the effort and thinking to place this one catchment in a broader context; however, the different methods and models used when estimating MRT can have significant impacts on the resolution MRT and the entire travel time distribution. Caution is needed when comparing absolute MRT with other catchments. I think if the authors want to keep these comparisons, more information needs to be added (like a column or two in Table 4) indicating the model type and technique used to estimate MRT. Further, a richer discussion of the impacts of the modeling assumptions should be provided as they pertain to this region. There has been significant research and

literature on these topics over the last decades and it seems some of the more modern interpretations are missing from this study. All in all, I would anticipate a more thoughtful consideration of the assumptions behind the convolution approach you are implementing here.

Response: Thank you for the suggestion. Indeed, MRT estimation based on different models may impact the intercomparison between various catchments and studies. Thus, we removed MRT studies that were not calculated by exponential model and added more information about the model type and technique in Table 5 (line 396-399).

***Table 5.*** *Statistics of MRT-related research results.*

| Site | Altitude (m) | Water type | Model type | Tracer | Data length (years) | MRT | References |
|---|---|---|---|---|---|---|---|
| **Our study site**[a] | 5100–5435 | Stream water | Exponential | $\delta^{18}O$ | 8 | 100 days | This study |
| Huanjiang t[b] | 272–627 | Stream water | Exponential | $\delta^{18}O$ | 2 | 300 days | (Wang et al., 2020) |
| Mandava[b] | 383 | Stream water | Exponential | $\delta^{18}O$ | 2 | 444 days | (Sanda et al., 2017) |
| Upper Váh[b] | 1500 | Stream water | Exponential | $\delta^{18}O$ | 4 | 390–570 days | (M. et al., 2011) |
| Dee[b] | 1000 | Stream water | Exponential | $\delta^{18}O$ | 3 | 601 days | (Soulsby et al., 2010) |
| Minjiang upper[b] | 300–7100 | Stream water | Exponential | $\delta^{18}O$ | 1 | 698 days | (Xia et al., 2021) |
| **Our study site**[a] | 5100–5435 | Groundwater | Exponential | $\delta^{18}O$ | 5 | 255 days | This study |
| Himalaya[a] | 1600–5200 | Groundwater | Exponential | $\delta D$ | 1 | 4.5 months | (Shah et al., 2017) |
| Vermigliana[b] | 1221 | Groundwater | Exponential | $\delta^{18}O$ | 1 | 1.3 years | (Chiogna et al., 2014) |
| Allt[b]Mharcaidh[b] | 300–1111 | Groundwater | Exponential | $\delta^{18}O$ | 4 | >5 years | (Soulsby et al., 2000) |
| Huanjiang[b] | 272–627 | Groundwater | Exponential | $\delta^{18}O$ | 2 | 161–1407 days | (Wang et al., 2020) |

Note: "a" indicates the catchment covered by permafrost; "b" indicates the catchment not covered by permafrost. The symbol "—" indicates no data were available in the references.

Meanwhile, the assumptions of exponential model have been added in "Materials and methods" (line 170-172) and "Uncertainty and limitations" sections (line 441-460):

*In this study, long-term stable isotopic data of stream and supra-permafrost water were used to estimate water MRT and determine the mechanism underlying MRT variability in a high-altitude permafrost catchment of the TP. Nonetheless, some uncertainty remains in the results of MRT estimation, including model assumptions, spatial variability of isotope input and output and isotopic fractionation.*

*Different transit time distribution (TTD) models are applicable to different watershed conditions (Małoszewski and Zuber, 1998), which may affect the assessment of residence time. The exponential model, a commonly used model for MRT estimation, describes the catchment with flow times that are exponentially distributed (Mcguire and Mcdonnell, 2006), which assumed that the system is in steady-state conditions and operates as a perfect mixer (Sánchez-Murillo et al., 2015; Smith, 1984; Chiogna et al., 2014). This perfect mixer indicates that the mixing between input and baseflow is rapid and complete, whereas an ideal mixing cannot exist in an aquifer, which is an important uncertainty source of the applied model (Małoszewski et al., 1983; Fenicia et al., 2010). Nevertheless, exponential model is suitable for MRT estimation in unconfined aquifers with shallow sampling points (Małoszewski and Zuber, 1998; Małoszewski et al., 1983; Stewart and Mcdonnell, 1991). In effect, the exponential TTD*

*model could also approximate TTD in some non-steady cases (Haitjema, 1995; Rodhe et al., 1996). In this study area, the underlying surface was relatively uniform with less landscape heterogeneity and characterized by rapid hydrological processes. Moreover, the active layer of permafrost belonged to an unconfined aquifer and functioned as a water reservoir, thereby allowing for more precipitation recharge into the active layer to mix with old water. The amplitudes of output isotopes (stream and supra-permafrost water) were much lower than those of input (precipitation) and the dominant contribution of supra-permafrost water to stream water, both of which indicated that the precipitation was well mixed with other water within the catchment. Thus, the exponential model is suitable for application in permafrost catchment to some extent.*

In addition, if there is a connection between the MRT and the unique processes in permafrost environment, it would be more insightful to describe them explicitly. Modeling literature (e.g. Frampton and Destouni, 2015) exists on the subject and would help reduce the ambiguity connecting water movement and process as they are considered in this study. Further, and connected with this comment, there is need to separate the result and discussion section in to two separate sections. Given the amount of data being presented and the analysis put forward, plenty of material for results. Also, mixing the two sections together as is currently done creates confusion about what your data show and how you are interpreting it relative to the science. And it would be good in a separate section of the discussion to consider more the potential limitations of the current study as they pertain to assumptions, data representativeness and the models being considered.

Response: Thank you for the suggestion. To be clear, we have described accordingly and explicitly the connection between the MRT and hydrological processes of permafrost in the revised manuscript as follows:

*In this study, the estimated MRT of supra-permafrost water was distinctly longer than that of stream water, which reflects the more complex water movement and recharge processes for supra-permafrost water. On the one hand, this is because supra-permafrost water stored in active layers is replenished by more old water compared with surface runoff. On the other hand, it is related to the longer flow path for supra-permafrost water since the active layer increases the length of water flow path (Frampton and Destouni, 2015; Ma et al., 2019).* (line 380-384)

*In this study, the significant positive correlation between MRT and active layer thickness support previous findings showing that the MRT of permafrost catchments is highly dependent on the depth of the active layer due to the warming effects (Frampton and Destouni, 2015). From the mechanism perspective, the deepening of the active layer can increase the length of water flow pathway and reduce transport velocities due to a shift in flow direction from horizontal saturated groundwater flow to vertical flow infiltrate into deeper subsurface, thereby increasing water MRT in permafrost catchment (Frampton and Destouni, 2015).* (line 404-408)

We separated the "Results and discussion" into two sections, and added a section "Uncertainty and limitations" to the "Discussion" section, where we discussed the uncertainty of estimated MRT, including model assumptions, spatial variability of isotope input and output, and isotopic fractionation. The following statements have been added in the revised manuscript (line 440-498):

*4.3 Uncertainty and limitations*

*In this study, long-term stable isotopic data of stream and supra-permafrost water were used to estimate water MRT and determine the mechanism underlying MRT variability in a high-altitude permafrost catchment of the TP. Nonetheless, some uncertainty remains in the results of MRT estimation, including model assumptions, spatial variability of isotope input and output and isotopic fractionation.*

*Different transit time distribution (TTD) models are applicable to different watershed conditions (Małoszewski and Zuber, 1998), which may affect the assessment of residence time. The exponential model, a commonly used model for MRT estimation, describes the catchment with flow times that are exponentially distributed (Mcguire and Mcdonnell, 2006), which assumed that the system is in steady-state conditions and operates as a perfect mixer (Sánchez-Murillo et al., 2015; Smith, 1984; Chiogna et al., 2014). This perfect mixer indicates that the mixing between input and baseflow is rapid and complete, whereas an ideal mixing cannot exist in an aquifer, which is an important uncertainty source of the applied model (Małoszewski et al., 1983; Fenicia et al., 2010). Nevertheless, exponential model is suitable for MRT estimation in unconfined aquifers with shallow sampling points (Małoszewski and Zuber, 1998; Małoszewski et al., 1983; Stewart and Mcdonnell, 1991). In effect, the exponential TTD model could also approximate TTD in some non-steady cases (Haitjema, 1995; Rodhe et al., 1996). In this study area, the underlying surface was relatively uniform with less landscape heterogeneity and characterized by rapid hydrological processes. Moreover, the active layer of permafrost belonged to an unconfined aquifer and functioned as a water reservoir, thereby allowing for more precipitation recharge into the active layer to mix with old water. The amplitudes of output isotopes (stream and supra-permafrost water) were much lower than those of input (precipitation) and the dominant contribution of supra-permafrost water to stream water, both of which indicated that the precipitation was well mixed with other water within the catchment. Thus, the exponential model is suitable for application in permafrost catchment to some extent.*

*In general, measurement inputs represent spatial and temporal inputs for the entire catchment (Mcguire and Mcdonnell, 2006). At the catchment scale, elevation, air temperature, and rainfall intensity may cause considerable variation in isotopic composition of precipitation, particularly in mountainous areas (Ingraham, 1998). Thus, inputs of tracer to the catchment system are highly variable in space and time and important sources of uncertainty in interpretation of catchment response (Mcguire and Mcdonnell, 2006; Hrachowitz et al., 2009). A previous study suggested that precipitation at high altitudes is characterized by high isotopic amplitudes (Jasechko et al., 2016), which may result in underestimation of MRT in the study area due to one sampling site for precipitation. In practice, the isotopic composition of precipitation is often sampled at one site (Mcguire and Mcdonnell, 2006).*

*Considering the catchment area of our study was relatively small (2.7 km$^2$) with an altitude drop of 300 m. The size of the selected catchment in this study was much smaller than that of most catchments previously reported (Mcguire and Mcdonnell, 2006). Therefore, the effects of elevation on meteorological data and precipitation isotopic variability are minor and one precipitation sampling location could represent the whole catchment to some extent. Additionally, this study only collected supra-permafrost water from one sampling point due to economic and logistical constraints in the alpine regions, which is a limitation in estimating MRT. Given that the supra-permafrost water is primarily derived from precipitation, the spatial variability of isotopes in supra-permafrost water may also be minor in such small catchment. Even so, the spatial variability of isotopes in supra-permafrost water may result in underestimation of MRT in the study area.*

*The fractionation effects attributed to evaporation may potentially increase the uncertainty of water age estimation due to its impact on isotopic compositions and signals (Richardson and Kimberley, 2010; Mcdonnell et al., 2010; Song et al., 2017). Hence, the fractionation effects during the transformation from actual precipitation to effective input must be considered (Mcdonnell et al., 2010; Rusjan et al., 2019). In the study area, the atmospheric precipitation is primarily solid; the solid precipitation will be melted rapidly over a short period following deposition to form liquid water that enters soil and river channels, therefore it is difficult for snowpack to exist within this catchment. Thus, we did not collect snowpack or snow melt water as an input signal for MRT estimation. Nonetheless, solid precipitation may be subjected to evaporative fractionation to some degree when melted to be surface and subsurface runoff, thereby increasing the uncertainty of MRT estimation. Considering the rapid transformation of snow into infiltrated water and low air temperature, the potential effect of evaporation on the isotopic composition in precipitation, and consequently on MTT estimates, is relatively limited, which was not considered in the MRT estimation in this catchment.*

*To further analyse the uncertainty of MRT derived from the seasonal variability of isotope composition in hydrological component, we used the amplitude coefficient of input and output to estimate the uncertainty of MRT and found it larger for water with long residence time. Regression analysis showed that after considering the uncertainty of MRT, an improved R$^2$ for the thickness of the active layer with fewer differences for other factors. This suggests that uncertainty of estimated MRT may affects the sensitivity of MRT to specific factors (Hu et al., 2020), indicating that the uncertainty of estimated MRT should be considered when discussing MRT influencing factors. Therefore, future research should consider the uncertainty of MRT and improve the accurate assessment of MRT in alpine catchments.*

*Overall, although there remain uncertainty and limitations for MRT estimation in our study, isotope-based MRT estimation is valuable for identifying changes in hydrological processes of the permafrost regions, where there is a lack of observational data. Thus, it is necessary to utilize more measurements in different sub-catchments to augment the data representativeness in future research.*

Given the complexity of sampling precipitation in cold regions, more information is needed to help the reader understand how you were sampling here. For example, were how was snow treated throughout the sampling? Were snowpacks or snow melt water collected and considered as inputs in any sense? Also, looking at the variation in elevation in the region, how representative of the catchment is the one meteorological station and precipitation sampling location? Rainfall isotopic composition is rather variable with elevation and snowpack and snow melt rates are really variable. How is the isotopic input variability considered within this study? It seems ignored based on the methodology presented.

Response: Thank you for the suggestions. We added more information about precipitation sampling in the revised manuscript as follows (line 118-120):

*Liquid precipitation samples were collected immediately following every precipitation event using bulk collector to minimize the effects of evaporation. Solid precipitation (snow) samples were collected into a plastic bag and taken to a warm place to be thawed, following which water samples were transferred into 50-mL PE bottles.*

The elevation, air temperature, and rainfall intensity may cause considerable variation of isotopic composition in precipitation, particularly in mountainous areas (Ingraham, 1998), which may result in underestimation of MRT in the study area due to one sampling site for precipitation. Considering the size of the selected catchment in our study is small, we believed that the effects of elevation on meteorological data and precipitation isotopic variability are minor and one precipitation sampling location could represent the whole catchment to some extent. The following statements have been added to the manuscript (line 462-472):

*At the catchment scale, elevation, air temperature, and rainfall intensity may cause considerable variation in isotopic composition of precipitation, particularly in mountainous areas (Ingraham, 1998). Thus, inputs of tracer to the catchment system are highly variable in space and time and important sources of uncertainty in interpretation of catchment response (Mcguire and Mcdonnell, 2006; Hrachowitz et al., 2009). A previous study suggested that precipitation at high altitudes is characterized by high isotopic amplitudes (Jasechko et al., 2016), which may result in underestimation of MRT in the study area due to one sampling site for precipitation. In practice, the isotopic composition of precipitation is often sampled at one site (Mcguire and Mcdonnell, 2006). Considering the catchment area of our study was relatively small ($2.7 \ km^2$) with an altitude drop of 300 m. The size of the selected catchment in this study was much smaller than that of most catchments previously reported (Mcguire and Mcdonnell, 2006). Therefore, the effects of elevation on meteorological data and precipitation isotopic variability are minor and one precipitation sampling location could represent the whole catchment to some extent.*

In the study area, the atmospheric precipitation is primarily solid; the solid precipitation will be melted rapidly over a short period following deposition to form liquid water that enters soil and river channels, therefore it is difficult for snowpack to exist within this catchment. Thus, we did not collect snowpack or snow melt water as an input signal for MRT estimation (see Fig 1e). Thus, we did not

collect snowpack or snow melt water as an input signal for MRT estimation. The following statements have been added to the manuscript (line 480-487):

*In the study area, the atmospheric precipitation is primarily solid; the solid precipitation will be melted rapidly over a short period following deposition to form liquid water that enters soil and river channels, therefore it is difficult for snowpack to exist within this catchment. Thus, we did not collect snowpack or snow melt water as an input signal for MRT estimation. Nonetheless, solid precipitation may be subjected to evaporative fractionation to some degree when melted to be surface and subsurface runoff, thereby increasing the uncertainty of MRT estimation. Considering the rapid transformation of snow into infiltrated water and low air temperature, the potential effect of evaporation on the isotopic composition in precipitation, and consequently on MTT estimates, is relatively limited, which was not considered in the MRT estimation in this catchment.*

[Figure]

Figure 1e Photograph of the underlying surface in the catchment and meteorological station taken in June 2018. We added this picture to the study area map in the revised manuscript.

The input variability and source water variability of only having one location for monitoring supra-permafrost water sampling seems as if it could confound the results and interpretation to some extent. Specifically, if there are large frozen regions upstream of the stream sampling location, these would have significant impacts on the ability of precipitation to transfer to the stream over the entire catchment. Variability of isotopic compositions in springs and sub-watersheds is well documented (e.g. Lyon et al. 2018). The spatial variability at play in the catchment must be either accounted for or the potential impacts at least taking into consideration via discussion within this study.

Response: Thank you for the suggestion. This study only collected supra-permafrost water from one sampling point due to economic and logistical constraints in alpine regions, which is a limitation for estimating MRT in our study. Considering the small size of the selected catchment in this study and the supra-permafrost water mainly derived from the contribution of precipitation, we believed that the

spatial variability of isotopes in supra-permafrost water also may be minor in such a small catchment. Even so, the spatial variability of isotopes in supra-permafrost water may also result in underestimation of MRT in the study area. The following statement have been added to the manuscript (line 469-476):

*The size of the selected catchment in this study was much smaller than that of most catchments previously reported (Mcguire and Mcdonnell, 2006). Therefore, the effects of elevation on meteorological data and precipitation isotopic variability are minor and one precipitation sampling location could represent the whole catchment to some extent. Additionally, this study only collected supra-permafrost water from one sampling point due to economic and logistical constraints in the alpine regions, which is a limitation in estimating MRT. Given that the supra-permafrost water is primarily derived from precipitation, the spatial variability of isotopes in supra-permafrost water may also be minor in such small catchment. Even so, the spatial variability of isotopes in supra-permafrost water may result in underestimation of MRT in the study area.*

Finally, some consideration of uncertainty should be presented. There are several fitted relationships that are being compared across the research. In and of themselves, these are wrought with uncertainty and confidence intervals that can impact the significance of the findings. I would want to see some assessment of the robustness of the results relative to the uncertainty or lack of representativeness of the data being presented. At the least, the two-component hydrograph can directly incorporate the uncertainty via the approach put forward by Genereux (1998). Without characterization of the uncertainty, I am left wondering how much of the results is driven by under-represented variability in the sampling at a catchment scale, the simplifying assumptions within the model, and the fitted equations that smooth out all the between event variability and extremes. That last point is rather important given potential flashy nature of these systems during certain times of the year and more dampened responses as the systems thaw seasonally.

Response: Thank you for the suggestions. The uncertainty of MRT results has been recalculated by the method described in Morales and Oswald (2020). We considered the uncertainty of the estimated MRT in the regression analysis. The results showed that after considering the uncertainty of MRT, the $R^2$ of the regression analysis for the thickness of permafrost active layer has been improved, while less difference for other factors (see Table 4, line 346-350). The following statement have been added to the manuscript (line 488-494):

*To further analyse the uncertainty of MRT derived from the seasonal variability of isotope composition in hydrological component, we used the amplitude coefficient of input and output to estimate the uncertainty of MRT and found it larger for water with long residence time. Regression analysis showed that after considering the uncertainty of MRT, an improved $R^2$ for the thickness of the active layer with fewer differences for other factors. This suggests that uncertainty of estimated MRT may affects the sensitivity of MRT to specific factors (Hu et al., 2020), indicating that the uncertainty of estimated MRT should be considered when discussing MRT influencing factors. Therefore, future*

*research should consider the uncertainty of MRT and improve the accurate assessment of MRT in alpine catchments.*

*Table 4. Relationships between active layer thickness, soil temperature, air temperature, precipitation, NDVI, and MRT. x indicates the factor as the independent variable.* (line 342)

| | | Regression based on mean MTT (days) | | | Regression based on MTT uncertainty (days) | | |
|---|---|---|---|---|---|---|---|
| | *Factor* | *Regression equation* | *$R^2$* | *Sig* | *Regression equation* | *$R^2$* | *Sig* |
| *Supra-permafrost water* | *ALT* | $y=4.23exp(0.24x)+48.62$ | *0.11* | *P>0.05* | $y=1.43exp(0.63x)+150.68$ | *0.44↑* | **P<0.05** |
| | *ST* | $y=929x+1299$ | *0.87↑* | **P<0.05** | $y=956x+1337$ | *0.78↑* | **P<0.05** |
| | *AT* | $y=752x+4307$ | *0.69↑* | **P<0.05** | $y=767x+4283$ | *0.67↑* | **P<0.05** |
| | *P* | $y=-1.89x+1169$ | *0.58↓* | **P<0.01** | $y=3479exp(-0.005x)-125$ | *0.64↓* | **P<0.01** |
| | *NDVI* | $y=0.103x^{-4.405}$ | *0.51* | *P=0.07* | $y=0.143x^{-4.286}$ | *0.65↓* | **P<0.05** |
| *Stream water* | *ALT* | $y=4.67exp(1.11x)+71.56$ | *0.81↑* | **P<0.01** | $y=3.02exp(0.37x)+59.23$ | *0.59↑* | **P<0.01** |
| | *ST* | $y=(1.7x+1.35)^{-0.26}$ | *0.81↑* | **P<0.001** | $y=3.79E+14exp(35.47x)+76$ | *0.79↑* | **P<0.001** |
| | *AT* | $y=51x+372$ | *0.05* | *P>0.05* | $y=42x+325$ | *0.01* | *P>0.05* |
| | *P* | $y=4.94E+5(x^{-1.77})$ | *0.47↓* | **P<0.01** | $y=1.67E+10(1.7x)^{-3.25}+52.6$ | *0.32↓* | **P<0.01** |
| | *NDVI* | $y=1.422x^{-2.394}$ | *0.20↓* | **P<0.05** | $y=1.50x^{-2.384}$ | *0.26↓* | **P<0.05** |

*Note: ALT = active layer thickness, ST = Soil temperature (℃), AT = air temperature (℃), P = Precipitation (mm), NDVI= normalized differential vegetation index; Sig indicates statistical significance; ↑ and ↓ indicates significant trend of increase and decrease, respectively; Bold font indicates that it passed significance test of 0.05.*

**Minor Comments**

L100: This sentence is random and does not make sense here. Further, not sure what you mean with efficiently?

Response: Thank you for the suggestion. This sentence has been deleted in the revised manuscript.

L171: This first sentence is odd. Separate the results and discussions to increase presentation clarity.

Response: Thank you for the suggestion. We have removed this sentence, spilt the "results and discussion" into two parts, and added a section "Uncertainty and limitations" to the "Discussion" section, which discussed the uncertainty of estimated MRT, including model assumptions, spatial variability of isotope input and output, and isotopic fractionation (line 440-498).

**References**

[revised manuscript text omitted]

**Response to Anonymous Referee #2**

We are very grateful for your valuable and instructive comments and suggestions. The review comments are listed below and marked in blue, followed by the detailed responses marked in black. The sentences added in the revised manuscript were marked in red and italic.

Kind regards,

Xiaobo He

(on behalf of the co-authors)

The paper by Wang and co-authors entitled "*Estimation of water residence time in a permafrost catchment in the Central Tibetan Plateau using long-term water stable isotopic data*" leverages a unique data set in a remote environment to shed light on the 'mean residence time' (MRT) of groundwater and stream water using transit time approaches. They seek to highlight how the active layer and permafrost influence MRT and draw inferences on permafrost hydrology.

The paper is well written and clear. The figures are straight-forward and interpretable. I have a number of editorial comments at the end, yet I have considerable concerns about the analysis that I believe must be addressed before this manuscript is suitable for publication. The data is novel and is of considerable value to the hydrological and permafrost community, yet there are large uncertainties and at times I believe mis-application of methods that the authors need to consider before this manuscript is suitable for publication.

Response: Thanks a lot for the comments and suggestions. We analyzed the uncertainty of runoff separation and MRT estimation, and modified the inaccurate application of correlation analysis methods in the revised manuscript. The detailed statements have been added to the subsequent reply and revised manuscript.

**General Comments:**

The authors use the exponential model as opposed to the more widely used gamma distribution. I am curious as to why this is. This will have considerable impact on the MRT calculation and should be discussed.

Response: Thank you for the comments. The reasons that we use the exponential model to assess MRT are as follows:

(1) The exponential model describes a catchment with flow times that are exponentially distributed, including pathways with very short transit times (Mcguire and Mcdonnell, 2006), which assumed that the system is in steady-state conditions and operates as a perfect mixer (Chiogna et al., 2014; Sánchez-Murillo et al., 2015a; Smith, 1984). This perfect mixer means that the mixing between input and baseflow is rapid and complete, whereas an ideal mixing cannot exist in an aquifer. Therefore, exponential model is suitable for MRT estimation in unconfined aquifers with shallow sampling points (Małoszewski et al., 1983; Stewart and Mcdonnell, 1991; Xia et al., 2021). In effect, the exponential

TTD model could also approximate TTDs in some non-steady cases (Haitjema, 1995; Rodhe et al., 1996). In our study, underlying surface is relative uniform with less landscape heterogeneity and characterized by rapid hydrological processes. Moreover, the active layer of permafrost belonged to an unconfined aquifer and functioned as a water reservoir, allowing more precipitation recharges into the active layer to mix with old water. Thus, the applicability of exponential model can be satisfied basically to some extent;

(2) The exponential model is essentially a special case of the gamma model with the shape factor parameter $\alpha$ equal to 1 (Mcguire and Mcdonnell, 2006). If catchments are modelled as well-mixed reservoirs, their travel times follow an exponential distribution (Kirchner et al., 2000). In our study, the amplitudes of output isotopes (stream and supra-permafrost water) were much lower than that of the input (precipitation), in effect, indicating that the precipitation was well mixed with other water within the catchment;

(3) We calculated the MRT of stream water using partial isotope data (2014-2020) based on exponential and gamma models, respectively, and found that these two results were very consistent ($R^2$=0.97, $P$<0.001). Given that exponential model is the most widely used model for MRT assessment over the past decades (Mcguire and Mcdonnell, 2006; Seeger and Weiler, 2014; Rusjan et al., 2019) and the calculation process of MRT using exponential model is often more convenient compared with the gamma model.

Aboveall, exponential model was selected to assess water residence time in our study.

Meanwhile, the following statements have been added to the revised manuscript as follows (line 446-460):

*The exponential model, a commonly used model for MRT estimation, describes the catchment with flow times that are exponentially distributed (Mcguire and Mcdonnell, 2006), which assumed that the system is in steady-state conditions and operates as a perfect mixer (Sánchez-Murillo et al., 2015b; Smith, 1984; Chiogna et al., 2014). This perfect mixer indicates that the mixing between input and baseflow is rapid and complete, whereas an ideal mixing cannot exist in an aquifer, which is an important uncertainty source of the applied model (Małoszewski et al., 1983; Fenicia et al., 2010). Nevertheless, exponential model is suitable for MRT estimation in unconfined aquifers with shallow sampling points (Małoszewski and Zuber, 1998; Małoszewski et al., 1983; Stewart and Mcdonnell, 1991). In effect, the exponential TTD model could also approximate TTD in some non-steady cases (Haitjema, 1995; Rodhe et al., 1996). In this study area, the underlying surface was relatively uniform with less landscape heterogeneity and characterized by rapid hydrological processes. Moreover, the active layer of permafrost belonged to an unconfined aquifer and functioned as a water reservoir, thereby allowing for more precipitation recharge into the active layer to mix with old water. The amplitudes of output isotopes (stream and supra-permafrost water) were much lower than those of input (precipitation) and the dominant contribution of supra-permafrost water to stream water, both of which*

*indicated that the precipitation was well mixed with other water within the catchment. Thus, the exponential model is suitable for application in permafrost catchment to some extent.*

The correlation analysis is highly flawed and must be revisited. It appears the authors use any type of correlation against variables with different units, etc., to 'look around for relationships'. This is not statistically robust in any way. Values must be transposed/normalized to compare among factors, and the type of correlation must be explained. Figure 6 shows two 'best fit' lines with either low or no relationships. This is a regression analysis. Furthermore, this data is ALL serially correlated which needs to be accounted for. As it stands, the authors have a lot of work to do to justify this analytical approach. Binning data together from across seasons, etc., truly make this confounding as the active layer changes.

Response: Thank you for the suggestions. Part of the statistical results of the correlation matrix in the first manuscript did not make sense. Thus, the correlation matrix was adjusted to the correlation analysis chart by bring together all the data from across seasons (Fig 5, line 255). Additionally, we added the type of correlation to these correlation analysis chart. The soil moisture and temperature data were normalized in the correlation analysis.

[Figure]

***Figure 5***. *Correlation analysis between stable isotopes and influencing factors. (a) and (b) correlations between stream water $\delta^{18}O$ with: soil temperature and moisture, respectively; (c) and (d) correlations between supra-permafrost water $\delta^{18}O$ with: soil temperature and stream water $\delta^{18}O$, respectively; (e) and (f) correlations between precipitation $\delta^{18}O$ with: stream and supra-permafrost water $\delta^{18}O$, respectively; Red line is the fitted line. Soil moisture and temperature data used in correlation analysis were normalized; "-" presents dimensionless.*

More information on the IHS method is needed.

Response: Thank you for the suggestions. In the first manuscript, we applied IHS to the entire period to get component proportions of stream water, but lacks the analysis of the mechanism of runoff process. Thus, we have assessed stream water components on a monthly scale and added relevant information on the IHS method in the revised manuscript. We added more information on the IHS method in the "Materials and methods" section in the manuscript as follows (line 148-158):

*In this study, the precipitation-weighted average of precipitation isotopes was used to assess stream water components on a monthly scale to determine the mechanism of runoff process in permafrost*

*catchments.*

*The uncertainty in hydrograph separations generally included two aspects, one is the analysis error of tracer concentrations, while the other is the spatial and temporal variations of the tracer of components (Uhlenbrook and Hoeg, 2003), calculated using the Gaussian error propagation technique (Genereux, 1998):*

$$w_y = \sqrt{\left(\frac{\partial y}{\partial x_1}w_{x1}\right)^2 + \left(\frac{\partial y}{\partial x_2}w_{x2}\right)^2 + ... + \left(\frac{\partial y}{\partial x_n}w_{xn}\right)^2} \qquad (5)$$

*where w represents the uncertainty in the variable specified in the subscript and y is the contribution of a specific streamflow component x to streamwater.*

More appropriate literature is needed, along with less bold statements about the importance/influence of this work.

Response: Thank you for the suggestions. We added the following appropriate literatures in the revised manuscript accordingly:

1. Woo, M.K., 1990. Consequences of climatic change for hydrology in permafrost zones. Journal of Cold Regions Engineering, 4(1):15-20. (line 29, 421)

2. Hinzman, L.D., Kane, D.L., Woo, M. K., 2005. Permafrost Hydrology, Encyclopedia of Hydrological Sciences. (line 29, 421)

3. Woo, M.K., Kane, D.L., Carey, S.K., Yang, D.,: Progress in permafrost hydrology in the new millennium. Permafrost and Periglacial Processes, 19(2): 237-254, 2008. (line 37)

4. Woo, M. K. and Xia, Z.: Effects of Hydrology on the Thermal Conditions of the Active Layer, Hydrology Research, 27, 129-142, https://doi.org/10.2166/nh.1996.0024, 1996. (line 231)

5. Sugimoto, A., Naito, D., Yanagisawa, N., Ichiyanagi, K., Kurita, N., Kubota, J., Kotake, T., Ohata, T., Maximov, T. C., and Fedorov, A. N.: Characteristics of soil moisture in permafrost observed in East Siberian taiga with stable isotopes of water, Hydrological Processes, 17, 1073-1092, https://doi.org/10.1002/hyp.1180, 2003. (line 197, 234)

6. Throckmorton, H. M., Newman, B. D., Heikoop, J. M., Perkins, G. B., Feng, X., Graham, D. E., O'Malley, D., Vesselinov, V. V., Young, J., Wullschleger, S. D., and Wilson, C. J.: Active layer hydrology in an arctic tundra ecosystem: quantifying water sources and cycling using water stable isotopes, Hydrological Processes, 30, 4972-4986, https://doi.org/10.1002/hyp.10883, 2016. (line 203)

7.Woo, M. K. and Xia, Z.: Effects of Hydrology on the Thermal Conditions of the Active Layer: Paper presented at the 10th Northern Res. Basin Symposium. Hydrology Research, 27(1-2): 129-142, 1996. (line 231)

8. Sugimoto, A, Naito, et al.: Characteristics of soil moisture in permafrost observed in East Siberian taiga with stable isotopes of water. Hydrological Prochydrological Processes, 2003. (line 234)

Additionally, we adjusted and revised some sentences about importance and influence of this work accordingly:

*The findings from our study will expand our understanding of the hydrological process in permafrost regions under global warming.* (line 72, line 513-514).

*Therefore, changes in hydrological processes in permafrost watersheds can be investigated by assessing water MRT in the context of climate and environmental change* (line 438-439).

Many of the conclusions are not supported by the data.

Response: Thank you for the suggestions. We have revised and explained the conclusions that are not supported by the data based on the line comments. Specific revisions are detailed in line comment response and the revised manuscript accordingly.

We explained and rephrased this sentence "Line 329 These strong correlations indicate that soil and air temperature are potentially efficient predictors of supra-permafrost water MRT":

*The increase of air temperature can alter the temperature of shallow permafrost due to strong land-atmosphere interactions. This, in turn, increases the thickness of permafrost active layer, thereby allowing soil water to move into the deeper soil layer. In this study, the significant positive correlation between MRT and active layer thickness support previous findings showing that the MRT of permafrost catchments is highly dependent on the depth of the active layer due to the warming effects (Frampton and Destouni, 2015). From the mechanism perspective, the deepening of the active layer can increase the length of water flow pathway and reduce transport velocities due to a shift in flow direction from horizontal saturated groundwater flow to vertical flow infiltrate into deeper subsurface, thereby increasing water MRT in permafrost catchment (Frampton and Destouni, 2015)* (line 401-409).

We deleted these sentences "Line 364: However, it remains unclear whether a positive feedback mechanism exists between vegetation and permafrost active layer changes or not" and "Line 365: Moreover, the optimum residence time for vegetation growth should be elucidated in future studies as well."

The reason for removing these sentences is that we do not have sufficient evidence to demonstrate whether longer retention times or thicker active layers affect vegetation growth. Although the length of residence time may hinder or promote plants growth, because water shortage in soil layer may occur when water MRT is too short, but longer MRT can hamper the water infiltration consequently to root anoxia, thus, affecting the plant growth (Ma et al., 2019). Thus, these sentences have been deleted in the revised manuscript.

**Line Comments:**

Line 36. "The progressive increase of the permafrost active layer thickness has exacerbated the increased water storage capacity of permafrost and exerted a higher contribution of groundwater to river

water." This is statement is incorrect, do the authors mean the active zone? I do not think the active capacity of permafrost has increased.

Response: This sentence in our manuscript is inaccurate. Based on the findings of Hin zman et al (2005) and Woo et al (2008), we rephrased this sentence as (line 36-37):

*This degradation, accompanied by progressive increase of the active layer, could higher water content in the soil column, thereby increasing the groundwater storage capacity.*

Line 40. "However, it remains unclear how permafrost changes would alter water storage and movement in permafrost catchment." Is this true? I think there is considerable literature suggesting otherwise.

Response: Thank you for the comment. This sentence in our manuscript is incorrect. Because, by searching the literature, changes in hydrological process caused by permafrost changes have been extensively observed (Throckmorton et al., 2016; Li et al., 2020a). This sentence has been deleted in the manuscript.

Line 52: "Therefore, the influence of permafrost changes, climatic factors, and vegetation variations on catchment MRT in a high-altitude permafrost catchment is seldom evaluated". This is true, but MRT and water ages have been reported and should be cited.

Response: Thank you for the comment. We added relevant studies on water MRT in permafrost catchments in the revised manuscript (line 55-57):

*For instance, Song et al. (2017) and Yang et al. (2021) have investigated water age in permafrost catchment in the hinterland of TP and explored the influence of vegetation and climatic factors on water age. Effects of permafrost freeze–thaw cycles on water MRT in Arctic permafrost catchment have also been reported (Tetzlaff et al., 2018).*

Line 68: "The findings from our study will deepen our understanding of the hydrological process in permafrost regions and will be important for water resources supply and safety in the TP." I am unsure if this manuscript does this. There is little talk of water supply and safety and the last sentence of the introduction should be strengthened.

Response: Thank you for the suggestion. Although MRT estimation has broad implications for evaluating water quality and contaminant transport (Jasechko et al., 2016; Tetzlaff et al., 2014; Yang et al., 2021), we did not conduct the relationship between MRT and pollutant transport or water quality in this manuscript due to a lack of water quality data. Thus, we rephrased this sentence as (line 73-74):

*The findings from our study will expand our understanding of the hydrological process in permafrost regions under global warming.*

Line 171: The first sentence makes no sense and the first paragraph beginning line 170 is very confusing.

Response: Thank you for the comment. We deleted the first two sentences of this paragraph in the revised manuscript.

Line 178: What other source waters would there be other than precipitation?

Response: Precipitation and ice meltwater from a deeper soil layer was the important source waters in the permafrost catchment that not covered by snow (Sugimoto et al., 2003a; Throckmorton et al., 2016). But a previous study found that melting ground ice in permafrost had little contribution to observed runoff variations (Landerer et al., 2010a). We added the statement in the revised manuscript as follows (line 199-203):

*In addition, ice meltwater from deeper soil layers is also a source of water replenishment (Wang et al., 2022; Sugimoto et al., 2003b; Throckmorton et al., 2016), although a previous study found that melting ground ice in permafrost had little contribution to the observed runoff variations (Landerer et al., 2010b).*

Line 183. I do not believe Xia et al., 2021 is the appropriate reference.

Response: Thank you for the suggestion. We removed this reference Xia et al (2021) and added the appropriate historical reference "Throckmorton et al (2016)" in the revised manuscript (line 204).

Line 185. I am unsure how the thawing and freezing of soils affects this. More details are needed. Also the next sentence regarding the slower slope associated with longer residence time. This is confusing and I'm not sure correct. The final sentence (line 188/9) also needs appropriate support.

Response: Thank you for the comments. With the thickening of the active layer, part of ice melt water (old water) will be mixed with precipitation. This old water tends to have a large water age and is subjected to evaporation over a long period, thereby lowing the slope of water lines (Throckmorton et al., 2016; Song et al., 2017). Thus, we speculate that the lower slope of supra-permafrost water may be related to the long residence time. In fact, the result of the longer residence time of water on the permafrost layer calculated later also confirmed this deduction. The following statement have been added in the revised manuscript as follows (line 205-208):

*During freezing and thawing, with the thickening of the active layer, part of ice melt water (old water) will be mixed with precipitation. This old water tends to have a large water age and is subjected to evaporation over a long period, thereby lowing the slope of water lines (Throckmorton et al., 2016; Song et al., 2017).*

Line 212. Appropriate historical reference are needed.

Response: Thank you for the suggestion. The following references have been added accordingly in the revised manuscript (line 231):

1.Woo, M. K. and Xia, Z.: Effects of Hydrology on the Thermal Conditions of the Active Layer: Paper presented at the 10th Northern Res. Basin Symposium. Hydrology Research, 27(1-2): 129-142, 1996.

2. Sugimoto, A, Naito, et al. Characteristics of soil moisture in permafrost observed in East Siberian taiga with stable isotopes of water. Hydrological Prochydrological Processes, 2003.

Line 214. More information on how freeze-thaw cycles affect isotopic composition are needed if the authors are invoking it. The correlations yield some results that do not make a lot of sense and literature cited is incorrect. For example, line 221-223 not supported, and the Landerer 2010 reference is form a very different scale endnote appropriate.

Response: Thank you for the suggestion. We added relevant information on how freeze-thaw cycles affect isotopic composition in the revised manuscript as follows (line 231-234):

*Rising air temperature promotes permafrost thawing and thickening of the active layer, thereby increasing soil moisture since more water sources with different isotopic compositions, such as atmospheric precipitation and meltwater from subsurface ice, input or release into the permafrost active layer (Sugimoto et al., 2003b; Throckmorton et al., 2016; Song et al., 2017).*

Part of the statistical results of the correlation matrix in the first manuscript is not make a lot of sense. Thus, we adjusted the correlation matrix to the correlation analysis chart (Fig 5, line 260) and removed the inaccurate literature citations (Landerer 2010 reference) in the revised manuscript.

[Figure]

***Figure 5****. Correlation analysis between stable isotopes and influencing factors. (a) and (b) correlations between stream water $\delta^{18}O$ with: soil temperature and moisture, respectively; (c) and (d) correlations between supra-permafrost water $\delta^{18}O$ with: soil temperature and stream water $\delta^{18}O$, respectively; (e) and (f) correlations between precipitation $\delta^{18}O$ with: stream and supra-permafrost water $\delta^{18}O$, respectively; Red line is the fitted line. Soil moisture and temperature data used in correlation analysis were normalized. "-" presents dimensionless.*

Line 233/4 needs to be rewritten.

Response: Thank you for the suggestion. We have rewritten this sentence as "*These findings reveal the differences in the water movement mechanisms at different stages of permafrost freezing and thawing processes*" (line 244-245).

The paragraph starting Line 235 does not make sense to me. Particularly at the end. Precipitation obviously has an influence on active layer water - I'm not sure what the authors are getting at. Is it that

there is no relation between the isotopic composition of active layer waters and precipitation isotopes? If so, this should clearly be stated.

Response: We agree with the reviewer's comment. We re-examined the relationship between stream, supra-permafrost water isotopes, and precipitation isotopes, and found that there are strong relations between the isotopic composition of stream water and supra-permafrost water and precipitation isotopes. We deleted this paragraph and revised these results in the revised manuscript as follows (line 250-257):

*The isotopic compositions in stream water are negatively correlated with soil temperature (Fig 5a: $R^2 = 0.08$, $P < 0.000$) and soil moisture (Fig 5b: $R^2 = 0.31$, $P < 0.000$). They are also strongly positively correlated with the isotopic compositions in precipitation (Fig 5e: $R^2 = 0.44$, $P < 0.000$). These findings indicate that the stream water was controlled by precipitation and freeze-thaw cycle of the permafrost active layer. Notably, our correlation results are in line with the previous studies in the Zuomaokong watershed of central TP (Song et al., 2017). At the same time, a very strong correlation was observed between stream water and supra-permafrost water isotopes (Fig 5d: $R^2 = 0.23$, $P < 0.000$). The significant positive correlation between precipitation and permafrost isotopes and stream water isotopes indicates that precipitation and supra-permafrost water are important recharge sources of stream water.*

It is not clear how the two-component IHS is applied, and how the values are determined. Are the average precipitation values volume weighted? How was snow accounted for? Was the IHS applied for the entire period to get these numbers (62 and 38%?). To compare these results to others in the literature, there is a lot more information that is needed. Did the authors consider IHS among years to assess mechanisms of variability to link to process?

Response: Thank you for the suggestion. In the first manuscript, we applied IHS to the entire period to get component proportions of stream water, but lacked the analysis of the runoff process mechanism. Thus, we use the precipitation-weighted average of precipitation isotopes to assess the components of stream water on a monthly scale. Because there is no snow cover in this study area due to the rapid melting of solid precipitation (snow) (Fig 1e), thus, snow is not considered as one of the component endmembers.

[Figure]

Figure 1e Photograph of the underlying surface in the catchment and meteorological station taken in June 2018. We added this picture to the study area map in the revised manuscript.

Meanwhile, we added more information on the IHS method in the "Materials and methods" section in the manuscript as follows:

*2.4 Isotope hydrograph separation method* **(line 148-158)**

[revised manuscript text omitted]

Line 271 to 273 are very confusing and need to be rewritten.

Response: Our aim is to highlight isotopic variation and the important input of precipitation to steam water and groundwater. Thus, this sentence has been rewritten as "*Moreover, we found that different water isotopes showed obvious seasonal variation, and precipitation was an important input of stream and supra-permafrost water.* " *in the revised manuscript* (line 293-294).

Line 299/300. "The longer MRT reflects more complex soil water retention and recharge processes (Ma et al., 2019b)." This is not clear at all. Why? A link to process must be made.

Response: Thank you for the suggestion. Because the longer MRT is related to the longer flow path for supra-permafrost water due to the existence of active layer increases the length of water flow path (Frampton and Destouni, 2015; Ma et al., 2019). Additionally, compared with surface runoff, supra-permafrost water stored in active layers is replenished by more old water, resulting in the longer MRT of supra-permafrost water. The following statements have been added in the revised manuscript as follows (line 380-384):

*In this study, the estimated MRT of supra-permafrost water was distinctly longer than that of stream water, which reflects the more complex water movement and recharge processes for supra-permafrost water. On the one hand, this is because supra-permafrost water stored in active layers is replenished*

*by more old water compared with surface runoff. On the other hand, it is related to the longer flow path for supra-permafrost water since the active layer increases the length of water flow path (Frampton and Destouni, 2015; Ma et al., 2019).*

Line 327 - correct the terminology.

Response: Very sorry for these incorrect terminologies. "Land surface-atmosphere interactions" was changed to "Land-atmosphere interactions" (line 406); "the active layer thickness of permafrost" was changed to "thickness of permafrost active layer" in the revised manuscript (line 401-402).

Line 329: "These strong correlations indicate that soil and air temperature are potentially efficient predictors of supra-permafrost water MRT". I do not support this statement at all. What is the mechanism? Is it the correlation analysis? It is a bold statement with little support.

Response: Thank you for the comment. MRT of permafrost catchments is highly dependent on depth of the active layer due to the warming effects (Frampton and Destouni, 2015). Air and soil temperature may affect the MRT variability by changing the thickness of permafrost active layer. Thus, we rephrased these statements as (line 404-403):

*The increase of air temperature can alter the temperature of shallow permafrost due to strong land-atmosphere interactions. This, in turn, increases the thickness of permafrost active layer, thereby allowing soil water to move into the deeper soil layer.*

*In this study, the significant positive correlation between MRT and active layer thickness support previous findings showing that the MRT of permafrost catchments is highly dependent on the depth of the active layer due to the warming effects (Frampton and Destouni, 2015). From the mechanism perspective, the deepening of the active layer can increase the length of water flow pathway and reduce transport velocities due to a shift in flow direction from horizontal saturated groundwater flow to vertical flow infiltrate into deeper subsurface, thereby increasing water MRT in permafrost catchment (Frampton and Destouni, 2015)..*

Line 336: "These results also support the findings from previous studies in terms of a relationship between permafrost changes and residence time. In particular, (Wright et al., 2009) have stated that MRT of permafrost catchments is highly dependent on the annual development of the active layer." I could not find any information on MRT in the Wright paper.

Response: Very sorry for this incorrect cite. A new literature has been added in the text. We rephrased this statement in the text as follows (line 408-410):

*In this study, the significant positive correlation between MRT and active layer thickness support previous findings showing that the MRT of permafrost catchments is highly dependent on the depth of the active layer due to the warming effects (Frampton and Destouni, 2015).*

 "The larger precipitation corresponds to lower temperature, yielding a thinner active layer, which, in turn making the active layer water to be saturated sooner". Is this with respect to this study? This is no a general or predicted finding.

Response: Thank you for the comment. Our aim is to emphasize that the effects of precipitation on the development of permafrost active layers, thus, affecting MTR variability. Previous studies have found that increased precipitation thinned permafrost active layer on the Tibetan Plateau (Zhou et al., 2021; Luo et al., 2020). Thus, we rephrased this statement in the text as follows (line 415-418):

*In central TP, the increase in precipitation thinned the permafrost active layer by decreasing soil heat flux, thereby cooling the soil and alleviating permafrost degradation (Zhou et al., 2021; Luo et al., 2020). This phenomenon subsequently triggered more water to rapidly flow into the river channel in the form of surface runoff, thereby, reducing the MRT of catchment water in the end.*

 "Interestingly, we found that the stream and supra-permafrost water MRT are both negatively correlated with NDVI ($R^2$= 0.29 and 0.53, respectively)" The entire issue of linking MRT to 'factors' in a regression analysis is flawed. Processes and explanations must be provided. Why would this be? There is some speculation but this could easily be spurious.

Response: Thank you for the comment. We re-examined the regression analysis results and added the trend direction and significance level to the Table 5. We have found that changes of the active layer affect the water MRT variability. Moreover, the decline of vegetation coverage induced increases of soil temperature and moisture, in which case they accelerated the active soil thawing and thickened the permafrost active layer (Wang et al., 2012). Therefore, we believed that changes in vegetation coverage may influence the water MRT by changing the thickness of permafrost active layer. We rephrased this statement in the revised manuscript as follows (line 427-431):

*The decline of vegetation coverage elevated soil temperature and moisture, which in turn accelerated the permafrost thawing and thickened the active layer (Wang et al., 2012). Moreover, water MRT variability can be influenced by the development of the active layer; therefore, we believed that the effect of vegetation on MRT is also driven by the variations of thickness of permafrost active layer.*

 "However, it remains unclear whether a positive feedback mechanism exists between vegetation and permafrost active layer changes or not." There is considerable literature on this that should be referenced.

Response: We agree with the reviewer's comment. In the previous analysis, we found that the effect of vegetation on MRT variability is driven by changing the thickness of permafrost active layer. The length of residence time may hinder or promote plants growth, because water shortage in soil layer may occur when water MRT is too short, but longer MRT can hamper the water infiltration consequently to root anoxia, thus, affecting the plant growth (Ma et al., 2019). However, we do not have sufficient evidence

to demonstrate whether longer retention times or thicker active layers affect vegetation growth. Thus, this sentence has been deleted in the revised manuscript.

Line 365: "Moreover, the optimum residence time for vegetation growth should be elucidated in future studies as well." I am unsure as to what the authors mean.

Response: Thank you for the comment. Likewise, we do not have sufficient evidence to show whether longer retention times or thicker active layers affect vegetation growth. Thus, this sentence has been deleted in the revised manuscript.

Line 371: "It can be deduced that the estimated MRT of supra-permafrost water is valuable for evaluating the extent of permafrost degradation. Most importantly, it can be used to infer the effects of long-term climate, permafrost changes, and vegetation on the hydrologic regime in permafrost regions." This sentence is clearly wishful thinking and I am not sure the authors have shown this at all. If they have, they need to suggest how and why and what the implications are.

Response: We agreed with the reviewer's comment. MRT is a fundamental indicator of hydrological function within catchment. Through the previous analysis of the manuscript, in the permafrost watershed of the TP, the variation of permafrost active layer is an important factor in controlling water MRT. At the same time, both climatic and vegetation factors affect the MRT by influencing the development of active layer. Therefore, we can explore the changes in hydrological processes in permafrost watersheds by assessing water MRT in the context of climate and environmental change. Thus, we rephrased this statement in the revised manuscript as follows (line 437-439):

*Considering that MRT is a fundamental descriptor of hydrological function within catchment (Shah et al., 2017; Mcguire and Mcdonnell, 2006). Therefore, changes in hydrological processes in permafrost watersheds can be investigated by assessing water MRT in the context of climate and environmental change.*

Table 5. The data in this table is highly specific and incorrectly applies regression methods within and among data sets.

Response: Thank you for the suggestion. We re-examined the regression analysis results and added the trend direction and significance level to the Table 4 (line 342).

*Table 4. Correlations between active layer thickness, soil temperature, air temperature, precipitation, NDVI, and MRT. x indicates the factor as the independent variable.*

| | | Regression based on mean MTT (days) | | | Regression based on MTT uncertainty (days) | | |
|---|---|---|---|---|---|---|---|
| | Factor | Regression equation | R² | Sig | Regression equation | R² | Sig |
| | ALT | y=4.23exp(0.24x)+48.62 | 0.11 | P>0.05 | y=1.43exp(0.63x)+150.68 | 0.44 ↑ | **P<0.05** |
| Supra-permafrost | ST | y=929x+1299 | 0.87 ↑ | **P<0.05** | y=956x+1337 | 0.78 ↑ | **P<0.05** |
| water | AT | y=752x+4307 | 0.69 ↑ | **P<0.05** | y=767x+4283 | 0.67 ↑ | **P<0.05** |
| | P | y=-1.89x+1169 | 0.58 ↓ | **P<0.01** | y=3479exp(-0.005x)-125 | 0.64 ↓ | **P<0.01** |

| | | | | | | |
|---|---|---|---|---|---|---|
| | *NDVI* | $y=0.103x^{-4.405}$ | *0.51* | *P=0.07* | $y=0.143x^{-4.286}$ | *0.65 ↓* | *P<0.01* |
| *Stream water* | *ALT* | $y=4.67exp(1.11x)+71.56$ | *0.81 ↑* | *P<0.01* | $y=3.02exp(0.37x)+59.23$ | *0.59 ↑* | *P<0.01* |
| | *ST* | $y=(1.7x+1.35)^{-0.26}$ | *0.81 ↑* | *P<0.001* | $y=3.79E+14exp(35.47x)+76$ | *0.79 ↑* | *P<0.001* |
| | *AT* | $y=51x+372$ | *0.05* | *P>0.05* | $y=42x+325$ | *0.01* | *P>0.05* |
| | *P* | $y=4.94E+5(x^{-1.77})$ | *0.47↓* | *P<0.01* | $y=1.67E+10(1.7x)^{-3.25}+52.6$ | *0.32 ↓* | *P<0.01* |
| | *NDVI* | $y=1.422x^{-2.394}$ | *0.20↓* | *P<0.05* | $y=1.50x^{-2.384}$ | *0.26 ↓* | *P<0.05* |

*Note: ALT = active layer thickness, ST = Soil temperature (℃), AT = air temperature (℃), P = Precipitation (mm), NDVI= normalized differential vegetation index; Sig indicates statistical significance; ↑ and ↓ indicates significant trend of increase and decrease, respectively; Bold font indicates that it passed significance test of 0.05.*

References: The authors reference largely literature from China when discussing general permafrost hydrological knowledge. While I am not dismissing any of this work, suggesting that permafrost acts as an aquiclude, then citing Gao et al. 2021. This is not 'new information' and has been identified for many decades in the North American and Russian literature. Perhaps it has also been long-identified in the Chinese literature, and I would suggest the authors here and elsewhere cite appropriate historical works as opposed to ones focused on the TP unless the work is directly related to processes in the TP and not ones that are more general.

Response: Thank you for the suggestion. The appropriate historical literatures involving permafrost hydrology have been added in the revised manuscript accordingly as follows:

1. Woo, M.K., 1990. Consequences of climatic change for hydrology in permafrost zones. Journal of Cold Regions Engineering, 4(1):15-20. (line 29, 421)

2. Hinzman, L.D., Kane, D.L., Woo, M. K., 2005. Permafrost Hydrology, Encyclopedia of Hydrological Sciences. (line 29, 421)

3. Woo, M.K., Kane, D.L., Carey, S.K., Yang, D.,: Progress in permafrost hydrology in the new millennium. Permafrost and Periglacial Processes, 19(2): 237-254, 2008. (line 37)

[revised manuscript text omitted]

Sugimoto, A., Naito, D., Yanagisawa, N., Ichiyanagi, K., Kurita, and N.: Characteristics of soil moisture in permafrost observed in East Siberian taiga with stable isotopes of water, Hydrological Prochydrological Processesesses, 17, 1073-1092, https://doi.org/10.1002/hyp.1180

2003a.

Sugimoto, A., Naito, D., Yanagisawa, N., Ichiyanagi, K., Kurita, N., Kubota, J., Kotake, T., Ohata, T., Maximov, T. C., and Fedorov, A. N.: Characteristics of soil moisture in permafrost observed in East Siberian taiga with stable isotopes of water, Hydrological Processes, 17, 1073-1092, https://doi.org/10.1002/hyp.1180, 2003b.

Tetzlaff, D., Birkel, C., Dick, J., Geris, J., and Soulsby, C.: Storage dynamics in hydropedological units control hillslope connectivity, runoff generation, and the evolution of catchment transit time

distributions, Water Resources Research, 50, 969-985, https://doi.org/10.1002/2013WR014147, 2014.

Tetzlaff, D., Piovano, T., Ala-Aho, P., Smith, A., Carey, S. K., Marsh, P., Wookey, P. A., Street, L. E., and Soulsby, C.: Using stable isotopes to estimate travel times in a data-sparse Arctic catchment: Challenges and possible solutions, Hydrological Processes, 32, 1936-1952, https://doi.org/10.1002/hyp.13146, 2018.

Throckmorton, H. M., Newman, B. D., Heikoop, J. M., Perkins, G. B., Feng, X., Graham, D. E., O'Malley, D., Vesselinov, V. V., Young, J., Wullschleger, S. D., and Wilson, C. J.: Active layer hydrology in an arctic tundra ecosystem: quantifying water sources and cycling using water stable isotopes, Hydrological Processes, 30, 4972-4986, https://doi.org/10.1002/hyp.10883, 2016.

Uhlenbrook, S. and Hoeg, S.: Quantifying Uncertainties in Tracer-Based Hydrograph Separations: A Case Study for Two-, Three- and Five-Component Hydrograph Separations in a Mountainous Catchment, Hydrological Processes, 17, 431-453, https://doi.org/10.1002/hyp.1134, 2003.

Wang, G., Guangsheng, L., Chunjie, L., and Yan, Y.: The variability of soil thermal and hydrological dynamics with vegetation cover in a permafrost region, Agricultural and Forest Meteorology, 162-163, 44-57, https://doi.org/10.1016/j.agrformet.2012.04.006, 2012.

Wang, L., Zhao, L., Zhou, H., Liu, S., Du, E., Zou, D., Liu, G., Xiao, Y., Hu, G., Wang, C., Sun, Z., Li, Z., Qiao, Y., Wu, T., Li, C., and Li, X.: Contribution of ground ice melting to the expansion of Selin Co (lake) on the Tibetan Plateau, The Cryosphere, 16, 2745-2767, 10.5194/tc-16-2745-2022, 2022.

Xia, C., Liu, G., Zhou, J., Meng, Y., Chen, K., Gu, P., Yang, M., Huang, X., and Mei, J.: Revealing the impact of water conservancy projects and urbanization on hydrological cycle based on the distribution of hydrogen and oxygen isotopes in water, Environmental Science and Pollution Research, 28, 40160-40177, 10.1007/s11356-020-11647-6, 2021.

Yang, Y., Wu, Q., Yun, H., Jin, H., and Zhang, Z.: Evaluation of the hydrological contributions of permafrost to the thermokarst lakes on the Qinghai–Tibet Plateau using stable isotopes, Global and Planetary Change, 140, 1-8, https://doi.org/10.1016/j.gloplacha.2016.03.006, 2016.

Yang, Y., Weng, B., Yan, D., Gong, X., Dai, Y., and Niu, Y.: A preliminary estimate of how stream water age is influenced by changing runoff sources in the Nagqu river water shed, Qinghai-Tibet Plateau, Hydrological Processes, 35, e14380, https://doi.org/10.1002/hyp.14380, 2021.

Zhou, Z., Zhou, F., Zhang, M., Lei, B., and Ma, Z.: Effect of increasing rainfall on the thermal—moisture dynamics of permafrost active layer in the central Qinghai—Tibet Plateau, Journal of Mountain Science, 18, 2929-2945, https://doi.org/10.1007/s11629-021-6707-5, 2021.

Zhu, X., Wu, T., Zhao, L., Yang, C., Zhang, H., Xie, C., Li, R., Wang, W., Hu, G., Ni, J., Du, Y., Yang, S., Zhang, Y., Hao, J., Yang, C., Qiao, Y., and Shi, J.: Exploring the contribution of precipitation to water within the active layer during the thawing period in the permafrost regions of central Qinghai-Tibet Plateau by stable isotopic tracing, Science of The Total Environment, 661, 630-644, https://doi.org/10.1016/j.scitotenv.2019.01.064, 2019.

**Response to Anonymous Referee #2**

We are very grateful for your valuable and instructive comments and suggestions. We have thoroughly revised the manuscript by addressing all the comments point by point. The review comments are listed below and marked in blue, followed by the detailed responses marked in black. The sentences added in the revised manuscript were marked in red and italic.

Kind regards,

Xiaobo He

(on behalf of the co-authors)

One issue is that the authors argued in their response to Referee #1's comment (https://tc.copernicus.org/ preprints/tc-2022-17/) that this study considered the permafrost change effects on stream water ages beyond Song et al (2017) study, this argument did not convince me since the permafrost change data are not from this study region (detailed below). The permafrost temperature and active layer data should be representative. Anyway, this is a great study. The paper may be publishable after resolved those issues properly.

Response: Thank you for the comments and suggestions. The permafrost data we observed in our site is limited, which cannot support us to conduct a comprehensive regression analysis. Although the permafrost data (the soil active layer bottom temperature and active layer thickness) we used are regional average data, our study site was located in this region. Moreover, the data we currently have are in good agreement with the permafrost data reported in the Blue Book on Climate Change in China 2021, with $R^2=0.80$ and $P=0.000$ for active layer thickness and $R^2=0.56$ and $P=0.003$ for active layer bottom temperature, respectively. Thus, these regional data of permafrost can be used for regression analysis in our study site to a certain extent.

L56: Yang 2021 reference is missing in the reference list.

Response: Sorry for this mistake. We have added this (Yang et al., 2021) reference in the revised manuscript.

*Yang, Y., Weng, B., Yan, D., Gong, X., Dai, Y., and Niu, Y.: A preliminary estimate of how stream water age is influenced by changing runoff sources in the Nagqu river water shed, Qinghai-Tibet Plateau, Hydrological Processes, 35, e14380, https://doi.org/10.1002/hyp.14380, 2021.* (line 729-729)

L93 and Figure 2: At what depths were the soil moisture and temperature measured? Please clarify. Soil moisture and temperature can have large differences between the upper and deeper layer. Authors must specify the exact depth measured.

Response: Thank you for the suggestion. In our study, the soil moisture and temperature data were measured at depths of 5, 20, 40, 60, 100, 160, 220, and 300 cm. We have added this statement in the revised manuscript as follows:

*The soil moisture and temperature data were measured at depths of 5, 20, 40, 60, 100, 160, 220, and 300 cm.* (line 93-94)

Table 1: This table provides a lot data that seems not measured in this study. For example, according to L102-103, the soil active layer bottom temperature and active layer thickness are from the Blue Book on Climate Change in China 2021 (Fig 3.5 as I know). Those data are regional average data along the Kunlun Mountain-Liangdaohe section, which may not be suitable to represent the permafrost and active layer condition in this study area. More importantly, these data were used in the regressions in Table 4, which brings more uncertainty to the results. Please use the permafrost and active layer observation data in your site. If these data are unavailable, please at least clarify the data source in Table 1 and uncertainties in Table 4.

Response: Thank you for the comment and suggestion. The permafrost data we observed in our site is limited, which cannot support us to conduct a comprehensive regression analysis. Although the permafrost data (the soil active layer bottom temperature and active layer thickness) we used are regional average data, our study site was located in this region. Moreover, the data we have are in good agreement with the permafrost data reported in the Blue Book on Climate Change in China 2021, with $R^2$=0.80 and *P*=0.000 for active layer thickness and $R^2$=0.56 and *P*=0.003 for active layer bottom temperature, respectively. Thus, these regional data of permafrost can be used for regression analysis in our study site to a certain extent. Meanwhile, we clarified the data source again and added statement of uncertainty in the revised manuscript as follows:

*The soil temperature data (active layer bottom temperature) and active layer thickness of permafrost were obtained from the book of Blue Book on Climate Change in China 2021, which published long-term data, related to the permafrost along the Kunlun Mountain to the southern slope of Tanggula Mountain in the central TP (Cma Climate Change Centre, 2021).* (line 103-105)

*In addition, the permafrost data used in regression analysis are regional average data, which may increase the uncertainty of the regression analysis (Table 4).* (line 504-506)

L146: Two isotopic tracers can track three water sources. Only one isotope tracer is needed to perform two-source separation. So which isotope do you use in the two-component hydrograph separation?

Response: Thank you for the suggestion. In this study, the $\delta^{18}$O data were used in the two-component hydrograph separation. Because, in previous studies, $\delta^{18}$O have been widely used to segment river components. We have added this statement in the revised manuscript as follows:

*In this study, the $\delta^{18}$O data were used in the two-component hydrograph separation.* (line 148-149)

Section 2.5: Kirchner in 2016 (www.hydrol-earth-syst-sci.net/20/279/2016/ and its companion paper) showed that the young water fraction (Fyw) is also a useful descriptor of stream water age. Your dataset can estimate the Fyw reliably. Why this study uses MRT not Fyw?

Response: Thank you for the suggestion. Indeed, the young water fraction (Fyw) is also a useful descriptor of stream water age, especially in heterogeneous and nonstationary catchments. In this study, our catchment is a typical permafrost catchment, its underlying surface was relatively uniform with less landscape heterogeneity and characterized by rapid hydrological processes. The estimated MRTs have low bias in such catchment. Thus, MRT may be more intuitive and clear indication of water age than the proportion of young water within the permafrost catchment. Nevertheless, in future research, we might attempt to contrast the application between these two parameters in permafrost catchment.

L197: All the citations (Wang et al., 2022; Sugimoto et al., 2003; Throckmorton et al., 2016) are missing in the reference list.

Response: Thank you. We have checked all references in the manuscript.

Figure 7: Stream water in June 2018, June 2019, June & July 2020 are completely precipitation sourced. Why?

Response: Thank you for the comment. Because we did not observe the generation of supra-permafrost water in the sampling well in June 2018 and June 2019 due to the low air temperatures and thin active layer of permafrost, resulting in not much water within the permafrost. The monthly mixing diagram in June and July 2020 using the mean $\delta^{18}$O and $\delta$D showed that the isotope values in stream water are very close to the that of precipitation (Fig. 6). Thus, in these stages, the components of stream water were dominated by precipitation. These statements have been added in revised manuscript (line 266-267 and 278-279).

**References**

Yang, Y., Weng, B., Yan, D., Gong, X., Dai, Y., and Niu, Y.: A preliminary estimate of how stream water age is influenced by changing runoff sources in the Nagqu river watershed, Qinghai-Tibet Plateau, Hydrological Processes, 35, e14380, https://doi.org/10.1002/hyp.14380, 2021.

**Response to editor**

We are very grateful for your valuable comments. We have thoroughly revised the manuscript by addressing your comments. The review comments are listed below and marked in blue, followed by the responses marked in black. The sentences added in the revised manuscript were marked in red and italic. Kind regards,

Xiaobo He

(on behalf of the co-authors)

Thank you for submitting a revised version of your manuscript and replying to the comments from the referee. I have one final clarification that I would like you to make, following up on one of the referee comments and your response. You have now clarified that soil moisture and temperature were measured at several depths, however, Figure 2 shows only one line for soil moisture and one line for soil temperature - which depths are used in this figure, and in Figure 5?

Response: Thank you for the comments. The soil moisture and temperature data that we used in Figure 2 and Figure 5 were the average value over all soil layers. Because we believe that the average value can represent the average soil temperature and moisture of the whole soil profile. We have added this statement in Figure 2 and Figure 5 of the revised manuscript as follows:

[Figure]

*Figure 2. Temporal variation of soil moisture, soil temperature, air temperature, and precipitation from 2017 to 2020. The soil moisture and temperature data were the average value over all soil layers (5, 20, 40, 60, 100, 160, 220, and 300 cm).*

[Figure]

*Figure 5. Correlation analysis between stable isotopes and influencing factors. (a) and (b) correlations between stream water $\delta^{18}O$ with: soil temperature and moisture, respectively; (c) and (d) correlations between supra-permafrost water $\delta^{18}O$ with: soil temperature and stream water $\delta^{18}O$, respectively; (e) and (f) correlations between precipitation $\delta^{18}O$ with: stream and supra-permafrost water $\delta^{18}O$, respectively; Red line is the fitted line; Soil moisture and temperature data used in correlation analysis were normalized. "-" presents dimensionless. The soil moisture and temperature data were the average value over all soil layers (5, 20, 40, 60, 100, 160, 220, and 300 cm).*

**Response to editor**

Thank you for your suggestions. We have thoroughly revised the manuscript by addressing your suggestions. The review comments are listed below and marked in blue, followed by the responses marked in black. The statements added in the revised manuscript were marked in red and italic.

Kind regards,

Xiaobo He

(on behalf of the co-authors)

Thank you for adding these revisions, however, I'm a bit confused by your clarification about the soil temperature and moisture data. It seems to me highly unlikely that the mean temperature for all these sensors would be so close to the air temperature in the summer, since the temperature at the lower depths should be much lower (even below zero for the deepest sensors?). To me, the soil temperature (and moisture) look realistic for the top sensor (5 cm depth), but not as a mean of all sensors. Could you please double check with your data that this statement is indeed correct and/or provide an explanation to how the mean value of these sensors can exhibit this variation over the year?

The easiest way, in my opinion, would be to include the data from all depths in the data file (figshare). There is also a typo in the y-axis label for the soil moisture (going from 0 to 0.6 percent) in figure 2. I assume that the scale should go from 0 to 60 percent.

Lastly, to further improve figure 2, please clarify if the data is daily, which will also help readers to interpret the precipitation data, which reaches up 20 mm, which is a high precipitation intensity for one day.

Response: Thank you for the suggestions and sorry for the error in the y-axis label for the soil moisture in Figure 2. We have double-checked the soil temperature data and confirmed that they are correct. There are several reasons why the average soil temperature for all sensors was close to the air temperature in warm season:(1) with the thickening of permafrost active layer in recent years, the soil temperature in the bottom (300 cm) can be greater than 0°C in warm season. (2) By analyzing the hourly variation of air and soil temperature, we found that the surface temperature (5-20 cm) in the daytime was higher than the air temperature (see Figure 1), which because the soil absorbs more solar radiation than the air during the daytime. But the air temperature at night was lower than that of soil due to permafrost layer has the function of heat preservation. Thus, the average soil temperature for all sensors was close to the air temperature in warm season. Moreover, the comparison with other sites we observed also shows these similar findings.

Based on the above analysis, we believe that the soil temperature data is correct. Meanwhile, we have updated and uploaded the soil moisture and temperature data for each measured soil layer in figshare (https://figshare.com/articles/dataset/tc-2022-17datasetV2_xlsx/21500229).

[Figure]

**Figure 1**. Hourly variation of soil temperature at different depths, taking the data from August 1 to August 7, 2017 as an example.(Not showed in the manuscript)

We have clarified that the data in Figure 2 is daily data. Figure 2 has been modified in the revised manuscript.

[Figure]

*Figure 2. Daily variation of soil moisture, soil temperature, air temperature, and precipitation from 2017 to 2020. The soil moisture and temperature data were the daily average value of all the sensor data in the soil layer.*